# Simultaneous single-cell three-dimensional genome and gene expression profiling uncovers dynamic enhancer connectivity underlying olfactory receptor choice

Honggui Wu [1,2,6], Jiankun Zhang [1,2,6], Fanchong Jian [1,2,3,6], Jinxin Phaedo Chen [5], Yinghui Zheng[1], Longzhi Tan [4] ✉ & X. Sunney Xie[1,2] ✉

The simultaneous measurement of three-dimensional (3D) genome structure and gene expression of individual cells is critical for understanding a genome's structure–function relationship, yet this is challenging for existing methods. Here we present 'Linking mRNA to Chromatin Architecture (LiMCA)', which jointly profiles the 3D genome and transcriptome with exceptional sensitivity and from low-input materials. Combining LiMCA and our high-resolution scATAC-seq assay, METATAC, we successfully characterized chromatin accessibility, as well as paired 3D genome structures and gene expression information, of individual developing olfactory sensory neurons. We expanded the repertoire of known olfactory receptor (OR) enhancers and discovered unexpected rules of their dynamics: OR genes and their enhancers are most accessible during early differentiation. Furthermore, we revealed the dynamic spatial relationship between ORs and enhancers behind stepwise OR expression. These findings offer valuable insights into how 3D connectivity of ORs and enhancers dynamically orchestrate the 'one neuron–one receptor' selection process.

Three-dimensional (3D) genome organization lays the physical foundation for gene expression and gene regulation[1–6]. Understanding the intricate relationship between genome architecture and gene expression necessitates the development of advanced techniques to simultaneously measure these two modalities with high sensitivity from the same cell[7–12]. Existing methods have severe limitations. Currently, imaging-based methods can only measure a limited number of genomic loci (1,000–3,660, namely every 1–3 Mb) and transcripts (70–1,000 genes) and therefore lack a genome-wide view[7–11]. The published sequencing-based methods, HiRES, had limited sensitivity (~0.3 million contacts per cell) because genomic DNA was damaged during reverse transcription, captured only nuclear RNAs because the cytoplasm was destroyed during the procedure and only detected the 3′ end of the transcript[12]. In addition, HiRES must be performed with a large number of cells, prohibiting analysis of low-input samples.

Here we report Linking mRNA to Chromatin Architecture (LiMCA), a sequencing-based method that simultaneously profiles single-cell 3D genome structure and full-length transcript information. In particular, LiMCA physically separates the nucleus and the cytoplasm of the same cell for measuring the 3D genome and transcriptome, respectively, and therefore does not compromise the detection sensitivity and performance of each modality.

[1]Biomedical Pioneering Innovation Center (BIOPIC), and School of Life Sciences, Peking University, Beijing, China. [2]Changping Laboratory, Beijing, China. [3]College of Chemistry and Molecular Engineering, Peking University, Beijing, China. [4]Department of Neurobiology, Stanford University, Stanford, CA, USA. [5]Present address: Department of Microbiology, Tumor and Cell Biology, Karolinska Institute, Stockholm, Sweden. [6]These authors contributed equally: Honggui Wu, Jiankun Zhang, Fanchong Jian. ✉e-mail: tttt@stanford.edu; sunneyxie@biopic.pku.edu.cn

To demonstrate the biological insights that LiMCA can generate, we applied LiMCA to the mouse olfactory system. Understanding how the 'one neuron–one receptor' paradigm is established during olfactory sensory neuron (OSN) development is a long-standing pursuit of the field. There are more than 1,000 olfactory receptor (OR) genes, which are presented as gene clusters distributed across 18 chromosomes in the mouse genome[13]; however, each mature OSN expresses only one OR gene out of such a large repertoire in a monoallelic and seemingly stochastic manner[14]. Recent bulk and single-cell chromosome conformation capture (3C/Hi-C) studies showed that OSNs establish strong and specific inter-chromosomal interactions between OR gene clusters, which are heterochromatically modified to assure the complete silencing of OR genes[15,16]. Such OR–OR gene cluster interactions bring multi-chromosomal OR enhancers (termed the 'Greek Islands' (GIs)) together to form a super-enhancer hub[17,18], which was proposed to activate the singular chosen OR gene, forging the 'silence all and activate one' model.

However, this model fails to address several unresolved issues. First, during OSN development, progenitors transiently express random sets of OR genes[19,20]. Additionally, the onset of multigenic OR expression precedes the formation of repressive OR–OR gene compartments. Furthermore, each OSN forms multiple enhancer aggregates, which means that simply being associated with enhancer hubs is insufficient to fully account for the singular OR gene. Unfortunately, existing bulk and single-cell techniques are unable to resolve these mysteries due to the lack of OR expression information and an inability to isolate a population expressing a random set of OR genes. Ideally, a technique that can simultaneously measure OR gene expression and 3D genome organization in the same cells would be necessary to elucidate how OR gene selection process is initiated and proceeded.

Using LiMCA and in combination with single-cell chromatin accessibility and a gene expression landscape of the developing OSNs, we gained an unprecedented view of how the accessibility of OR enhancers is regulated and how the association with multi-chromosomal enhancers regulates the stepwise OR gene selection from multigenic OR activation to singular OR gene determination.

## Results

### Development of LiMCA

To enable simultaneous measurement of transcriptional activity and chromatin architecture in the same cell with high sensitivity, we employed a strategy utilizing physical separation of cytoplasm (mRNA) and nucleus (chromatin). This procedure has been used in single-cell multi-omics technologies[21–23]. Specifically, the separated cytoplasm was subjected to Smart-seq2 amplification for transcriptome analysis[24], while the nucleus was proceeded to conventional chromosome conformation capture procedure[25] that included crosslinking, restriction enzyme digestion and proximity ligation (Fig. 1a and Methods). To further increase chromatin contact detection in single cells, we adopted our high-coverage transposon-based whole-genome amplification (WGA) method, META[26], to amplify the resulting nucleus. Then the messenger RNA library and 3C library were sequenced and integrated to obtain both modalities (Fig. 1a).

To evaluate whether LiMCA precisely captures high-order genome structure, we performed a proof-of-concept experiment on GM12878, a well-studied female human lymphoblastoid cell line with an extensively characterized genome structure[27]. LiMCA detected a median of 1.08 million unique chromatin contacts per cell ($n$ = 220, s.d. = 470,000, minimum = 130,000, maximum = 2.79 million) (Supplementary Table 1), which is comparable to our previously developed high-sensitivity single-cell Hi-C method, Dip-C[26] (Extended Data Fig. 1a). The composition of contacts is similar to Dip-C, with a greater proportion of short-range (<20 kb) and lower long-range (>20 kb) intra-chromosomal contacts. Ensemble chromatin interaction profiles (merged from 220 individual cells, referred to as ensemble LiMCA)

exhibited high concordance with a bulk in situ Hi-C contact map across various resolutions ranging from compartments to topologically associating domains and chromatin loops (Fig. 1b,c and Extended Data Fig. 1d–h). Furthermore, the gene expression profile of ensemble LiMCA displayed a high correlation with bulk RNA-seq data generated from the same cell line (Fig. 1d). Therefore, we concluded that LiMCA faithfully captures both the genome architecture and gene expression.

To examine the robustness of LiMCA, we further applied it to three different human cell lines (K562, eHAP and BJ) as well as mouse olfactory epithelium. This additional cell line dataset further validated our technique (Extended Data Fig. 1i and Extended Data Fig. 2a). Subsequently, we performed a comprehensive comparative analysis against published datasets, including HiRES[12] (single-cell joint Hi-C-RNA), Dip-C[18,26,28] (scHi-C) and single-cell RNA sequencing (scRNA-seq) data. Our results demonstrated that LiMCA detected a substantially higher number (2–5 folds) of contacts than HiRES and tissue datasets obtained through Dip-C (Fig. 1e, left). Furthermore, LiMCA exhibited a comparable number of genes detected when compared to Smart-seq, while surpassing the number of genes identified by HiRES and droplet-based scRNA-seq methods (Fig. 1e, right, and Extended Data Fig. 1c). Notably, LiMCA not only offers enhanced sensitivity but also provides full-length transcript information, in contrast to HiRES, which solely captured the 3′ end of genes. Therefore, LiMCA is capable of measuring both chromatin interactions and gene expression at high sensitivity and consistently performs well across diverse cell types.

We then performed clustering based on chromatin interaction (scA/B value; Methods) or gene expression profiles offered by LiMCA to evaluate its accuracy in distinguishing cell types. We found that both modalities clearly separated the four cell types (Fig. 1f and Extended Data Fig. 2b). To confirm accuracy, we calculated scA/B values for cell-type-specific marker genes, which showed specific enrichment in corresponding cell types (Extended Data Fig. 2c,d), consistent with our previous work[28]. Hi-C 'structural typing' identified an additional cluster containing cells from all four cell types, which belongs to the metaphase (Extended Data Fig. 2e–g). This is in line with the knowledge that the chromosome undergoes a homogeneous folding state during mitosis irrespective of cell type[29]. Furthermore, LiMCA accurately detected cell-type-specific chromatin loops (Extended Data Fig. 2h,i). Thus, we established a single-cell multi-omics assay that simultaneously measures genome-wide chromatin interactions and transcriptome-wide gene expression in hundreds of single cells.

### Relationship between gene expression and 3D genome structure

With our previously developed Dip-C algorithm[26], we showed that about 31% of GM12878 cells (68 of 220, with root-mean-square-deviation (r.m.s.d.) < 1.5) and 52% of eHAP cells (22 of 42, haploid cells) faithfully yielded 3D genome structures at a high resolution of 20 kb (Fig. 1a, Extended Data Fig. 1j and Methods). The pairwise 3D distance matrix obtained from individual single-cell structures exhibited a strong agreement with the ensemble and bulk contact maps (Extended Data Fig. 3a). With such high-resolution structures, we were able to pinpoint the spatial position of expressed genes in the nucleus.

To investigate the relationship between gene expression and chromatin structure, we focused on the *NFKB1* gene, a critical transcription factor for B cell development and function. We sorted GM12878 cells into two groups based on NFKB1 expression level and compared ensemble contact maps. Our findings revealed that highly expressed NFKB1 interacts more frequently with an upstream enhancer (Fig. 1g). This observation was further validated by downsample and random sample control analysis (Extended Data Fig. 3c–h). Similar results were observed for other genes analyzed (Extended Data Fig. 3i,j), demonstrating that gene expression dynamics are associated with changes in chromatin structure.

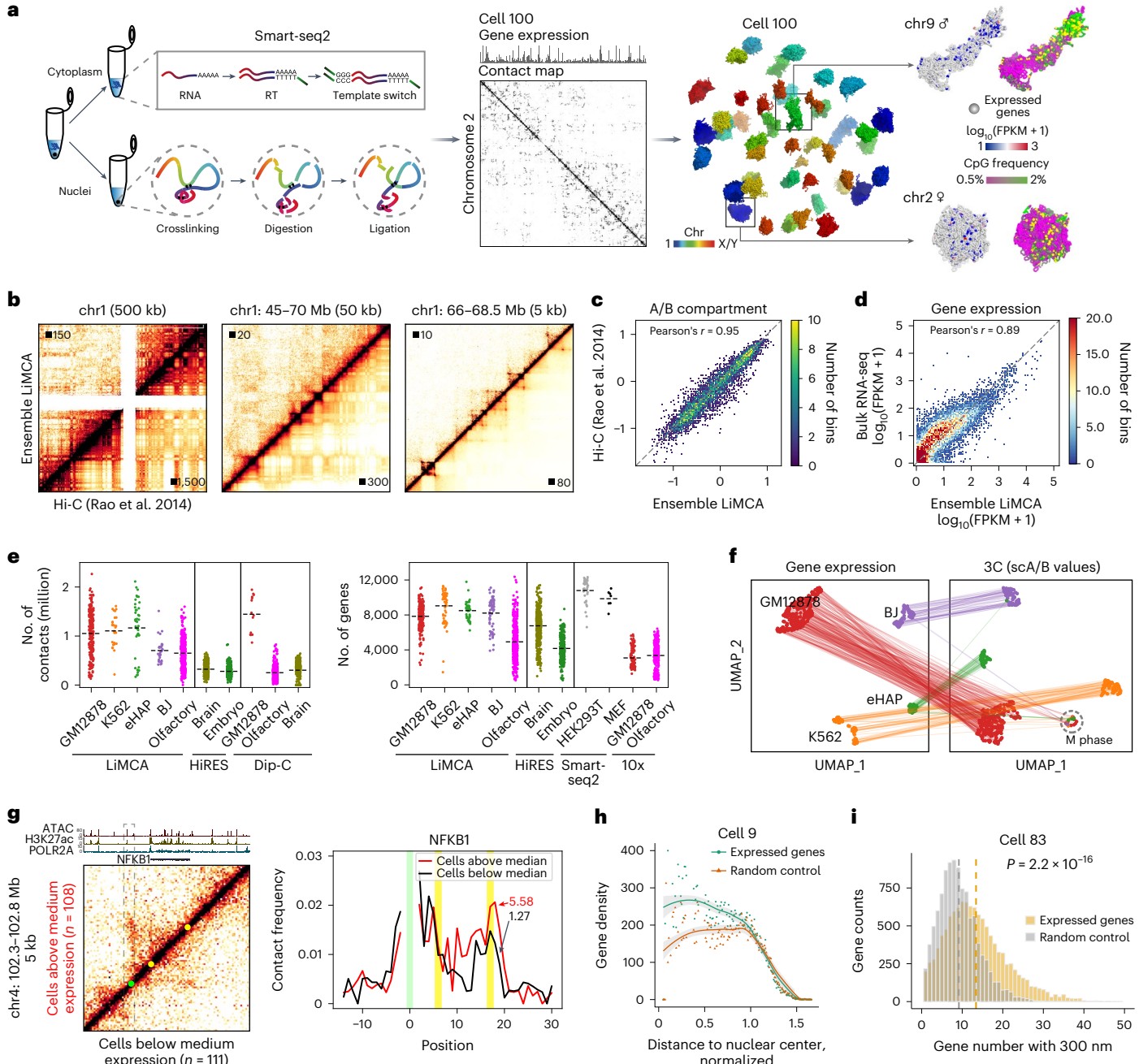

**Fig. 1 | Development of LiMCA. a**, Left, schematics of the LiMCA procedure. Right, the 20-kb resolution 3D genome structure of a representative cell; expressed genes are projected. RT, reverse transcription. **b**, Comparison of ensemble LiMCA and bulk Hi-C. The maximum intensity is indicated. **c**, Scatter-plot of the first eigen value between ensemble LiMCA and bulk Hi-C at 100-kb resolution. **d**, Scatter-plot of expression level (FPKM) between bulk RNA-seq (ENCODE ENCFF897XES) and combined expression profile of LiMCA. **e**, The median contact number of LiMCA (GM12878, 1.08 million, $n = 220$; K562, 1.14 million, $n = 28$; eHAP, 1.30 million, $n = 42$; BJ, 678,000, $n = 32$; olfactory, 652,000, $n = 411$) (left) is compared to HiRES (brain, 304,000, $n = 399$; embryo, 264,500, $n = 300$ (random sampled)) and Dip-C (GM12878, 1.45 million, $n = 14$; brain, 333,000, $n = 300$ (random sampled); olfactory, 252,000, $n = 409$). The median detected gene number of LiMCA (GM12878, 7,954, $n = 221$; K562, 9,806, $n = 63$; eHAP, 8,588, $n = 42$; BJ, 8,911, $n = 63$; olfactory, 4,528, $n = 411$) (right) is compared to HiRES (brain, 6,966, $n = 399$; embryo, 4,058, $n = 300$

(random sampled)), Smart-seq2 (HEK293T, 11,169, $n = 35$; MEF, 10,173, $n = 7$) and 10x chromium (olfactory, 3,556, $n = 300$ (random sampled of this study); GM12878, 2,754, $n = 100$ (random sampled)). **f**, Uniform Manifold Approximation and Projection (UMAP) embedding of four profiled cell lines based on gene expression profiles (left) or single-cell A/B values (right). The same cells are connected with lines. **g**, Contact matrices (left) around NFKB1, representing ensemble Hi-C data of NFKB1-high group (top left) and NFKB1-low (bottom right). Normalized contact frequency plot (right), centered at the NFKB1 upstream enhancer. The green and yellow dot/line indicates the position of the candidate enhancer and the transcription start site (TSS) and the transcription termination site (TTS) of NFKB1, respectively. **h**, Radial distribution of gene density; nucleus is sliced to 0.01 thickness. The error bands represent the 95% confidence interval (CI). **i**, Histograms show the distribution of cluster size within 300 nm for expressed genes and random control. Data were analyzed by a two-sided Mann–Whitney $U$-test.

The positioning of genes within the nucleus, such as the radial positioning and the association with nuclear landmarks, is known to influence their expression[30]. To examine how gene positioning influences gene expression in single cells, we explored the spatial distribution of expressed genes within the nucleus. Our analysis revealed that expressed genes have a higher density in the nuclear interior and

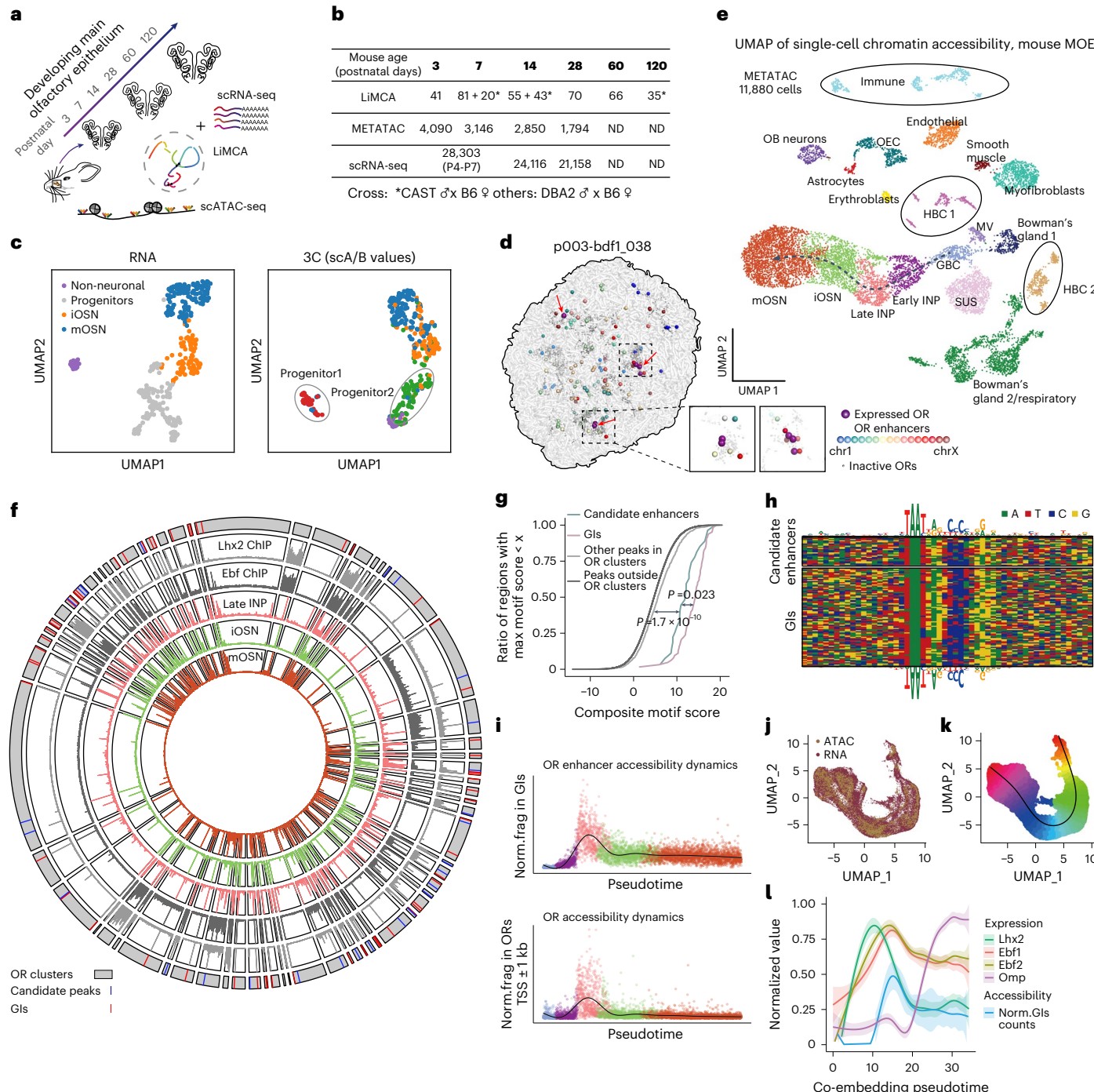

**Fig. 2 | Multi-omics profiling of the developing OSNs at single-cell resolution. a**, Schematics of overall experimental design. **b**, Summary of number of single cells of the multimodal atlas at each developmental stage. ND, not done. **c**, UMAP visualization of single cells based on gene expression (left) and chromatin structure (scA/B compartment values) (right) of the LiMCA multi-omics dataset. **d**, The 3D positioning of OR genes and enhancers in a progenitor cell expressing multiple ORs. Enhancers and ORs are shown as spheres with radii of 0.9 and 0.27 particle radii (54 and 18 nm), respectively. Expressed ORs were displayed with radii of 1.2 particle radii (72 nm). **e**, Single-cell chromatin accessibility atlas of developing MOE. HBC, horizontal basal cell. OEC, olfactory ensheathing cell; OB, olfactory bulb; SUS, sustentacular cell; MV, microvillous cells. **f**, Circos plot showing 27 newly identified and 63 known OR enhancers; five sectors from interior to periphery were ensemble

METATAC signals of INP, iOSN and mOSN, and Lhx2 and Ebf ChIP-seq track of mOSN from elsewhere[33]. **g**, Cumulative curves of score matching Lhx2/Ebf composite motif for GIs, newly identified OR enhancers, other ATAC peaks within OR clusters and ATAC peaks outside of OR clusters. A two-tailed Kolmogorov–Smirnov test was performed. **h**, Alignment of composite motif sequences from candidate enhancers (top) and known enhancers (bottom). **i**, The normalized chromatin accessibility dynamics of OR enhancers and OR genes along OSN differentiation from GBC to mOSN. **j**, UMAP shows the co-embedding of METATAC and scRNA-seq data of the OSN lineage from GBC to mOSN. **k**, UMAP showing the pseudotime trajectory of the integrated data. **l**, Line-plot shows the expression changes of Lhx2, Ebf and Omp, and chromatin accessibility changes of OR enhancers from GBC to mOSN based on the integrated data. The error bands represent 95% CI.

more neighbors at a given 3D distance than randomly selected controls (Fig. 1h,i and Extended Data Fig. 3k,l). Though population average radial position negatively correlated with expression level (Extended Data

Fig. 3o), this is not observed in single cells (Extended Data Fig. 3m,n). Notably, our analysis may be confounded by the fact that we could not distinguish the allele-specific gene expression.

## A multi-omics atlas of the developing OSNs

With our established multi-omics assay, we then sought out to explore how the 'one neuron–one receptor' rule is established. Traditional bulk assays and imaging-based methods are unable to delineate this process due to the fact that each progenitor cell transiently expresses a random set of 5–15 OR genes during the progenitor stage[19,20], followed by a single OR gene outcompeting others during OSN maturation; however, our technique allows for simultaneous probing of both OR gene expression and 3D chromatin structure, providing an unprecedented insight into this complex process.

We created a joint 3D genome and gene expression multi-omics atlas of the developing OSNs with LiMCA, consisting of 411 cells from the mouse main olfactory epithelium (MOE) across six time points (postnatal day 3, 7, 14, 28, 60 and 120) (Fig. 2a,b). We obtained an average of 650,000 unique chromatin contacts per cell (s.d. = 298,000, minimum = 119,000 and maximum = 2.8 million) (Supplementary Table 2), of which 224 (54%) have high-quality 20-kb resolution 3D structures (r.m.s.d. ≤ 1.5; Supplementary Table 2). For gene expression, we detected a median of 4,528 genes per cell (Supplementary Table 2).

Upon embedding based on gene expression (Fig. 2c), we identified four clusters in RNA embedding: non-neuronal, progenitors, immature OSNs and mature OSNs (Fig. 2c, left, and Extended Data Fig. 4a). When examining the Hi-C embedding, we observed that the progenitors in RNA embedding were split into two distinct clusters, referred to as progenitor1 and progenitor2 (Fig. 2c, right, and Extended Data Fig. 5a,b). We further validated the separation of progenitor1 and progenitor2 by integrating our published mouse MOE data[18], excluding the potential influence of mouse lines or contact numbers (Extended Data Fig. 5c,d).

We observed a continuous trajectory in OSN genesis, from progenitors to immature OSNs and finally to mature OSNs (Extended Data Fig. 4c,d). As expected, our LiMCA profiles recapitulated known characteristic chromatin reorganization during OSN maturation, including gradually increased chromosomal intermingling, OR–OR gene interaction and enhancer–enhancer interactions (Extended Data Fig. 5e–i). With the expression profiles of OR genes, we were able to reveal the spatial relationship between expressed OR genes and OR enhancers (Fig. 2d).

To comprehensively understand the underpinning chromatin state of OR enhancers along OSN development, we additionally generated a single-cell chromatin accessibility and gene expression atlas of the developing mouse MOE with our high-sensitivity METATAC[31] and droplet-based scRNA-seq, consisting of 11,880 cells and 73,577 cells (Fig. 2e and Extended Data Fig. 8b,c), respectively. We utilized the scRNA-seq atlas as a reference to annotate the cell types in our METATAC atlas (Extended Data Fig. 6g). The atlas allowed us to capture the dynamics of chromatin accessibility and gene expression throughout OSN development. For assay for transposase-accessible chromatin (ATAC), we detected a median of 66,000 ATAC fragments per cell (Extended Data Fig. 6b), and the gene expression yielded a median of 3,346 genes (8,651 unique molecular identifiers (UMIs)) per cell (Extended Data Fig. 8a).

Our datasets validated the changes in cell type composition between multi-potent progenitor cells and developing OSNs during the first postnatal month of development (Extended Data Fig. 6e and 8h). Notably, our dataset precisely recapitulated known cell types in MOE and their marker genes (Fig. 2c, Extended Data Fig. 6c–g and Extended Data Fig. 8b–d). Specifically, both of our single-cell chromatin accessibility and gene expression atlases captured the continuous developmental trajectory of OSNs from globose basal cell (GBC) to immediate neuronal precursor (INP), then to immature OSN (iOSN) and mature OSN (mOSN) (Fig. 2e and Extended Data Fig. 8b). Our high-resolution single-cell chromatin accessibility atlas offers a new opportunity to understand the epigenetic regulatory mechanism underlying multiple lineage specification of MOE.

## Chromatin accessibility dynamics of OR enhancers

Using our high-resolution single-cell chromatin accessibility atlas, we identified 27 new enhancers (Fig. 2f and Supplementary Table 3) according to previous definitions[32,33], which were located within OR gene clusters, exhibited ATAC peaks in mOSN, co-bound by LHX2 and EBF (Fig. 2f and Extended Data Fig. 7c–e) and contained the characteristic composite motif of LHX2 and EBF (Fig. 2g,h). The comprehensive characterization of OR enhancers proves that almost all OR gene clusters harbor at least one enhancer, implying the critical role of *cis*-enhancer in the regulation of OR gene expression. The absence of identified enhancers in certain small clusters may be due to the low abundance of OSNs expressing these specific OR genes.

We then analyzed the chromatin accessibility dynamics of OR genes and OR enhancers during OSN differentiation. Using our METATAC dataset, we performed pseudotime analysis to trace the developmental lineage from the GBC stage to mOSNs (Fig. 2e and Methods). Our findings revealed that OR genes initially had a closed state at the GBC stage, followed by a pervasive accessibility state at the late INP stage, corresponding to multigenic OR expression. During OSN maturation, OR genes returned to a fully inaccessible state (Fig. 2i, bottom), even more closed than non-OSN cell types (Extended Data Fig. 7b), indicating robust OR gene repression. OR enhancers were completely inaccessible at GBCs but rapidly reached peak accessibility at the late INP stage before decreasing to a lower level as OSNs matured to mOSN (Fig. 2i, top). Analysis of master transcription factors (TFs) of OR enhancers with our single-cell gene expression atlas showed that Lhx2/Ebf expression followed similar dynamics as OR enhancers along OSN development (Extended Data Fig. 8g). To further determine the temporal relationship between *Lhx2* expression and OR-enhancer accessibility, we integrated METATAC and scRNA-seq data by extracting the continuous developmental trajectory from GBC to mOSN (Fig. 2j, Extended Data Fig. 8i and Methods). The integrated pseudotime analysis confirmed that *Lhx2/Ebf* expression clearly precedes OR-enhancer activation (Fig. 2k,l). These results suggest that the accessibility change of OR enhancers is elicited by LHX2, consistent with *Lhx2* knockout eliminating GI accessibility in mOSNs[33].

Overall, our study reveals that LHX2-activated OR enhancers reach their highest accessibility at the late INP stage, creating a highly activated environment for OR gene expression and explaining the multigenic OR activation at this stage. As OSN maturation, multiple OR genes initially activated in progenitors are silenced, leaving only one active OR gene. At the same time, the accessibility of OR genes and OR enhancers decreases as OSNs further matured, ensuring singular OR gene expression and silencing of excess ORs in mOSNs.

---

**Fig. 3 | Stepwise OR determination observed with single-cell joint profiling of chromatin architecture and gene expression. a**, Cells were classified into three stages based on the total OR expression level and the ratio of OR with highest expression level of the developing OSNs (progenitor, iOSN and mOSN). **b**, The 3D positioning of ORs and enhancers in a representative cell at multigenic OR activation stage with expressed ORs depicted in detail, revealing that *cis*-enhancer activates their expression. **c**, Histogram summarizing the percentage of *cis*-enhancers and *trans*-enhancers within 150 nm of expressed ORs for three OR expression stages. **d**, The 3D positioning of ORs and enhancers in a representative cell at silencing stage with the dominant and silencing OR depicted in detail. **e**, The 3D positioning of ORs and enhancers in a representative cell at singular OR activation stage with the selected OR depicted in detail. **f**, Number of enhancers within 300 nm of active dominant OR and second-highest expressed OR of the same cell (connected with a line). Statistical significance is labeled. A two-sided Wilcoxon signed-rank test was used. **g**, Number of OR enhancers of the largest enhancer aggregate, second-largest enhancer aggregate and active OR-residing enhancer aggregate of the same cell within 5 particle radii (300 nm) (connected with a line, $n = 18$). A two-sided Wilcoxon signed-rank test for paired data was used, ***$P < 0.01$. **h**, Illustration showing the stepwise OR gene determinations and their coordination with OR enhancers, the accessibility of OR enhancers and OR genes, and the expression of Lhx2 along OSN development is shown below.

## Spatial relationship between active OR genes and enhancers

With the paired 3D genome structure and OR gene expression profiles within the same cell, we explored the spatial relationship between OR enhancers and expressed OR genes to understand how OR gene is activated and selected. The presence of truncated and nonfunctional OR transcripts necessitates the utilization of full-length transcript information, a feature uniquely provided by LiMCA as opposed to HiRES. This capability plays a crucial role in accurately discerning genuine OR gene expression (Extended Data Fig. 9). According to OR gene expression profiles, we classified developing OSNs (progenitor, iOSN and mOSN) into three stages (Fig. 3a, Extended Data Fig. 10a–c and Supplementary Table 4): the multigenic OR activation stage (stage 1)

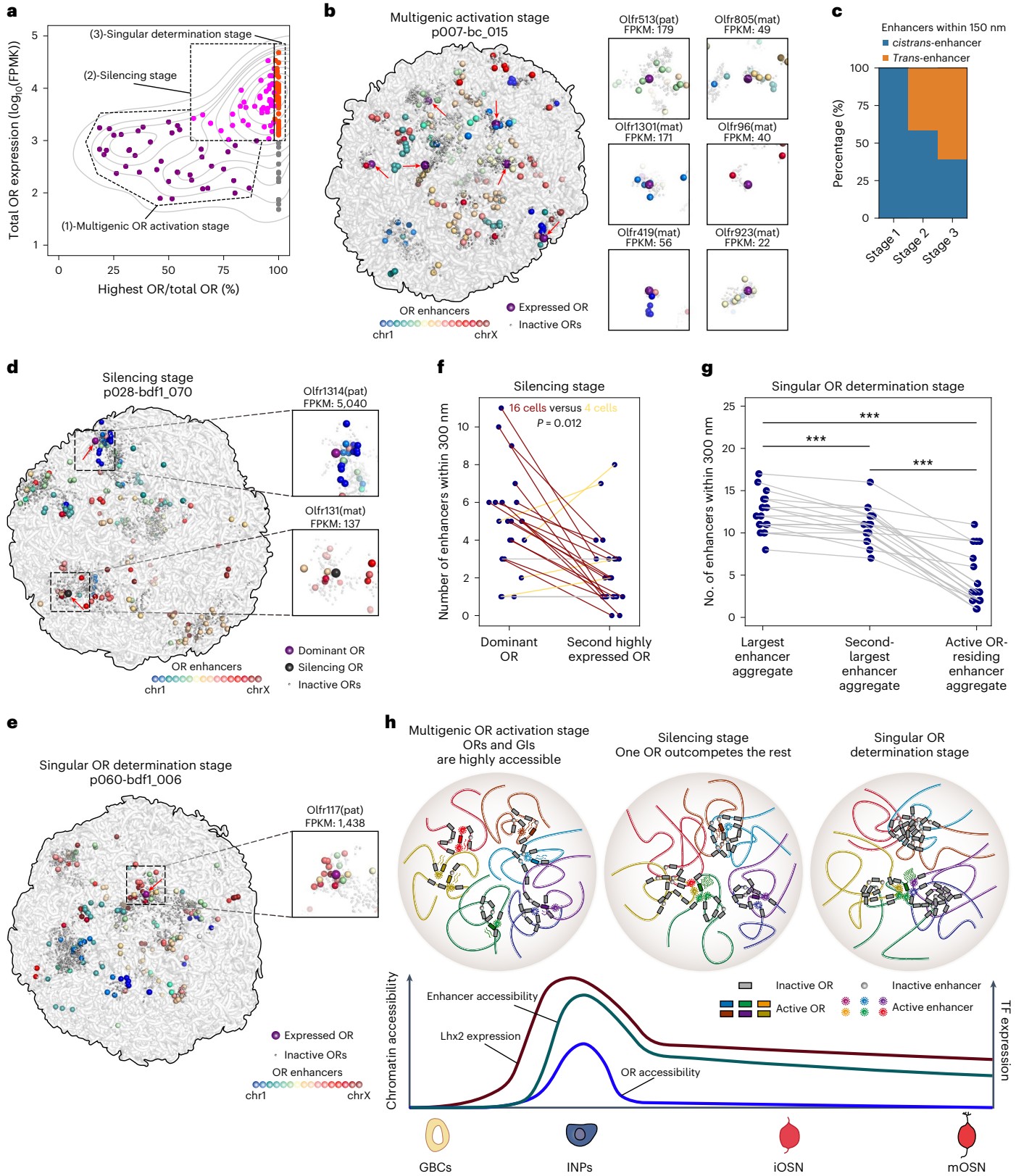

with multiple lowly expressed OR genes and without dominant ones; the silencing stage (stage 2) with one dominant OR gene and several weakly expressed OR genes; and the singular OR determination stage (stage 3) with only one highly expressed OR gene. We hypothesized that these three stages represent the stepwise OR gene expression starting with multigenic OR activation followed by one OR gene outcompeting the others and finally becoming the singularly determined one.

We then investigated the 3D connectivity between expressed OR genes and their enhancers at different stages. For this analysis, we included the newly identified OR enhancers. First, we focused on the progenitor stage where the expression of OR genes begins, which is rarely studied in previous research due to the lack of an available technique. At the activation stage, we observed that most activated OR genes have nearby enhancers (median distance to nearest enhancer was 2.29 radii of particle), which are predominantly *cis*-enhancers from the same chromosome (Fig. 3b,c and Extended Data Fig. 10d). This is consistent with weak inter-chromosomal (*trans*) OR gene interactions at this stage. This supports the importance and sufficiency of *cis*-enhancers for OR gene activation, preceding the establishment of inter-chromosomal OR–OR gene cluster interactions. Furthermore, this concept is supported by previous reports that deletion of specific GIs only downregulates the expression of limited numbers of nearby OR genes belonging to the same OR gene cluster[34–36].

After multigenic OR gene activation, one specific OR gene outcompetes other OR genes and becomes the 'winner' before achieving singular OR gene expression. Nevertheless, how the association with GIs contributes to its dominance remains unclear. When analyzing OSNs undergoing OR gene silencing (stage 2), we observed that the dominant OR gene typically associates with a greater number of proximal enhancers compared to these OR genes undergoing silencing (Fig. 3d,f and Extended Data Fig. 10e). Specifically, within a proximity of 150 nm, 11 cells with the prevailing OR gene associate with more enhancers than silencing ones versus four cells showing the opposite trend; in the case of 300 nm, this is 16 cells versus 4 cells. Moreover, our contact map-based analysis further confirms this finding by illustrating that the dominant OR gene displays more specific and stronger interactions with *trans*-GIs (Extended Data Fig. 10i–k). These results suggest that an increased number of enhancers provide the associated OR gene with more transcriptional sources, thus contributing to its competitive advantage. This suggests a potential positive feedback mechanism between enhanced enhancer connectivity and higher expression levels.

Previous bulk 4C/Hi-C study on fluorescence-activated cell sorting (FACS)-purified OSNs expressing a specific OR gene suggests that the active OR gene interacts frequently with *trans*- and long-range enhancers[17,32]. Single-cell Hi-C on OSNs showed that each OSN harbors multiple enhancer aggregates and proposed that the active OR gene presumably resides in the largest enhancer aggregates according to the bulk observations[18]. To validate whether the finally chosen OR gene is associated with the largest number of enhancers, we inspected OSNs expressing a singular OR gene; however, we found that the ultimately selected OR genes are typically not located in the largest enhancer aggregates (Fig. 3e,g and Extended Data Fig. 10h). This result refutes the previous speculation that the finally determined OR gene is linked to the largest enhancer aggregate.

Through our investigation into the regulation of OR gene expression, we have developed a comprehensive understanding of how OR enhancers are associated with this process (Fig. 3h). During the GBC stage, both OR genes and OR enhancers are inaccessible, resulting in no OR activation. Subsequently, LHX2 and EBF induce the OR enhancers to become highly accessible, which serve as *cis*-enhancers and lead to multigenic OR activation. As this process continues, one OR gene associates with multiple enhancers to become the dominant one, while the rest of the OR genes gradually turn off. Ultimately, only a small set of OR enhancers are retained to support singular OR gene expression.

## Discussion

In this study, we developed a single-cell multi-omics profiling method that enables the efficient and accurate measurement of both 3D genome structure and gene expression. The throughput of this method could be increased with the help of an automated liquid handler or microwell system equipped with liquid-dispensing capabilities in the future. By applying this assay to the developing OSNs and in combination with the single-cell chromatin accessibility and gene expression atlases, we have comprehensively investigated the regulation of OR expression. We have gained an unprecedented understanding of the stepwise process that governs OR gene determination and the dynamic changes in accessibility of OR enhancers at various stages of OR gene expression. Our multi-omics dataset provides valuable insight into the previously unexplored mechanisms before the establishment of the 'one neuron–one receptor' rule.

It remains unclear how OR gene expression occurs during OSN development. At the progenitor stage, multiple ORs are randomly activated, giving rise to two potential scenarios. The first scenario suggests that all but one of the activated OR genes become inactivated. Alternatively, in the second scenario, all initially activated OR genes are silenced, followed by a random reactivation process where one specific OR gene is chosen for final determination. Our hypothesis holds true if the finally selected OR gene is among those activated at the progenitor stage. However, these possibilities cannot be distinguished by current studies. This still needs to be explored in future research to fully understand the mechanism of OR gene determination.

Our finding uncovers that the active OR gene in mOSNs is usually not situated within the largest enhancer aggregate; however, this observation does not conflict with bulk Hi-C observations that demonstrated active OR genes interact most frequently with *trans*- and long-range *cis*-enhancers. The limitation of bulk Hi-C experiments is their inability to capture variability at the single-cell level. It is important to note that the highest contact frequency with GIs does not necessarily require that the active OR gene always interacts with the greatest number of enhancers in individual cells. Instead, the active OR gene interacts with a limited number but different sets of enhancers in individual OSN cells, which also explains the population-based observations.

To further reconcile why the dominant OR gene does not reside within the largest enhancer hub, one plausible explanation lies in the concept of OR 'zone' identity. OR gene selection is biased to predetermined sets of OR genes along the dorsoventral axis of MOE, referred to as 'zones' (refs. [37–39]). A recent study found that dorsal receptors form the strongest interactions across all zones, which are heterochromatic[40]. To investigate whether the ORs residing within the largest enhancer hub display a bias toward dorsal zone identity, we performed an analysis of OR zone identity on stage 2 and stage 3 OSNs harboring a dominant OR gene. Our findings revealed a significant difference in zone identity between the dominant OR gene and the OR genes located within the largest or second-largest enhancer hub (Extended Data Fig. 10l). Indeed, the largest enhancer hub typically encompasses more dorsal OR genes, which indicates that the largest enhancer hub tends to be inactive. This result potentially resolves the puzzle of why the active OR gene is not situated within the largest enhancer hub.

Previous studies proposed that intergenic OR enhancers facilitate specific and strong inter-chromosomal interactions among OR gene clusters across 18 chromosomes[17]. We observed that both the OR enhancers and their associated transcription factor, LHX2, exhibit peak activity during the INP stage. Notably, the inter-chromosomal contacts between OR–OR gene clusters are relatively weak at this stage. This suggests that additional mechanisms, such as the accumulation of repressive histone modifications on OR gene clusters during the maturation of OSNs[15], as well as their interaction with HP1 proteins, may govern the compartmentalization of ORs[16]. Heterochromatic protein-guided phase separation could be the potential driving force for the formation of OR gene heterochromatic aggregate, which plays a central role in B compartment formation[41].

The presence of multiple OR-enhancer aggregates in each OSN suggests that simply being associated with an OR-enhancer hub is insufficient for OR activation[42].

Our findings reveal a noteworthy pattern: OR enhancers undergo reduced accessibility along the course of OSN development, demonstrating that only a subset of enhancers remain active in mature OSNs. Consequently, it can be inferred that only the OR gene interacting with an active multi-chromosomal enhancer hub is expressed, while others remain silenced.

## Online content

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

## Methods

### Animals

The study was approved by the Peking University Institutional Animal Care and Use Committee. All animal experiments were conducted following their guidelines. F1 hybrids of CAST/EiJ (JAX 000928) × C57BL/6J (JAX 000664) and DBA/2J (JAX 000671) × C57BL/5J were used in this study, including LiMCA, METATAC and scRNA-seq experiments; for detailed sampling, please see Fig. 2b. All animals were cultured in specific-pathogen-free conditions and housed in a 12-h light–dark cycle with 40–60% humidity and room temperature (-25 °C).

### Cell types and culture conditions

We performed LiMCA on four human cell lines and the developing mouse MOE.

K562 (ATCC, CCL-243), chronic myeloid leukemia cells, were cultured in Iscove's modified Dulbecco's medium (Gibco, cat. no. 12440053) supplemented with 10% fetal bovine serum (FBS) (Gibco, Thermo Fisher Scientific, cat. no. 10099141) and 1% penicillin/streptomycin (Pen/Strep) (Gibco, Thermo Fisher Scientific, cat. no. 15140148).

GM12878 (Coriell Institute), B lymphoblastoid cells, were grown in Roswell Park Memorial Institute 1640 Medium (Gibco, Thermo Fisher Scientific, cat. no. 11875093) supplemented with 15% FBS and 1% Pen/Strep. GM12878 cells were grown from a single-cell clone.

BJ (ATCC, CRL-2522), fibroblasts, were grown in ATCC-formulated Eagle's minimum essential medium (ATCC, cat. no. 30-2003) with 10% FBS and 1% Pen/Strep.

eHAP (Cellosaurus), an engineered haploid chronic myeloid leukemia cell line[43], was grown in Iscove's modified Dulbecco's medium (Gibco, cat. no. 12440053) supplemented with 10% FBS and 1% Pen/Strep and passaged every 2–3 days at a 1:10 to 1:15 dilution.

When used, adherent cells (for example, K562 and eHAP) were washed twice in 1× PBS and 0.25% Trypsin-EDTA was added (Thermo Fisher Scientific, cat. no. 25200072) and incubated at 37 °C for 5 min, then diluted with complete culture medium to stop trypsinization. Cells were collected by centrifuge at 350$g$ for 5 min and resuspended in 1× PBS. All cell lines were maintained at 37 °C with 5% $CO_2$ at a recommended density.

### Dissociation of single cells from the mouse olfactory epithelium.

The MOE was dissected and minced into small pieces, then dissociated into a single-cell suspension with the Papain Dissociation System (Worthington Biochemical, cat. no. LK003150) at 37 °C for 15 min during incubation, with titration every 5 min with a wide-bore pipette tip according to previously described methods[20]. Then the suspension was filtered with a 30-μm strainer (MACS) and washed twice with ice-cold 1× PBS.

### Single-cell ATAC-seq (METATAC)

Single-cell ATAC-seq datasets were generated with our high-sensitivity METATAC method. METATAC was performed as described in our previous work[31]. We performed METATAC on mouse MOE at four time points during the first postnatal month, day 3, day 7, day 14 and day 28. In brief, dissociated single cells were stained with 7-AAD (eBioscience, cat. no. 00-6993-50), then FACS was used to sort viable cells. The FACS gating strategy is indicated in Extended Data Fig. 6a. Cells were counted and 50,000 cells were taken as input. The nuclei were extracted with 50 μl ATAC lysis buffer (10 mM Tris-HCl, pH 7.5, 10 mM NaCl, 3 mM $MgCl_2$, 0.01% digitonin, 0.1% Tween-20 and 0.1% IGEPAL-CA630) by incubating on ice for 5 min and then they were bulk transposed with META transposome (12.5 μl 2× TD buffer from Illumina Nextera kit, 10 μl 1× PBS (pH 7.4), 0.25 μl 1% digitonin, 0.25 μl 10% Tween and 2 μl 1.25 μM META transposome), then the transposed nuclei were sorted onto 96-well plates. The sorted nuclei were stored at −80 °C or proceeded to amplification.

### Droplet scRNA-seq

A scRNA-seq library was prepared according to the 10x Genomics guidance using the Single-cell Gene Expression 5′ RNA-seq kit v.1.1 (CG000331_ChromiumNextGEMSingleCell5-v2_UserGuide_RevE). In brief, a dissociated single-cell suspension was stained with 7-AAD, then subjected to flow cytometry to sort viable cells. We used about 40,000 cells as input for each reaction. A total of three 10x runs were generated. For the P4–P7 sample, cells from mice at postnatal day 4 and postnatal day 7 were pooled together to load on the same channel. For P14 and P28 samples, cells were from mice at postnatal day 14 and day 28, respectively. One male and one female mouse were used at each time point.

### LiMCA protocol

Our method was based on nucleus–cytoplasm physical separation, such as Trio-seq[21]. The nucleus was submitted to chromosome conformation capture processing and the cytoplasm mRNA was amplified according to the Smart-seq2 procedure[24]. A detailed step-by-step protocol is presented elsewhere[44].

**Single-cell nucleus–cytoplasm separation.** In brief, viable cells were picked into a single tube containing 7 μl soft cell lysis buffer (25 mM Tris, pH 8.3, 30 mM NaCl, 0.45% IGEPAL-CA630 and 1 U μl$^{-1}$ SUPERaseIn), incubated on ice for 30 min, followed by vortexing for 1 min. Samples were centrifuged at 500$g$ for 5 min at 4 °C, then 5 μl supernatant was carefully placed into a new tube. The supernatant was used for Smart-seq2 reverse transcription and amplification[24]. The nuclei were used for chromosome conformation capture.

**Cytoplasm Smart-seq2 procedure.** In brief, 1.25 μl oligo-dT-dNTP mix (1 μM oligo-dT30VN and 2 mM dNTP mix) was incubated at 72 °C for 5 min, then incubated at 4 °C for 5 min. Then, 7 μl reverse transcription mix (1× SSII first-strand buffer, 1 U μl$^{-1}$ RNase inhibitor, 10 U μl$^{-1}$ SSII RTase, 1 mM GTP, 5 mM dithiothreitol, 1 M betaine, 6 mM $MgCl_2$ and 1 μM template switch oligonucleotide) was added, incubated at 42 °C for 90 min, then, ten cycles of 50 °C for 2 min and 42 °C for 2 min, followed by 72 °C for 5 min. After reverse transcription, 14.75 μl amplification mix (14 μl KAPA HiFi Hotstart mix, 0.28 μl 10 μM ISPCR primer and 0.47 μl nuclease-free water) was added to each tube and incubated at 98 °C for 3 min, followed by 21 cycles of 98 °C for 20 s, 65 °C for 30 s and 72 °C for 4 min, followed by 72 °C for 5 min. Samples were purified with 0.7× AMPure XP beads.

**Single-nucleus chromosome conformation capture.** We added 8 μl 2.5% paraformaldehyde (EMS, 15714-S) to the remaining 2-μl pellet (containing the nucleus), vortexed to resuspend, incubated at room temperature for 10 min to fix nuclei, then added 10 μl 0.25 M glycine supplemented with 0.2 μl magnetic beads (Invitrogen, 65011) to quench. Samples were centrifuged at 500$g$ for 5 min at 4 °C and 17 μl supernatant was discarded. We added 17 μl Hi-C lysis (10 mM Tris, pH 8.0, 10 mM NaCl and 0.2% IGEPAL-CA630, supplemented with protease inhibitor) without incubation. Following centrifugation at 500$g$ for 5 min at 4 °C, we discarded 17 μl supernatant, leaving 3 μl. We added 2 μl 0.75% SDS to each tube (final 0.3%), vortexed to resuspend, incubated at 62 °C for 10 min, then added 5 μl 4% Triton X-100 and incubated at 37 °C for 15 min to quench. We added 10 μl digestion mix (2× rCutSmart buffer and 4 U μl$^{-1}$ NlAIII) and incubated it at 37 °C for 2 h with rotation. Following centrifugation at 500$g$ for 5 min at 4 °C, we discarded 17 μl supernatant. We added 17 μl 1× T4 buffer, then following centrifugation at 500$g$ for 5 min at 4 °C, we, again, discarded 17 μl supernatant. Then we added 17 μl ligation mix (1× T4 buffer containing 10 U μl$^{-1}$ T4 ligase) and incubated it at room temperature for 2.5 h with rotation. Following centrifugation at 500$g$ for 5 min at 4 °C, we discarded 18 μl supernatant. We then added 2 μl cell lysis buffer (20 mM Tris, pH 8.0, 40 mM NaCl, 30 mM dithiothreitol, 2 mM EDTA, 0.2% Triton X-100 and 3 mg ml$^{-1}$ QP). Samples were incubated at 50 °C for 1 h, 65 °C for 1 h and 70 °C for 15 min. After lysis, cells were stored at −80 °C.

**Single-cell WGA.** Single cells were amplified with our transposon-based state-of-the-art WGA method, META (Tn5 transposase (Vazyme, S111-01)), as previously described[18,26]. In this study, we use 20 META tags[26].

**Library construction.** The complementary DNA amplicons were quantified, taking 1–5 ng as input for Nextera Tn5 (Vazyme, TD502) tagmentation and library preparation. Cells were pooled for purification and first purified with 0.6× SPRI beads, then purified with 0.2× SPRI beads. The sequenced gDNA and RNA data from the same cells are integrated based on experimental labels.

### Sequencing
METATAC libraries were sequenced with paired-end 2 × 150 bp on Illumina Novaseq, sequenced at 9 Gb per 96-well plate. LiMCA libraries were sequenced with paired-end 2 × 150 bp on Illumina HiSeq x10 or Novaseq, sequenced at 3–6 Gb for gDNA and 0.2–0.6 Gb for cDNA per cell.

### Published data
Phased single-nucleotide polymorphism (SNP) files were downloaded from the Sanger Institute Mouse Genomes Project ('mgp.v5.merged. snps_all.dbSNP142.vcf.gz'). Bulk Hi-C or Micro-C was taken from the 4DN Data Portal (4DNFIXP4QG5B for GM12878, 4DNFIB59T7NN for HFFc6, 4DNFINSKEZND for HAP1, 4DNFII8UHVRO for K562, 4DNFI1T-BYKV3 for GBCs, 4DNFICUQ1N7S for INP, 4DNFIUH9FJR6 for mOSN, 4DNFI5AFARSZ for mOSN (*Olfr1507*) and 4DNFIB5G24G6 for mOSNs (*Olfr17*)). Bulk RNA-seq data of GM12878 were downloaded from ENCODE under accession no. ENCFF897XES.

Lhx2/Ebf ChIP-seq data and bulk ATAC-seq data of mOSNs were taken from the Gene Expression Omnibus under accession no. GSE93570 (ref. 33).

### METATAC analysis
**METATAC data preprocessing.** METATAC data were processed as described previously[31]. In brief, cell barcodes and META sequences were identified for each pair of reads using a custom script. Reads from each cell were split according to their barcodes. Adaptors were then trimmed using cutadapt (v.4.0) with parameters '-e 0.22 -a CTGTCTCTTATACA-CATCT' followed by parameters '-e 0.22 -g AGATGTGTATAAGAGACAG'. Cleaned reads were then mapped to the mm10 (GRCm38) reference genome using bowtie2 (v.2.3.4.3) with parameters '-X 2,000 –local–mm –no-discordant –no-mixed'. Duplicated reads were removed using custom script according to both their mapped location on the genome and META tags. Mapped paired reads were transformed into fragments and a bias of '+4' or '−5' was added to each end of each fragment to center the Tn5 insertion sites.

Fragments from all cells were then integrated. Fragments that may have arisen from contamination were identified and removed by a custom script as described previously[31]. The decontaminated fragments were then subjected to R (v.4.1.0) package ArchR (v.1.0.2) for quality control (QC). TSS enrichment scores and doublet scores of each cell were calculated using the default parameters of ArchR. Cells meeting any of the following conditions were considered to be of low quality and were excluded from downstream analyses: number of aligned reads $<5 \times 10^4$ or $>1 \times 10^6$; ratio of aligned reads <0.85; number of fragments $<3.16 \times 10^3$ or $>3.16 \times 10^5$; ratio of contaminated fragments >0.6; mitochondrial reads >5%; TSS enrichment score <5; promoter fragments <0.1; and doublet score >10.

**METATAC cell embedding and clustering.** Processed METATAC fragments after QC were analyzed using ArchR. First, gene activities were calculated using the addGeneScoreMatrix function with GEN-CODE v.M25 annotation of mm10 genome. Iterative Latent Semantic Indexing was performed with clustering parameters 'resolution = 0.2,

sampleCells = 10,000, n.start = 10'. UMAP embedding was calculated with parameters 'nNeighbors = 30, minDist = 0.5'. Then, cells were clustered (addClusters) with parameters 'maxCluster = 35, resolution = 0.8'. The cell type of each cluster was annotated manually with the help of the Enrichr database (https://doi.org/10.1093/nar/gkw377) according to their marker genes calculated by the getMarkerFeatures function with default parameters. Cell type-specific ATAC peaks were identified using addReproduciblePeakSet function of ArchR with macs2 (v.2.2.7.1). We identified the marker peaks for cell types of interest, including HBCs, GBCs, early/late INPs and immature/mature OSNs, using the getMarkerFeatures function on 'PeakMatrix', and the enriched TF-binding motifs in the corresponding marker peaks were identified using the peakAnnoEnrichment function.

**Integration of METATAC and scRNA-seq profiles.** We used the Seurat CCA-based algorithm to integrate the METATAC and 10x scRNA-seq data of MOE. According to the cell typing of previous single-assay analyses, we used GBCs, early/late INPs, iOSNs and mOSNs from the METATAC dataset, and the same group of cells as those used in scRNA-seq pseudotime analysis from the scRNA-seq dataset. ATAC fragments of these cells and the ArchR peaks associated with these cell types were extracted and analyzed using Signac (v.1.11.0). The gene activities of scRNA-seq variable genes were calculated by the GeneActivity function and normalized. The FindTransferAnchors function was used to perform a canonical correlation analysis and identify the anchors between the two assays. According to the anchors, pseudo-transcriptomes of ATAC cells are imputed and merged with the scRNA-seq dataset. Standard principal-component analysis (PCA) and UMAP embedding of Seurat were performed on the resulting integrated dataset. Similar to the processing of transcriptome dataset, pseudo-time analysis on the co-embedding space was performed by slingshot.

**METATAC pseudotime analysis.** We used the 'addTrajectory' function of ArchR with default parameters to reconstruct the trajectory of MOE development in the UMAP embedding space and assign pseudotime values for GBCs, early/late INPs, iOSNs and mOSNs.

### Identification and validation of candidate OR enhancers
We utilized the Lhx2 and Ebf ChIP-seq data in mOSNs and previously defined GIs from Monahan et al.[17]. We first interrogated in our peak set if there were peaks in OR gene clusters following the criteria of GIs (overlap with both Lhx2 and Ebf ChIP-seq peaks) but not identified as GIs previously. DNA sequences of the resulting 27 peaks and 63 previously identified GIs were extracted from the mm10 genome and subjected to the XSTREME online server (https://meme-suite.org/meme/tools/xstreme) for de novo motif discovery with default parameters. The resulting motif with the most significant E-value corresponded to the desired composite motif. We used fimo (v.5.3.3) with a *q* value threshold of $10^{-3}$ to further identify the location of the motif within GIs and candidate peaks, and determined the *q* values of all matched sites. GIs and identified regulatory peaks were visualized in a Circos plot with ChIP-seq and scATAC-seq (grouped by cell types) tracks of OR clusters using the R packages circlize (v.0.4.12) and ggplot2 (v.3.3.3).

ATAC footprinting was performed by the getFootprints function of ArchR with default parameters on the GIs and candidate peaks centered at the composite motif. Figures were generated by the plot-Footprints function of ArchR with the Tn5 insertion bias normalized by subtraction.

**Comparison of composite motif score.** We kept all results from FIMO scanning (without the *q* value < 0.1 filtering) and compared the distribution of FIMO motif scores of different groups (ATAC peaks outside OR gene clusters, ATAC peaks within OR gene clusters, candidate OR enhancers and GIs) using Kolmogorov–Smirnov tests.

## scRNA-seq data analysis

**10x scRNA data preprocessing.** 10x scRNA-seq reads were processed and mapped to mm10 genome using CellRanger (v.5.0.1). We used the R package Seurat (v4.0.4) for QC and downstream analysis.

**Cell filtering.** Barcodes with UMI counts over 1,000 and fewer than 25,000, number of detected genes over 200 and mitochondrial counts less than 10% were considered as high-quality cells.

**Embedding and clustering.** Cells from three batches (P4/P7, P14 and P28) that passed QC were merged, log-normalized and integrated using the Seurat IntegrateData function with anchors identified by the FindIntegrationAnchors function to correct batch effects. The integrated dataset was scaled and embedded using PCA followed by UMAP, using the ScaleData, RunPCA(npcs = 30), RunUMAP functions of Seurat, respectively. *K*-nearest neighbors of cells were identified using the Seurat FindNeighbors function and Louvain clustering was performed with a resolution of 1.0 using the FindClusters function. Then, cell types were annotated manually according to their marker genes identified by FindAllMarkers with parameters 'min.pct = 0.25, logfc.threshold = 0.25, only.pos = TRUE'. Then, we removed all cells except for GBCs, INPs, iOSNs and mOSNs. The remaining subset was scaled by SCTransform(vst.flavor = 'v2'), followed by RunPCA(npcs = 30), RunUMAP(dims = 1:15), and FindNeighbors and FindClusters(resolution = 0.5). Six out of the identified 17 subclusters, which mainly consisted of mOSNs, could not be aligned well on the trajectory from GBCs to mOSNs and therefore were removed.

**Pseudotime analysis.** We used slingshot (v.2.2.1) to construct a trajectory on the UMAP of the subset after removing the outlier subclusters, and assigned pseudotime values for each cell from GBCs to mOSNs.

**Calculation of correlation between METATAC and scRNA-seq.** To get the correlation between scATAC-seq and scRNA-seq data, we used the variable genes identified by Seurat and calculated the gene score matrix and the log-normalized UMI count matrix from the ATAC and RNA datasets, respectively. We calculated the mean scores or log-normalized counts for each cell type. The Pearson correlation coefficients between each pair of ATAC and RNA clusters over the variable genes were calculated using numpy (v.1.20.3) and visualized by pheatmap (v.1.0.12).

## LiMCA data preprocessing

The RNA data and Hi-C data were preprocessed separately.

**RNA data preprocessing.** For RNA data, we followed the Smart-seq2 processing workflow documented in the Human Cell Atlas Data Portal (https://broadinstitute.github.io/warp/docs/Pipelines/Smart-seq2_Single_Sample_Pipeline/README/). In brief, sequencing reads were mapped to transcriptomic references of hg38 (GRCh38) and mm10 (GRCm38) genome assembly for human and mouse data, respectively, using the hisat2 package. We then used RSEM to quantify the RNA reads and generate a gene count and fragments per kilobase of transcript per million mapped reads (FPKM) matrix.

**Single-cell Hi-C data preprocessing.** Single-cell Hi-C reads were processed as previously described[18]. In brief, contact pairs and contact maps were generated from raw sequencing reads using the hickit pipeline (https://github.com/lh3/hickit). The contact pairs files generated with hickit were then transformed to a Dip-C format for further analysis with Dip-C 'dip-c/scripts/hickit_pairs_to_con.sh' script (https://github.com/tanlongzhi/dip-c). As the human eHAP cell line contains the Philadelphia chromosome (t(9;22)(q34;q11)) and reconstructions of 3D genome structures are sensitive to chromosomal structural variations, for eHAP

Hi-C data, we extracted the exact breakpoints, generated a customized hg38 genome reference accordingly and mapped Hi-C reads to it.

**Haplotype imputation of contacts.** We used the Dip-C method to determine the haplotypes of contacts[18]. In brief, for each read of a contact pair (a leg), we assigned a haplotype if the read segment overlapped with a phased SNP and had a base quality >20. We then performed haplotype imputations of contacts. Specifically, contacts with known haplotypes were used to vote the haplotype of contacts with unknown haplotype; if the majority of voted haplotypes are consistent, then the haplotype of the contact is assigned confidently.

Juicebox (v.1.11.08) or cooltools (v.0.5.1) was used for contact map visualization.

## Criteria for cell exclusion

For human cell line data, cells with <100,000 unique contacts (5 of 389 cells) were excluded. For mouse OSN data, only 3 of 411 cells had <100,000 unique contacts, and so we kept all cells.

## LiMCA RNA data analysis

**Cell embedding and clustering.** The Seurat package (v.4.2.0) was used for QC and downstream analysis of RNA count matrices. We filtered cells with fewer than 200 genes detected and genes expressed in fewer than three cells. The filtered count matrices were then normalized using the NormalizeData (for human data) or SCTransform (for mouse data) function. We then performed PCA and UMAP embeddings and Louvain clustering with the following parameters: RunPCA(dims = 1:20), RunUMAP(dims = 1:15), FindNeighbors(dims = 1:10) and FindClusters(resolution = 0.4) for human data; RunPCA(dims = 1:15), RunUMAP(dims = 1:10), FindNeighbors(dims = 1:10) and FindClusters(resolution = 0.4) for mouse data. For each human cell, we calculated cell-cycle phase scores based on known cell-cycle markers and annotated cell-cycle phase using the CellCycleScoring function. Marker genes were identified using the FindMarkers function.

**Determination of active ORs in single neurons.** For olfactory receptor expression detection, the expression matrix was not filtered with the parameter 'genes expressed in fewer than three cells'. Genuine OR expression was determined by two criteria. First, we filtered out OR genes with an expression level <10 FPKM, as previously described[20]. Second, only OR genes with >90% of the exon region covered were kept, which is necessary to exclude map artifacts and truncated OR transcripts. For OR genes with multiple exons, it was required to detect splicing junctions. This was further confirmed by visual inspection in the Integrative Genomics Viewer (v.2.16.2).

The allele of the expressed OR genes was determined by the phASER package (v.0.9.8) (https://github.com/secastel/phaser).

**Pseudotime analysis.** Monocle3 was used to construct the continuous developmental trajectory from progenitors to mOSN; pseudotime values were assigned to individual cells.

## Analysis of contact maps

**Calculation of scA/B values.** The scA/B values were calculated from the single-cell contact map with the 'dip-c color2' function (with parameters '-b1,000,000 -H -c color/mm10.cpg.1m.txt'). The sex of the mouse MOE cells was confirmed by dissection of adult mice and inferred by analyzing the copy number of sex chromosomes for newborn mice.

**Structural cell typing.** We only retained autosomal bins that were present in all cells. The raw single-cell A/B values were rank-normalized to 0–1 in each cell with the scipy rankdata function. Then the rank-normalized scA/B value matrix was used for PCA and UMAP embedding analysis using the Python sklearn and UMAP

packages with the following parameters: PCA(n_components = 20) and UMAP(n_neighbors = 10).

**Analysis of ensemble contact maps.** Single-cell contact maps were pooled to obtain ensemble maps using the pairtools package (v.0.3.0) (https://github.com/open2c/pairtools). The .hic and .mcool formatted contact matrices were then generated and balanced with iterative correction or Knight–Ruiz, using the cooler (v.0.8.11) (https://github.com/open2c/cooler) and Juicer (v.1.19.02) (https://github.com/aidenlab/juicer/) package. We used the cooltools eigs-cis function (https://github.com/open2c/cooltools) to calculate A/B compartments at a 500-kb resolution and the cooltools insulation function to calculate insulation scores at a 100-kb resolution with a 1-Mb window. Chromatin loops were identified at a 10-kb resolution using the Chromosight (v.1.6.1) package (https://github.com/koszullab/chromosight) with '–min-dist 20,000 –max-dist 20,000,000'. The same analyses as above were performed on a published in situ Hi-C dataset for benchmark analysis. We further generated ensemble maps randomly sampling different numbers of cells ($n$ = 25, 50, 75, 100, 125, 150, 175 and 200) and called chromatin loops. We used the Bedtools (v.2.26.0) pairtopair function to overlap chromatin loops between datasets.

### Virtual 4C analysis
We grouped cells into sets with high (above the median) or low (below the median) expression of specific genes such as *NFKB1* and performed virtual 4C analysis. For each set of cells, we combined contact maps to generate a pseudobulk contact matrix. Contacts with the gene locus were extracted and normalized by the total number of contacts with the locus.

### Single-cell chromatin looping analysis
We first identified cell-type-specific chromatin loops at a 10-kb resolution from published bulk Hi-C or Micro-C maps with the diff_mustache.py script from the mustache package. We then iteratively compared one cell type to others and retained calls unique in that cell type as its cell type-specific chromatin loops. The coolpuppy package (v.0.9.7) was used to generate pileups of merged single-cell maps for each set of loops. We calculated the total contact count for each set of chromatin loops in individual cells using 'dip-c ard' function and combined the counts into a matrix. We noted that the number of chromatin loops was inconsistent across groups due to varying sequencing depths. To eliminate this impact, for each group of loops, we normalized the contact counts by the number of contacts used for loop calling.

### Mouse olfactory cell type annotation
For mouse olfactory data, we annotated cell types in two steps. In the first step, based on RNA counts of known marker genes of mouse olfactory epithelium, we manually annotated the four clusters into OSN progenitors, iOSNs, mOSNs and non-neuronal cells. In the second step, we visualized the above-mentioned cell type assignment on the UMAP plot of scA/B values derived from Hi-C data. Based on structural cell typing, the progenitor cluster in RNA embedding was segregated into two discrete clusters that we named progenitor1 and progenitor2, respectively. These two progenitor clusters did not overlap with each other on the UMAP plot of RNA data, further confirming our assignment.

### 3D genome structure analysis
**Generation of 3D genomes.** Single-cell 3D genome structures were reconstructed on haplotype-imputed contact maps using the hickit package (with parameters -M -i impute.pairs.gz -Sr1m -c1 -r10m -c2 -b4m -b1m -b200k -D5 -b50k -D5 -b20k). Then, the 3dg files were converted to Dip-C format (with scripts/hickit_3dg_to_3dg_rescale_unit.sh). The transformed 3dg files were further cleaned with the 'dip-c clean3' function to remove repetitive regions. For eHAP, no homolog imputation was needed due to it being a haplotype cell line. For K562 and BJ cells,

reconstructions were impractical due to gross chromosomal aberrations or lack of phased SNP information.

**3D genome structure alignment and uncertainty estimation.** For each single cell, three independent replicate structures were generated. Then, the 'dip-c align' function was used to calculate the median and root-mean-square (r.m.s.) of r.m.s.d. of the single-cell 3D genome structures over all 20-kb particles from three independent replicates. This calculation involved two steps; first, the r.m.s.d. was calculated for each 20-kb particle over three replicate pairs (1–2, 1–3 and 2–3), followed by calculating the median or r.m.s. value over all 20-kb particles. Only r.m.s.–r.m.s.d. < 1.5 cells were considered as low uncertainty and kept for downstream 3D genome structure analysis. The Y chromosome was excluded from further analysis due to its short genomic length and low mappability.

**3D genome structure visualization.** The 3dg files were transformed into a PyMol-compatible cif format with the 'dip-c color' function and visualized by PyMol (https://pymol.org/2/).

**Spatial analysis of active genes.** For this analysis, we considered expressed genes with ≥1 FPKM as active genes. The radial positions were calculated using 'dip-c color -C' and normalized by setting the genome-wide median to 1. First, we extracted genomic loci with active genes from 20-kb 3D genome structures and counted the number of active genes of each locus. We then calculated the total number of genes for different radial distances using 'dip-c color -R'.

To characterize active gene clustering for each cell, we extracted the midpoints of all active genes into a .leg file and then generated a .3dg file with the 'dip-c pos' function. We calculated the number of active genes within 3 particle radii from each active gene with 'dip-c color -r 3'. To investigate whether there was a radial preference or prominent clustering of active genes, we used random controls to evaluate background levels.

Specifically, in each single cell, we counted the total number of active genes and randomly sampled the same number of genes from all genes included in the gene annotation files (GENCODE v.M25) regardless of their expression levels. The above analysis was then performed on randomly sampled genes and we compared the distribution between active genes and random controls with a two-sided Mann–Whitney $U$-test. We used the Dip-C name_color_x_y_z_to_cif.sh script to convert the .3dg files of active genes to mmcif-formatted files, which were then used for PyMol visualization. Note that for each active gene, both parental alleles were included in the analysis because most genes express both alleles similarly.

### OR–GI 3D structure analysis
**3D structure visualization.** The 3D position of OR genes and GIs was located from the whole-cell 3D genome structure using the 'dip-c pos' function by providing the corresponding OR genes or GI leg file. Then the OR and GI 3dg files were transformed to cif files for visualization using PyMol. GIs were colored according to chromosomes.

**OR–GI spatial relationship analysis.** Pairwise distances between ORs or between ORs and GIs were calculated using 'dip-c pd'. The 'dip-c network_around.py' script was then used to record GIs or ORs within 2.5 or 5 particle radii from each OR. In each cell, we extracted the number of GIs from ORs that were actively expressed. OR and GI aggregates were identified in single cells as previously described[18]. For analysis of 3D genome structures, we only retained cells in which the allele of the dominant OR (the active OR with the highest expression level) could be determined based on heterozygous SNPs, as OR expression is monoallelic. The second-dominant OR in single cells was defined as the active OR that had the next highest expression level, while requiring determined allelic information.

**OR and GI interaction quantification.** For bulk Hi-C data, the normalized inter-chromosomal contact pileups between ORs and between GIs were generated using the coolpuppy package with '-trans-flanking 10,000,000'. We used the 'dip-c ard' function to calculate single-cell contact pileups between ORs and between GIs with parameters '-d10,000,000 -h100,000'. Contact pileups between active ORs and all 90 GIs were also calculated and aggregated across all cells. Note that for analysis of contact maps, the dominant and second-dominant ORs were defined as the ORs with the highest and second-highest expression levels, respectively. This analysis did not take into account whether the allelic status of the active ORs can be determined by SNPs. We defined the contact strength as the ratio between the mean contact value within 200 kb of OR pairs (or 100 kb of GI pairs) and the mean value in surrounding regions. A value of 1 indicated no contact enrichment.

**Random inactive ORs and permutation of OR expression analysis.** To test the prominence of chromatin interactions between active ORs and inter-chromosomal OR enhancers, we performed two additional analyses. The first one was randomly sampling the same number of inactive ORs from all 1,138 protein-coding ORs for each cell and the second one was permuted gene expressions of cells to mismatch OR expression and 3D genome structure, to investigate their interactions with inter-chromosomal enhancers.

### Reporting summary

Further information on research design is available in the Nature Portfolio Reporting Summary linked to this article.

### Data availability

Raw sequencing data generated in this study have been deposited in the Sequence Read Archive under accession no. PRJNA1002315. The processed data generated during this study have been uploaded to the Gene Expression Omnibus under accession code GSE240128. The cell type of each cluster was annotated manually with the help of the Enrichr database (https://doi.org/10.1093/nar/gkw377). Published MOE Dip-C data were downloaded under Gene Expression Omnibus accession code GSE121791. Published OSN bulk Hi-C data were downloaded from the 4DN database (https://data.4dnucleome.org/). Source data files have been uploaded to figshare (https://doi.org/10.6084/m9.figshare.24547162.v4)[45].

### Code availability

The code used in this study is available at GitHub (https://github.com/tanlongzhi/dip-c, https://github.com/lh3/hickit, https://github.com/zhang-jiankun/LiMCA and https://github.com/sunneyxielab/METATAC_pipeline)[46–49]. All plots were generated with matplotlib (v.3.7.0) and ggplot2 (v.3.3.3).

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

### Acknowledgements

We thank Beijing Berry Genomics for assistance in next-generation sequencing. We thank staff at the Peking University High-throughput Sequencing Center for helping with flow cytometry. This work was financially supported by the Changping Laboratory.

### Author contributions

H.W. and X.S.X. designed the study. H.W., J.P.C. and Y.Z. performed the experiments. J.Z. and F.J. analyzed the data. H.W., J.Z., F.J., L.T. and X.S.X. wrote the manuscript.

### Competing interests

The authors declare no competing interests.

### Additional information

**Extended data** is available for this paper at https://doi.org/10.1038/s41592-024-02239-0.

**Correspondence and requests for materials** should be addressed to Longzhi Tan or X. Sunney Xie.

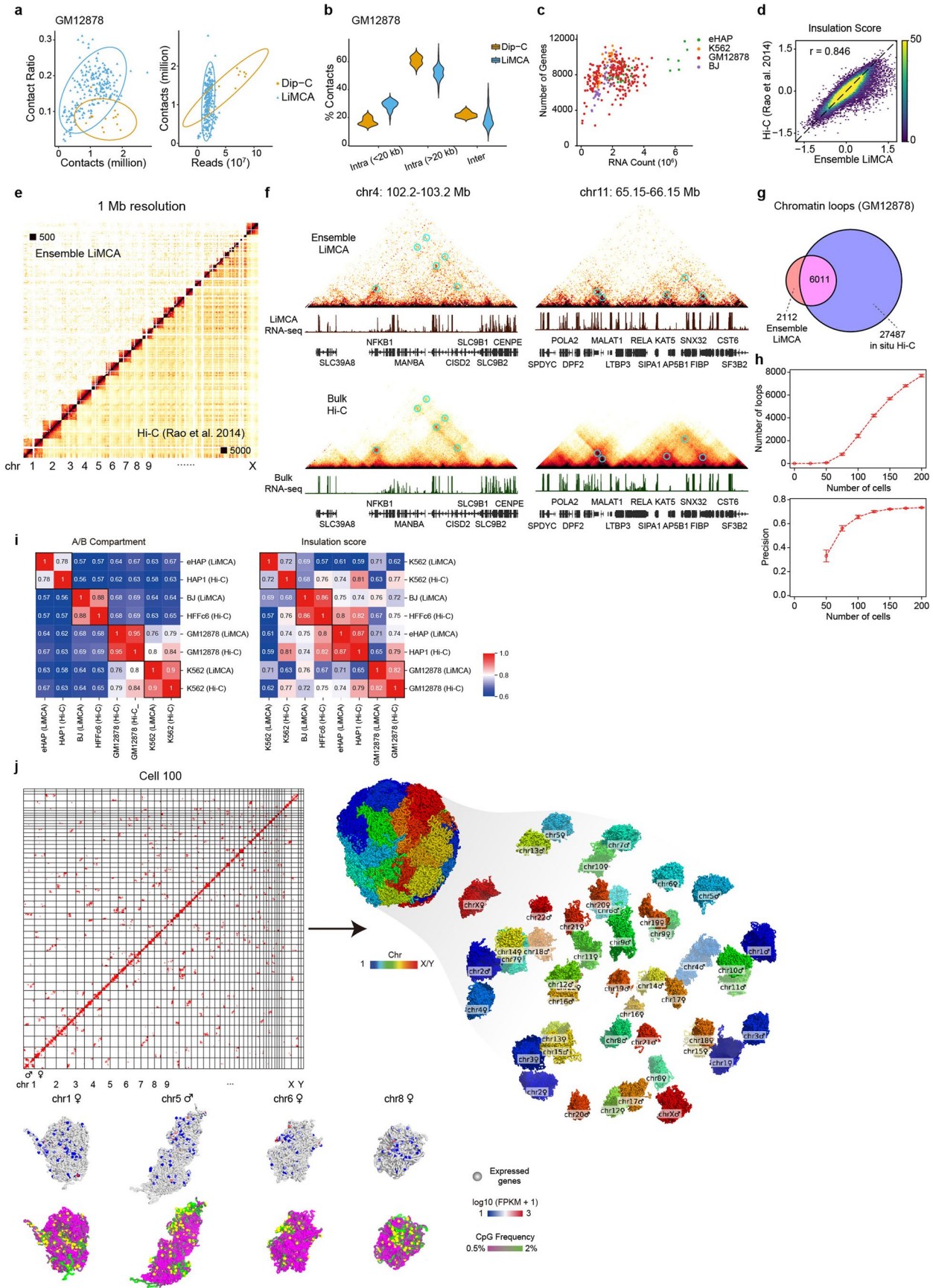

**Extended Data Fig. 1 | See next page for caption.**

**Extended Data Fig. 1 | Validation of LiMCA. a**, Comparison between LiMCA and Dip-C, scatter-plot for contacts number versus contact ratio (left) and reads number versus contact number (right). **b**, Violin plot showing the proportion of cis and trans contacts. **c**, Scatter-plot showing reads number versus detected genes for RNA. **d**, Insulation score of ensemble LiMCA is high concordant to bulk Hi-C, calculated at 50 kb resolution. **e**, Contact maps comparison between ensemble LiMCA and bulk Hi-C at 1 Mb resolution, all chromosomes are shown. **f**, Two selected regions showing ensemble can detect chromatin loops, RNA-seq tracks are shown below. **g**, Venn diagram showing chromatin loops detected by ensemble LiMCA and bulk Hi-C, loops are called with HICCUPS. **h**, Downsample analysis showing the relationship between number of detected chromatin loops (top panel) or precision rate of detected chromatin loops and downsampled cell number (bottom panel). Each cell number were independently sampled 5 times. **i**, Heatmap showing the correlation of A/B compartment score (first eigen value) (left) and insulation score (left) between ensemble LiMCA and in situ Hi-C. **j**, A imputed contact map of a representative GM12878 cells and the reconstructed 3D structure at 20 kb resolution (Top). Four chromosomes with expressed genes projected.

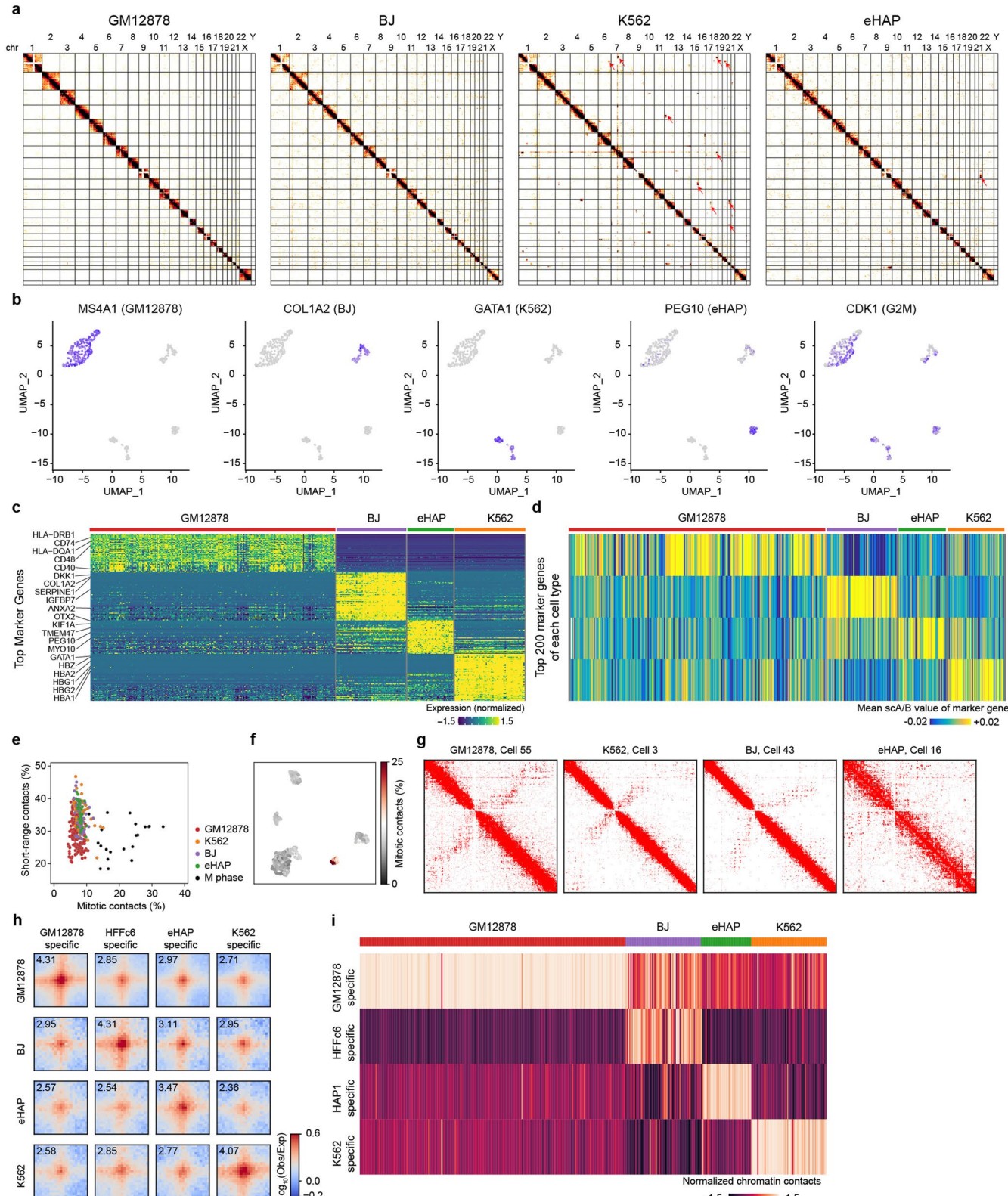

**Extended Data Fig. 2 | LiMCA accurately detects cell-type-specific gene expression and chromatin structures. a**, Ensemble contacts maps of four cell lines at 1 Mb resolution, translocations are highlighted with red arrow. **b**, UMAP showing four represented markers for each cell type and one maker gene of G2M phase. **c**, Expression of top cell-type-specific marker genes for each cell type, top 20 of each cell type are shown. **d**, Mean scA/B value of cell-type-specific marker genes among single cells. For each cell types, the top 200 marker genes

were identified from the paired transcriptome data. **e**, Scatter-plot showing the mitotic contact band (2–12 Mb) ratio versus short-range contacts (< 2 Mb) ratio. **f**, UMAP visualizing the mitotic band ratio of cell line embedding. **g**, Represented contact maps of cells in metaphase cluster. **h**, Pile-up of cell-type-specific chromatin loops using ensemble interaction profiles from each cell type. **i**, Heatmap showing the enrichment of cell-type-specific chromatin loops among single cells.

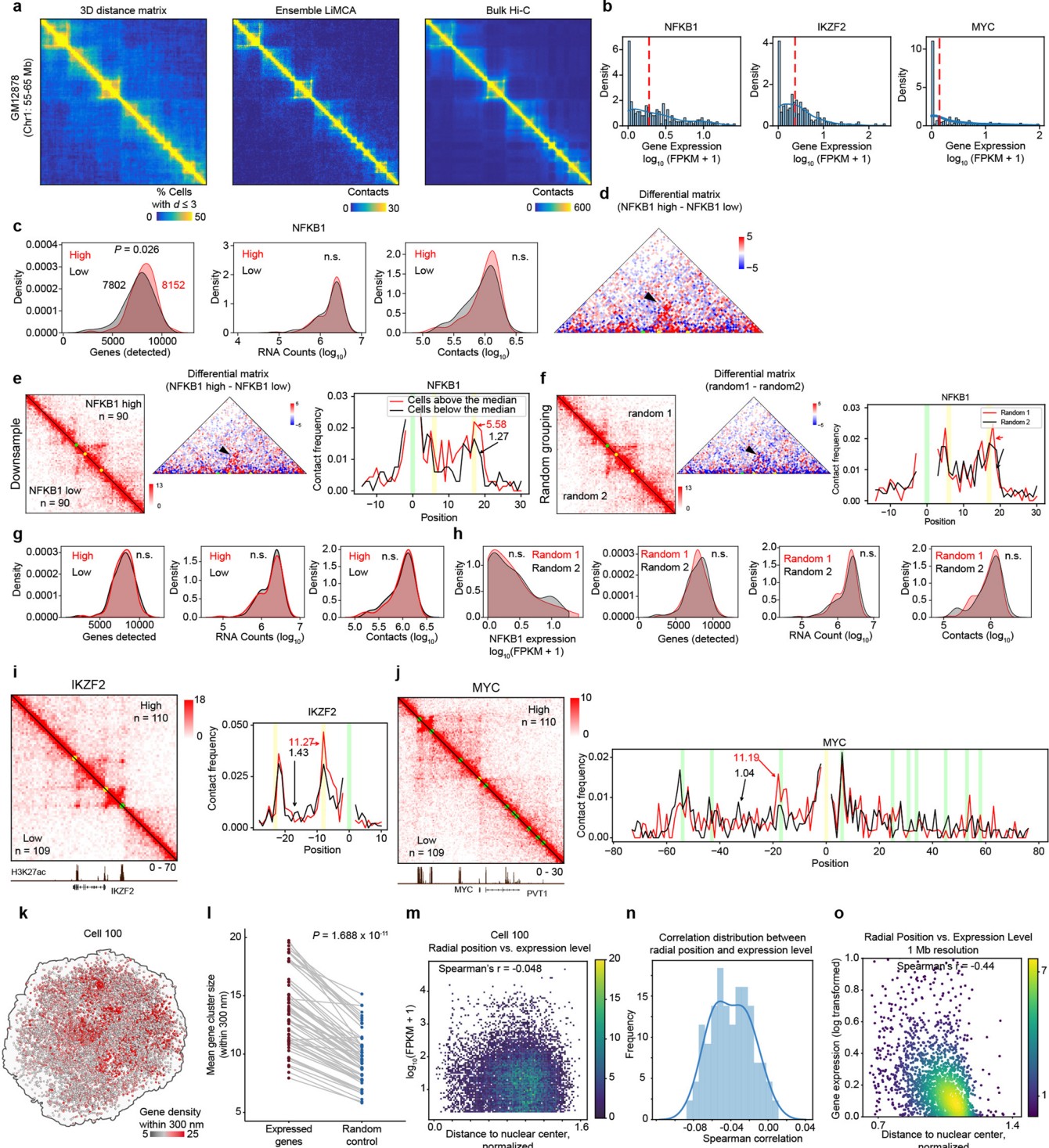

**Extended Data Fig. 3 | See next page for caption.**

**Extended Data Fig. 3 | The relationship between 3D genome organization and expression. a**, Comparison of pairwise 3D distance matrix measured, ensemble LiMCA and bulk Hi-C contact map. **b**, Histogram of expression level distribution for NFKB1 (left), IKZF2 (middle) and MYC (right). **c**, Histogram showing the distribution of detected gene numbers (left), RNA counts (middle) and contact numbers between NFKB-high and NFKB1-low group. n.s., no significance, two-sided Mann-Whitney U rank test. **d**, Differential contact matrix around NFKB1 (chr4: 102.3–102.8 Mb) between NFKB1-high and low groups. **e**, Downsample analysis for NFKB1 gene. Left: contact matrices around NFKB1 (chr4: 102.3–102.8 Mb), representing ensemble Hi-C data from NFKB-high (top left) and NFKB1-low (bottom right). Middle: Differential matrix between NFKB-high and NFKB1-low group. Right: Normalized contact frequency plot, centered at NFKB1 upstream enhancer. **f**, The same as **e**, but cells are randomly grouped. **g-h**, The same as **c**, showing downsampled groups (**g**) and randomly assigned groups (**h**). n.s., no significance, two-sided Mann-Whitney U rank test. **i**, Left:

Ensemble contact maps comparing IKZF2-high and (bottom left) and IKZF2-low (top right). Right: 4 C visualization of interactions between enhancer and the expressed gene, viewpoint centered at downstream enhancer. The green dot/line shows the position of candidate enhancer and the yellow dots/lines represent the position of TSS and TTS. **j**, The same as **i** but for MYC. **k**, Expressed genes spatial distribution of a representative GM12878 cell, genes are colored by gene number within 300 nm. **l**, Dotplot showing the median gene cluster size within 300 nm for expressed genes and randomly selected genes, the same cell was connected with line (n = 68). Two-sided Wilcoxon signed-rank test for paired data. **m**, Scatter-plot of expression level versus the normalized radial position at 20 kb resolution of a representative GM12878 cells. **n**, Correlation distribution of normalized radial position versus expression level among single cells. **o**, The same as **i**, but for mean expression level versus median radial position across cell population at 1 Mb resolution.

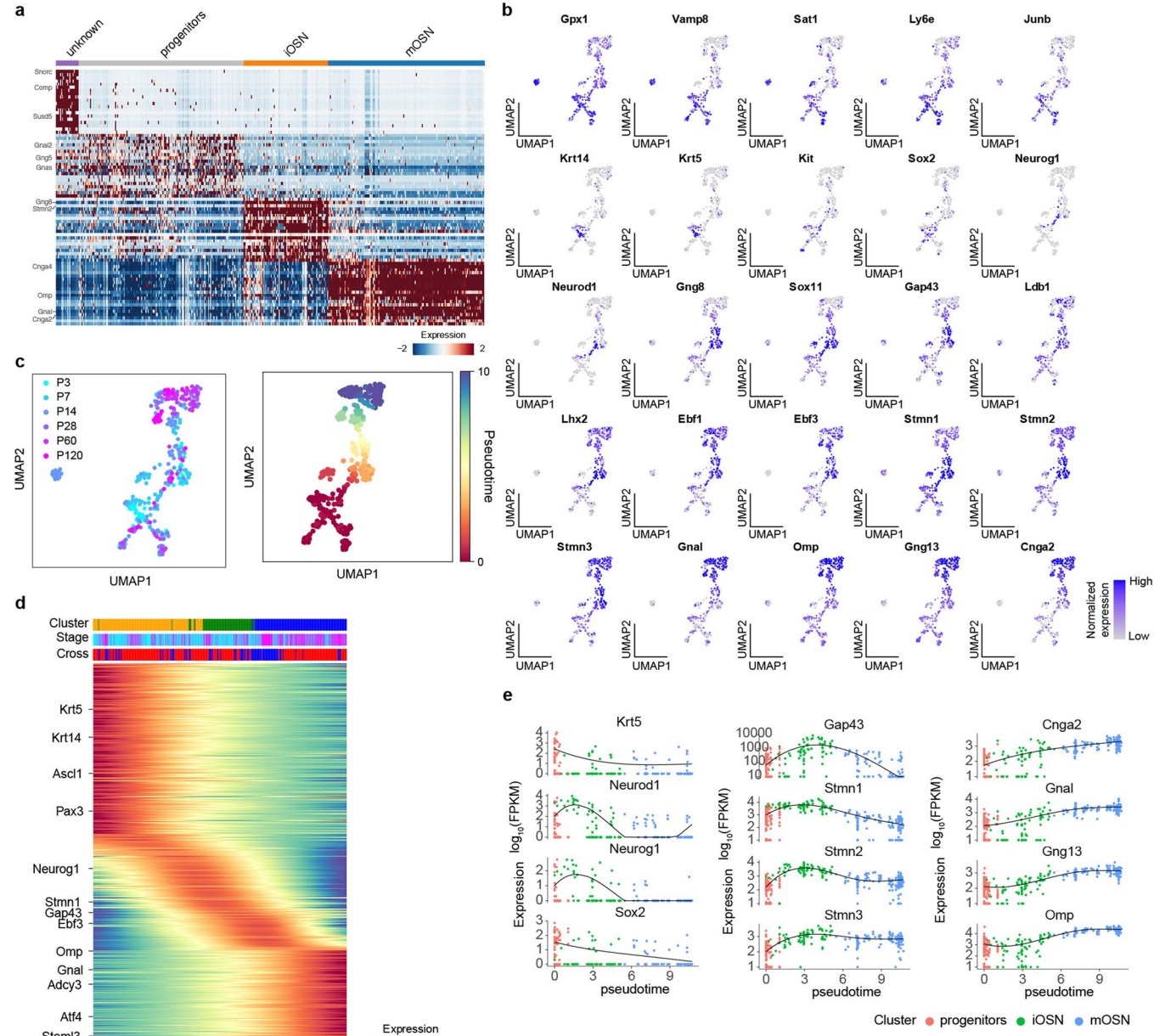

**Extended Data Fig. 4 | Gene expression of the single-cell LiMCA multi-omics atlas recapitulate the continuous OSN genesis and marker gene expression. a**, Expression of top cell-type-specific marker genes. The top 20 maker genes are plotted. **b**, UMAP projection of known maker genes for OSN progenitors, iOSNs and mOSNs. **c**, The same as Fig. 2c (left), colored by mouse age and pseudotime. **d**, Heatmap showing the continuous gene expression change along OSNs genesis using pseudotime. **e**, Scatter-plot showing the dynamic gene expression of known maker genes.

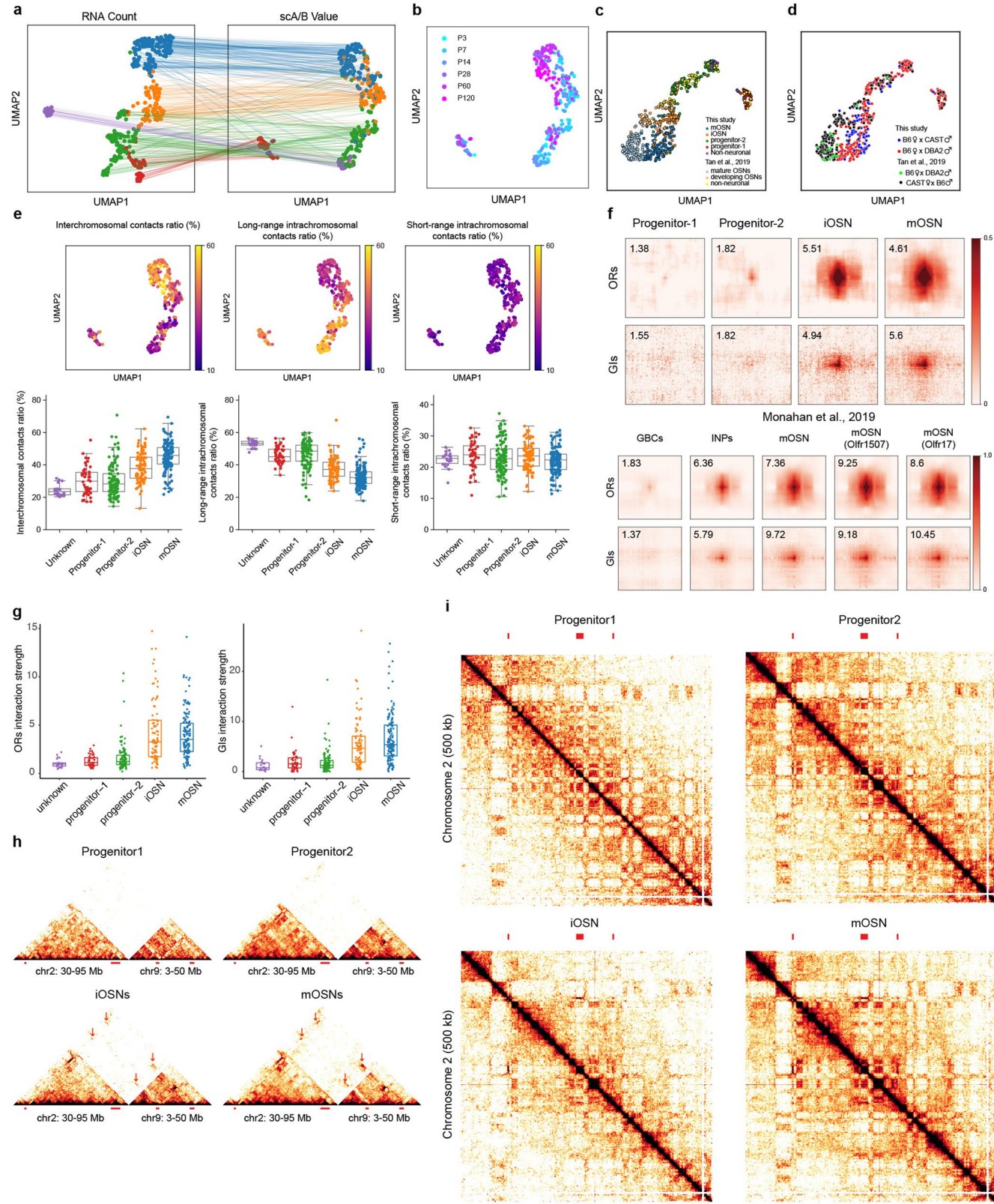

**Extended Data Fig. 5 | See next page for caption.**

**Extended Data Fig. 5 | LiMCA recaptures the characteristic 3D genome reorganization along OSNs development. a**, The same as Fig. 2c, The same cell was connected in the two embeddings. **b**, The same as Fig. 2d (right), cells were colored by mouse age. **c**, UMAP showing the embedding of integrated MOE cells of this study and published MOE data (Tan et al.[18], *Nat. Struct. Mol. Biol.* 2019), cells are colored by cell label in each dataset. **d**, The same as **c**, cells are labeled by different mouse crosses. **e**, UMAP visualization of inter-chromosomal contact ratio, long-range (> 20 kb) intra-chromosomal ratio and short-range (< 20 kb) intra-chromosomal ratio (top), and the boxplot quantification of these values, the black horizontal line and the box represent the median and quartiles, respectively (bottom). The whiskers indicates minima and maxima. (non-

neuronal, n = 22; progenitor1, n = 47; progenitor2, n = 111; iOSN, n = 85; mOSN, n = 146). **f**, Pile-up of interactions between ORs and GIs of ensemble interaction profiles from each cluster (top), and bulk Hi-C datasets from Monahan et al.[17], (2019) (bottom). **g**, Boxplot showing the gradual increasing of OR-OR, GI-GI interaction strength along OSN development. The box horizontal line and the box represent the median and quartiles, respectively. (non-neuronal, n = 22; progenitor1, n = 47; progenitor2, n = 111; iOSN, n = 85; mOSN, n = 146). **h**, Contact maps of ensemble interaction profiles of each cell cluster at regions of chr2: 30–95 Mb and chr9: 3–50 Mb, OR gene clusters are indicated. **i**, Contact maps of ensemble interaction profiles of each cell cluster of chromosome 2 (left).

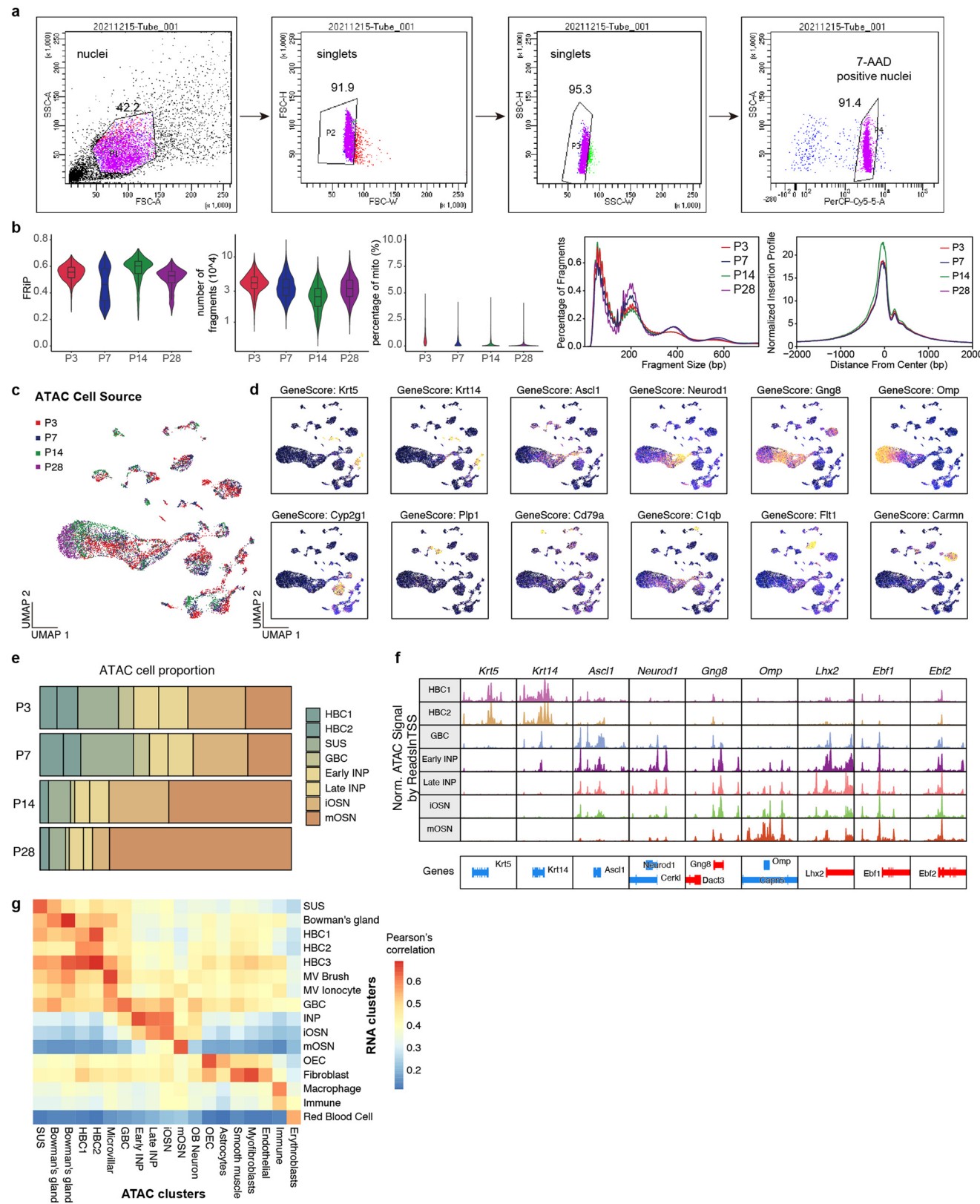

**Extended Data Fig. 6 | Quality control and overview of MOE METATAC results.**
**a**, FACS gating strategy to sort single nuclei for METATAC experiment. **b**, Quality control metrics for METATAC dataset, from left to right are Distribution of ratio of decontaminated fragments in peaks (FRiP), number of decontaminated fragments, percentage of mitochondrial fragments, fragment sizes distribution, and TSS enrichment for METATAC cells of four batches, respectively. Numbers of cells are 4090, 3146, 2850, 1794, for P3, P4, P14, and P28, respectively. The box horizontal line and the box represent the median and quartiles, respectively, and the whiskers extends 1.5*interquartile range. **c-d**, UMAP of MOE METATAC cells colored by (**c**) cell source batches, and (**d**) gene scores of marker genes. **e**, Proportion of cell types associated with MOE development for cells from P3, P7, P14, and P28 mice. **f**, Tracks of METATAC signals normalized by number of reads in TSS near the promoter of marker genes of HBCs, GBCs, INPs, and OSNs. **g**, Heatmap shows the correlation coefficients between cell clusters of MOE 10x scRNA-seq and METATAC.

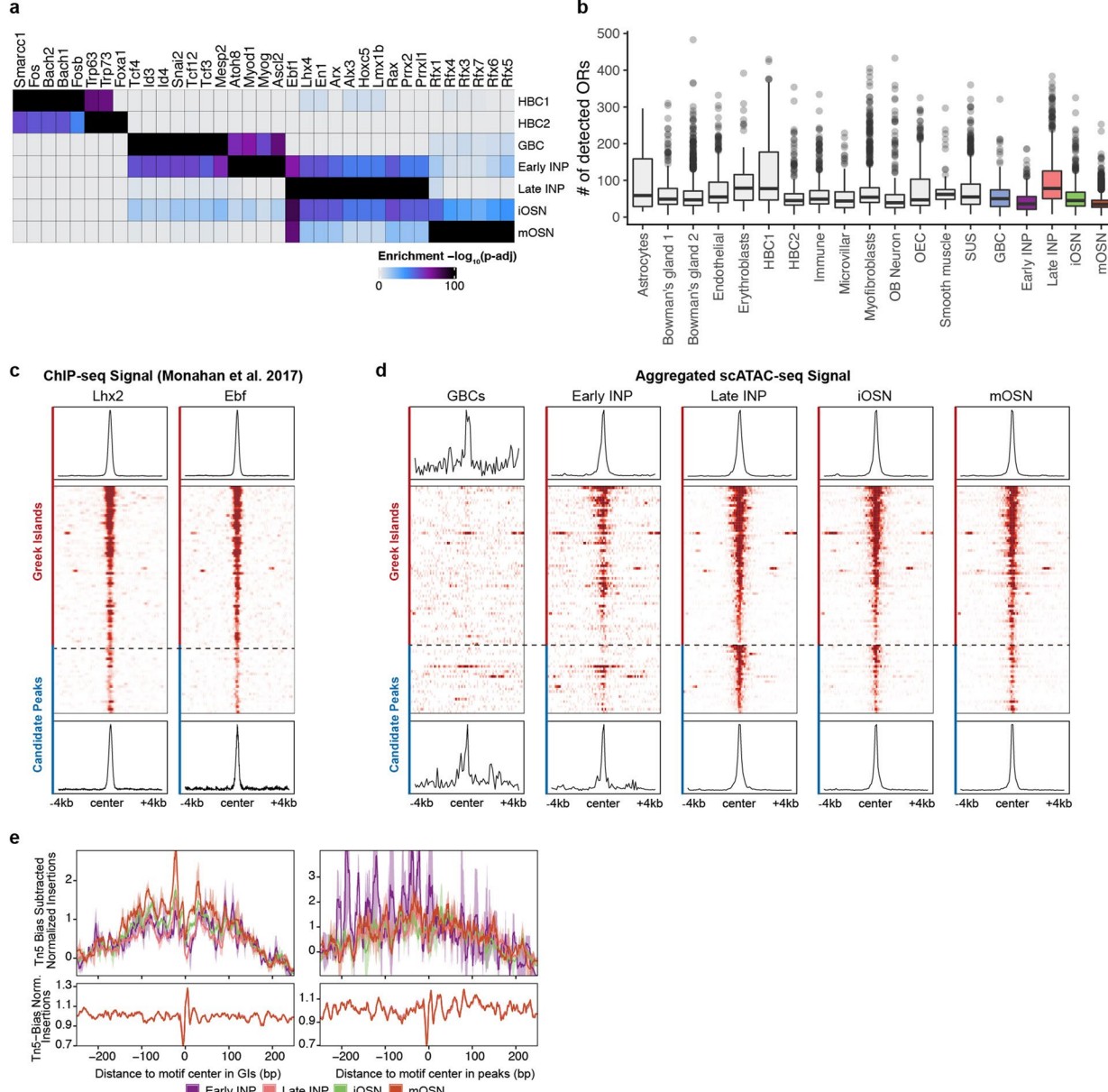

**Extended Data Fig. 7 | METATAC identifies new candidate OR enhancers.**
**a**, TF-binding motifs enriched in HBCs, GBCs, INPs, and OSNs. **b**, Numbers of detected OR genes in various cell types are shown as a boxplot (n = 11,880, for each cluster information is store at source data). An OR gene is considered to be detected if it has a gene score > 0 as calculated by ArchR. The box horizontal line and the box represent the median and quartiles, respectively, and the whiskers extends 1.5*interquartile range. **c-d**, Lhx2 and Ebf ChIP-seq signals (**c**) and METATAC signals of different MOE cell development stages (**d**) at previously defined 63 Greek Islands and 27 candidate regulatory peaks identified in this study. Aggregated signal of all GIs or peaks are shown above or below the heatmap, respectively. **e**, METATAC footprints at composite motif sites in GIs or identified peaks. Aggregated normalized METATAC insertions and the Tn5 bias corrected are shown.

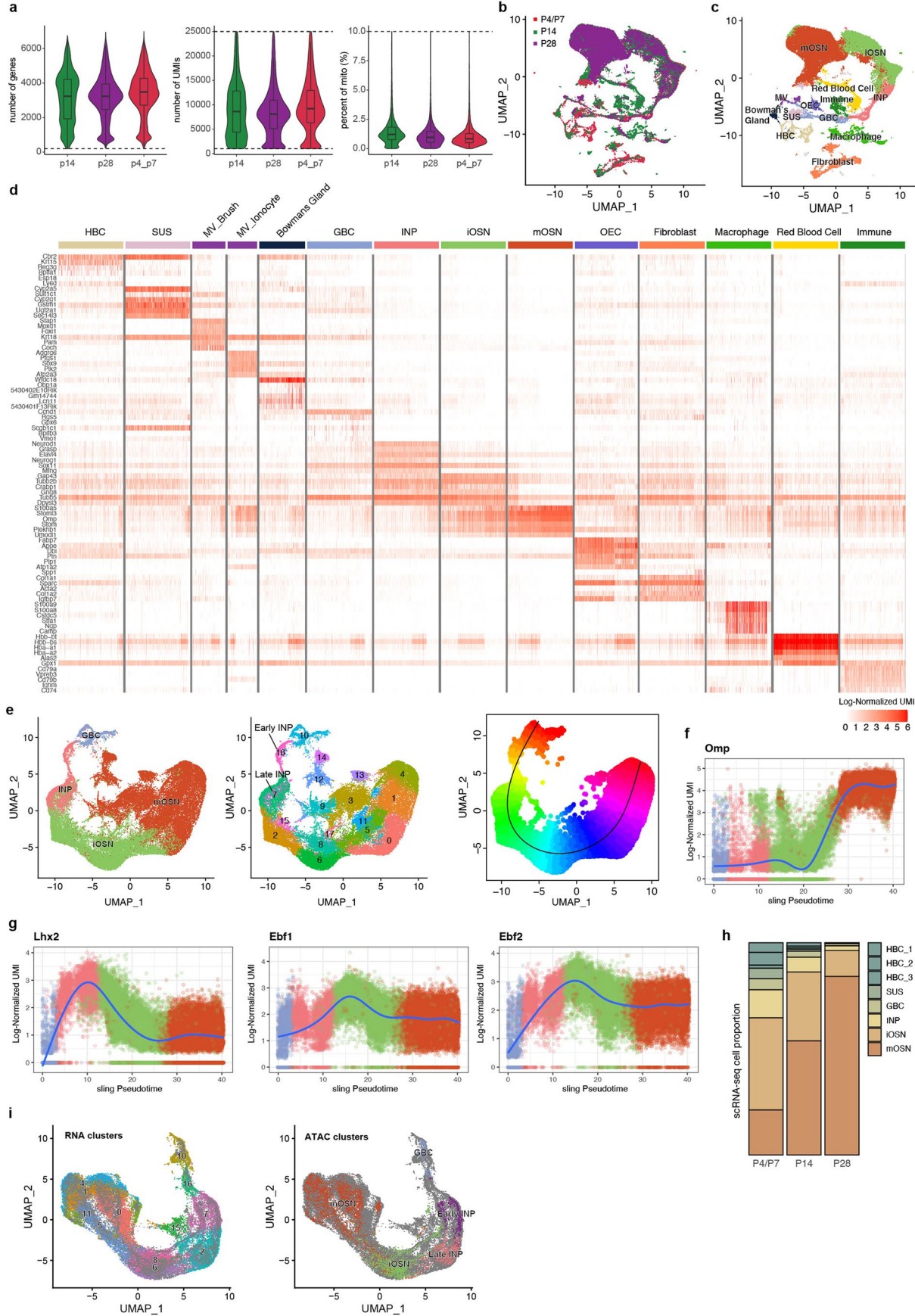

**Extended Data Fig. 8 | See next page for caption.**

**Extended Data Fig. 8 | Quality control and overview of MOE scRNA-seq results. a**, Number of detected genes, UMI counts, and percentage of mitochondrial UMIs for cells from P4/P7 (N = 28,303), P14 (N = 24,116), and P28 (N = 21,158) mice. **b-c**, UMAP of scRNA-seq data colored by origin batch (b) and cell types (c). The box horizontal line and the box represent the median and quartiles, respectively, and the whiles extends 1.5*interquartile range. **d**, Heatmap shows the expression of marker genes of each cell type. For cell types with more than 1,000 cells, top 1,000 representative cells with the highest UMI counts are shown. **e**, UMAP of the subset of scRNA-seq dataset including GBCs, INPs, iOSNs, and mOSNs, colored by previously identified cell types (left), new clusters (middle), and pseudotime (right). **f-g**, Dynamics of the expression of Omp (**f**), and Lhx2, Ebf1, Ebf2 (**g**) during the development of MOE. **h**, Proportion of cell types associated with MOE development for cells from P4/P7, P14, and P28 mice in the scRNA-seq dataset. **i**, UMAP of the co-embedded METATAC and scRNA-seq data from GBC to mOSN, colored by RNA clusters (left) and METATAC clusters (right).

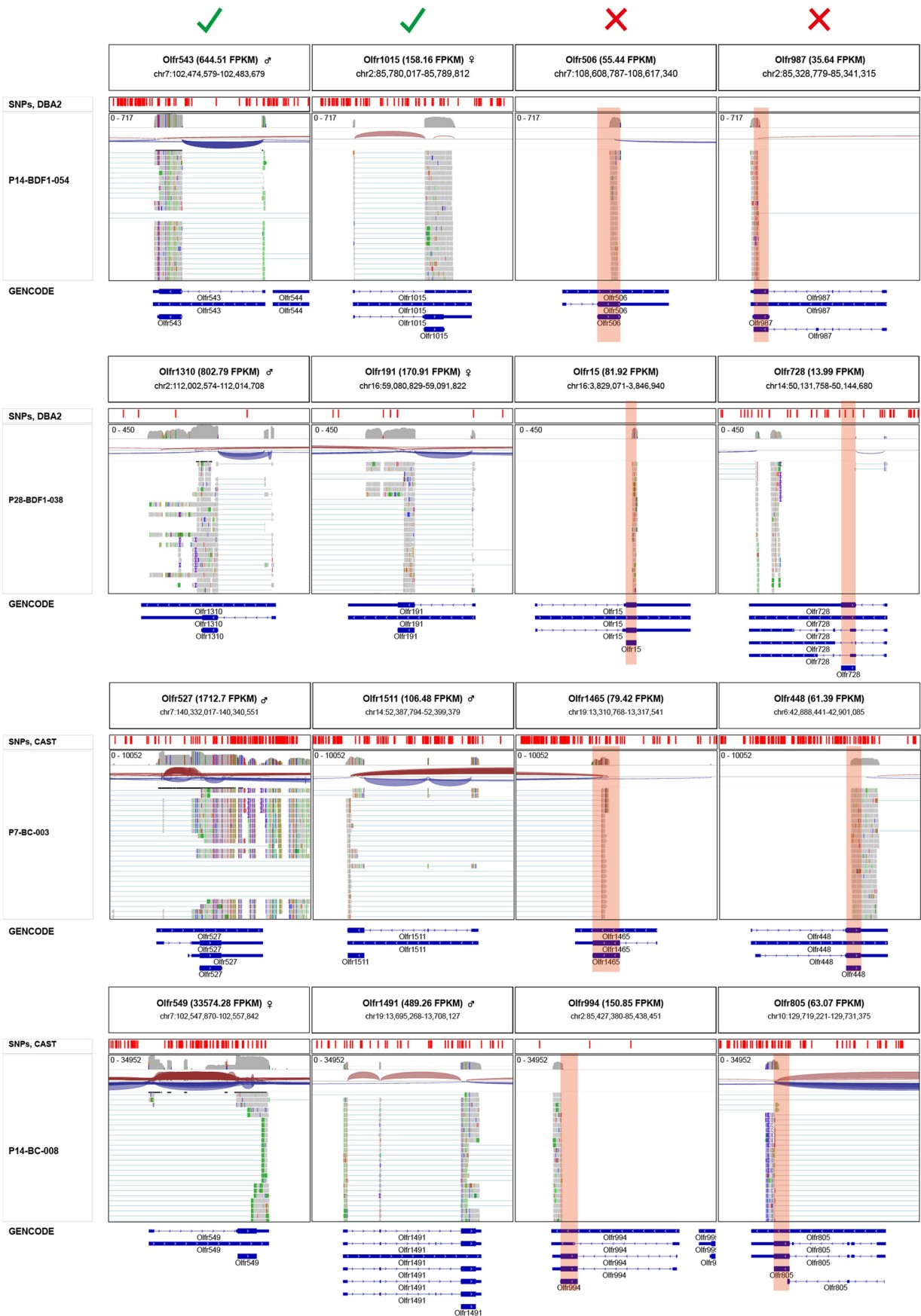

**Extended Data Fig. 9 | See next page for caption.**

**Extended Data Fig. 9 | Identification of genuine OR expression with LiMCA full-length transcript information.** Genome browser views of mRNA read coverage profiles for different OR genes in example single cells. Read bars in the top panel indicates heterozygous SNP sites of each cross. mRNA coverage, junctions, and reads are shown below. Gene annotations are taken from GENCODE. Coverage is set to logarithmic scale. The first two columns show genuine OR expression and the last two columns show false OR expression. Red shades highlight the incomplete read covrages. The identified OR-expressing alleles are represented by ♀ (maternal) or ♂ (paternal), respectively.

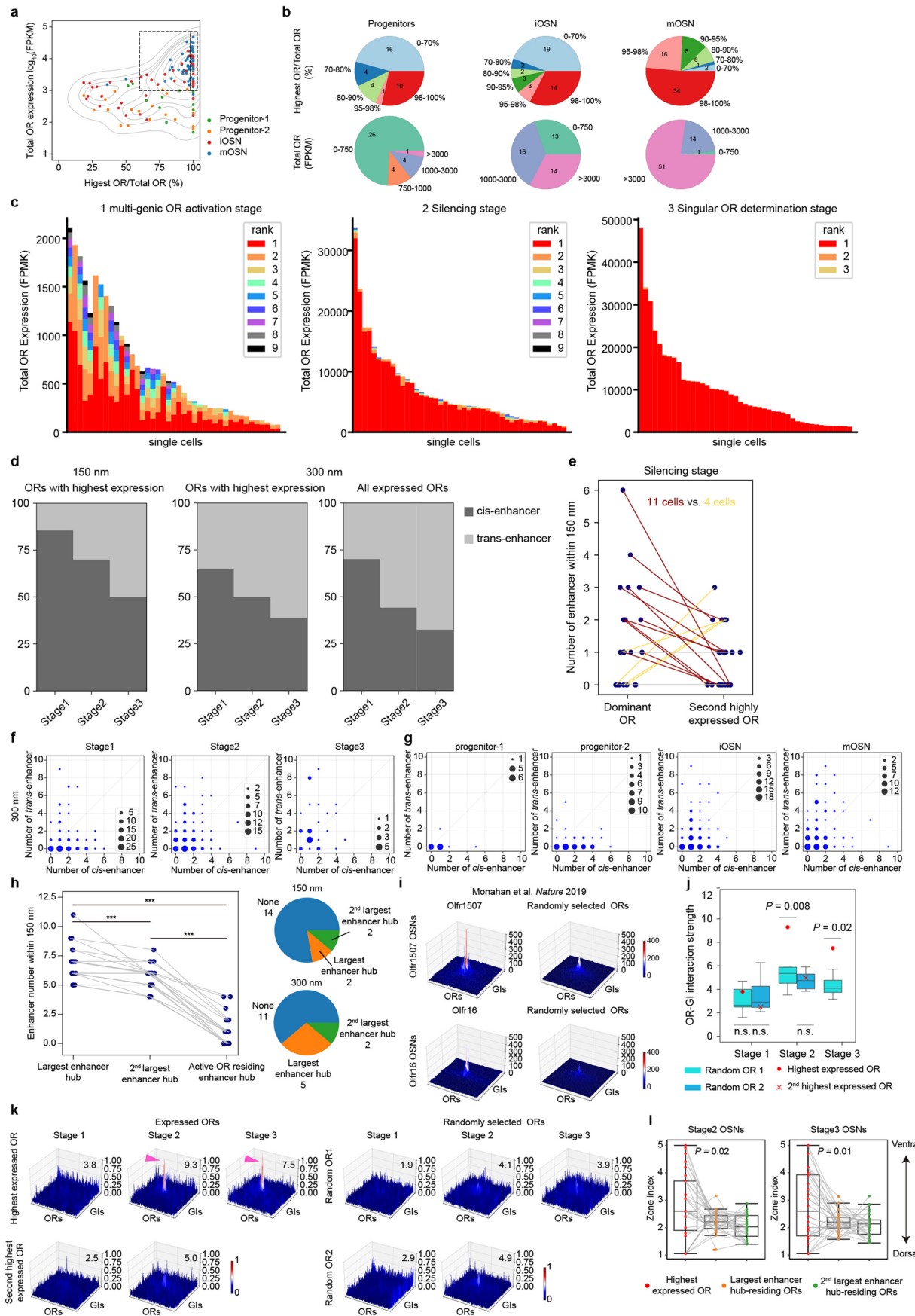

**Extended Data Fig. 10 | See next page for caption.**

**Extended Data Fig. 10 | The spatial relationship between expressed OR genes and their enhancers. a**, The same as Fig. 3a, cells are colored according to cell identity. **b**, Pie chats summarize the OR gene expression information. **c**, OR FPKM values within single cells of each stage. Each color represents an individual OR, except black, which represents all OR ranked below 8. **d**, Composition of *cis* or *trans* GIs within 150 nm or 300 nm from expressed ORs (considering all expressed ORs). **e**, Same as Fig. 3f but for the number of GIs within 2.5 particle radii (150 nm) from each expressed OR in single cells. **f**, Summary of the number of nearby *cis*- and *trans*-GIs for each OR in different stages. **g**, Same as **f** but grouped by different cell types. **h**, Left: Same as Fig. 3g but for within 2.5 particle radii (150 nm). Two-sided Wilcoxon signed-rank test for paired sample was used, *** indicates $P < 0.01$. Right: pie chart depicting the number of cells with active OR residing the largest GI aggregate, the second largest GI aggregate, and others. **i**, 3D surface plot showing the normalized interaction strength between active

OR or inactive ORs (100 randomly selected OR genes) and inter-chromosomal OR enhancers for bulk Hi-C data on OSNs expressing *Olfr1507* (left) and *Olfr16* (right). Data from ref. 17. **j**, Boxplot showing the quantification of OR-enhancer interactions for the highest expressed ORs and second highest expressed ORs (stage1 and stage 2 OSNs) and randomly selected inactive ORs, for the random OR control, 10 independent sampling was performed. P values are from two-sided one-sample t-tests. **k**, 3D surface plot showing the normalized interaction strength between active ORs and inter-chromosomal OR enhancers (left) or between randomly selected inactive ORs and inter-chromosomal OR enhancers (right) of LiMCA data. **l**, Boxplot showing the zone index of the highest expressed ORs and ORs residing within the largest or second largest enhancer hub, only stage 2 and stage 3 OSNs harboring a dominant OR were analyzed, the same cell was connected with a line. P values are from two-sided Wilcoxon signed-rank tests.

# Reporting Summary

## Statistics

For all statistical analyses, confirm that the following items are present in the figure legend, table legend, main text, or Methods section.

| n/a | Confirmed | |
|---|---|---|
| ☐ | ☒ | The exact sample size (*n*) for each experimental group/condition, given as a discrete number and unit of measurement |
| ☐ | ☒ | A statement on whether measurements were taken from distinct samples or whether the same sample was measured repeatedly |
| ☐ | ☒ | The statistical test(s) used AND whether they are one- or two-sided<br>*Only common tests should be described solely by name; describe more complex techniques in the Methods section.* |
| ☒ | ☐ | A description of all covariates tested |
| ☒ | ☐ | A description of any assumptions or corrections, such as tests of normality and adjustment for multiple comparisons |
| ☐ | ☒ | A full description of the statistical parameters including central tendency (e.g. means) or other basic estimates (e.g. regression coefficient) AND variation (e.g. standard deviation) or associated estimates of uncertainty (e.g. confidence intervals) |
| ☐ | ☒ | For null hypothesis testing, the test statistic (e.g. *F*, *t*, *r*) with confidence intervals, effect sizes, degrees of freedom and *P* value noted<br>*Give P values as exact values whenever suitable.* |
| ☒ | ☐ | For Bayesian analysis, information on the choice of priors and Markov chain Monte Carlo settings |
| ☒ | ☐ | For hierarchical and complex designs, identification of the appropriate level for tests and full reporting of outcomes |
| ☐ | ☒ | Estimates of effect sizes (e.g. Cohen's *d*, Pearson's *r*), indicating how they were calculated |

*Our web collection on statistics for biologists contains articles on many of the points above.*

## Software and code

Policy information about availability of computer code

| Data collection | No software was used. |
|---|---|
| Data analysis | 10X scRNA-seq data were analyzed with Cell Ranger (v 5.0.1) and Seurat (v 4.0.4).<br>METATAC data were processed with custom scripts (https://github.com/sunneyxielab/METATAC_pipeline) and further analyzed with ArchR (v1.0.2).<br>LiMCA RNA data were analyzed with Seurat (v4.2.0).<br>LiMCA Hi-C data were processed and analyzed with dip-c and hickit (r291) package (https://github.com/tanlongzhi/dip-c, https://github.com/lh3/hickit).<br>3D genome structures were visualized with PyMol (v2.4.0). All plots were generated with matplotlib (v3.7.0) and ggplot2 (v3.3.3).<br>Custom code related to this paper is available at https://github.com/zhang-jiankun/LiMCA.<br>Flow cytometry was analyzed with BD FACSDiva v9.0 Software.<br>Circos plot was generated with R package circlize (v0.4.12). |

For manuscripts utilizing custom algorithms or software that are central to the research but not yet described in published literature, software must be made available to editors and reviewers. We strongly encourage code deposition in a community repository (e.g. GitHub). See the Nature Portfolio guidelines for submitting code & software for further information.

## Data

Policy information about availability of data

All manuscripts must include a data availability statement. This statement should provide the following information, where applicable:

- Accession codes, unique identifiers, or web links for publicly available datasets
- A description of any restrictions on data availability
- For clinical datasets or third party data, please ensure that the statement adheres to our policy

Raw sequencing data generated in this study has been deposited to the Sequence Read Archive (SRA; https://www.ncbi.nlm.nih.gov/sra) under accession number PRJNA1002315. The processed data generated during this study has been uploaded to the Gene Expression Omnibus under accession number GSE240128.
Cell type of each cluster was annotated manually with the help of Enrichr database (https://doi.org/10.1093/nar/gkw377).
Published MOE Dip-C data was downloaded under GEO accession code GSE121791. Published OSN bulk Hi-C data was downloaded from 4DN database (https://data.4dnucleome.org/).

## Human research participants

Policy information about studies involving human research participants and Sex and Gender in Research.

| | |
|---|---|
| Reporting on sex and gender | Not applicable. There is no human participants involved in this study. |
| Population characteristics | Not applicable. |
| Recruitment | Not applicable. |
| Ethics oversight | Not applicable. |

Note that full information on the approval of the study protocol must also be provided in the manuscript.

# Field-specific reporting

Please select the one below that is the best fit for your research. If you are not sure, read the appropriate sections before making your selection.

☒ Life sciences    ☐ Behavioural & social sciences    ☐ Ecological, evolutionary & environmental sciences

For a reference copy of the document with all sections, see nature.com/documents/nr-reporting-summary-flat.pdf

# Life sciences study design

All studies must disclose on these points even when the disclosure is negative.

| | |
|---|---|
| Sample size | Sample size was not predetermined. For single-cell joint chromatin architecture and gene expression multi-omics data, it consisting of four cell lines, which includes 220 GM12878 cells, 63 K562 cells, 42 eHAP cells and 63 BJ cells, and 411 cells dissociated from the mouse main olfactory epithelium. For single-cell chromatin accessibility data, we profiled 11,880 single cells from the mouse main olfactory epithelium during the first postnatal month. For single-cell RNA-seq data, we collected 73,577 single cells from the mouse main olfactory epithelium during the first postnatal month. |
| Data exclusions | For the statistic analysis of spatial relationship between expressed ORs and their enhancers in Figure3, several cells were excluded due the unknown allele of expressed ORs of bad quality of 3D structures, as listed in supplementary table 4. |
| Replication | For single-cell ATAC-seq data, each mouse age was generated with two independent sampling replicates.  All attempts at replication were successful. |
| Randomization | Randomization was not required since our study is based on sequencing. For different group analysis, cells were allocated according to expression level OR the stage of olfactory receptor expression. Random grouping control was down in these analysis to confirm the conclusion. |
| Blinding | Blinding was not required since our sample is taken from wild-type mice or normal cultured cell line. Since the mice was not genetically engineered and cell line was taken from normal culture with perturbation. |

# Reporting for specific materials, systems and methods

We require information from authors about some types of materials, experimental systems and methods used in many studies. Here, indicate whether each material, system or method listed is relevant to your study. If you are not sure if a list item applies to your research, read the appropriate section before selecting a response.

## Materials & experimental systems

| n/a | Involved in the study |
|---|---|
| ☒ | ☐ Antibodies |
| ☐ | ☒ Eukaryotic cell lines |
| ☒ | ☐ Palaeontology and archaeology |
| ☐ | ☒ Animals and other organisms |
| ☒ | ☐ Clinical data |
| ☒ | ☐ Dual use research of concern |

## Methods

| n/a | Involved in the study |
|---|---|
| ☒ | ☐ ChIP-seq |
| ☐ | ☒ Flow cytometry |
| ☒ | ☐ MRI-based neuroimaging |

# Eukaryotic cell lines

Policy information about cell lines and Sex and Gender in Research

| | |
|---|---|
| Cell line source(s) | K-562 (ATCC) is derived from the pleural effusion of a 53-year-old female with chronic myelogenous leukemia in terminal blast crises. GM12878 (Coriell Institute) is a EBV-transformed B lymphocyte from a female. BJ (ATCC) cells are fibroblasts established from skin taken from normal foreskin from a neonatal male. eHAP (Cellosaurus) is haploid cell derived from HAP1, HAP1 is a near-haploid human cell line derived from KBM7, a human myeloid leukemia cell line developed from a 39-year-old male. |
| Authentication | All cell lines were validated with morphology and gene expression and other epigenetic states with published datasets. |
| Mycoplasma contamination | Mycoplasma contamination test is negative. |
| Commonly misidentified lines (See ICLAC register) | No commonly misidentified cell lines were used in the study. |

# Animals and other research organisms

Policy information about studies involving animals; ARRIVE guidelines recommended for reporting animal research, and Sex and Gender in Research

| | |
|---|---|
| Laboratory animals | Female and male, postnatal day 3-120, CAST/EiJ x C57BL/6J hybrid mice, female and male, postnatal day 1-60, DBA/2J x c57BL/6J hybrid mice. |
| Wild animals | No wild animals were used. |
| Reporting on sex | Both female and male mice were used. The conclusion derived from this study is not biased to specific sex. |
| Field-collected samples | No field-collected samples were used. |
| Ethics oversight | The study was approved by the Peking University Institutional Animal Care and Use Committee (IACUC). All the animal experiments were conducted following their guidelines. |

Note that full information on the approval of the study protocol must also be provided in the manuscript.

# Flow Cytometry

## Plots

Confirm that:

☒ The axis labels state the marker and fluorochrome used (e.g. CD4-FITC).

☒ The axis scales are clearly visible. Include numbers along axes only for bottom left plot of group (a 'group' is an analysis of identical markers).

☒ All plots are contour plots with outliers or pseudocolor plots.

☒ A numerical value for number of cells or percentage (with statistics) is provided.

## Methodology

| | |
|---|---|
| Sample preparation | The mouse main olfactory epithelium was dissociated into single cell suspension with papain. The single-cell suspension was filtered with 40 um strainer. Then 50000 cells were aliquot to ATAC-seq procedure, briefly, cells were permeabilized and transposed, then stain with 7-AAD. |

| | |
|---|---|
| Instrument | BD, FACS Aria SORP |
| Software | BD FACSDiva v9.0 Software |
| Cell population abundance | All nuclei were sorted without biased, the 7-AAD-positive nuclei was selected. |
| Gating strategy | Nuclei were distinguished from debris based on FSC-A and SSC-A, then the multiplets were removed by two step gating of FSC-W and FSC-H, SSC-W and SSC-H. Then nuclei were selected based on PerCP-cy5-5-A. |

☒ Tick this box to confirm that a figure exemplifying the gating strategy is provided in the Supplementary Information.

