## [Peer Review File · Nature Methods]

Peer Review Information

Manuscript Title: Simultaneous single-cell three-dimensional genome and gene expression profiling Uncovers Dynamic Enhancer Connectivity Underlying Olfactory Receptor Choice

Corresponding author name(s): Xiaoliang Xie

Editorial Notes:

Reviewer Comments & Decisions:

Decision Letter, initial version:

25th Sep 2023

Dear Dr Xie,

Your Article, "Simultaneous single-cell three-dimensional genome and gene expression profiling Uncovers Dynamic Enhancer Connectivity Underlying Olfactory Receptor Choice", has now been seen by 3 reviewers. As you will see from their comments below, although the reviewers find your work of considerable potential interest, they have raised a number of concerns. We are interested in the possibility of publishing your paper in Nature Methods, but would like to consider your response to these concerns before we reach a final decision on publication.

We therefore invite you to revise your manuscript to address these concerns. In the revision, all the technical questions raised by reviewer #2 should be fully addressed.

- * include a point-by-point response to the reviewers and to any editorial suggestions
- * please underline/highlight any additions to the text or areas with other significant changes to facilitate review of the revised manuscript
- * address the points listed described below to conform to our open science requirements

* ensure it complies with our general format requirements as set out in our guide to authors at www.nature.com/naturemethods

* resubmit all the necessary files electronically by using the link below to access your home page

[REDACTED]

We hope to receive your revised paper within 12 weeks. If you cannot send it within this time, please let us know. In this event, we will still be happy to reconsider your paper at a later date so long as nothing similar has been accepted for publication at Nature Methods or published elsewhere.

OPEN SCIENCE REQUIREMENTS

REPORTING SUMMARY AND EDITORIAL POLICY CHECKLISTS

IMAGE INTEGRITY

When submitting the revised version of your manuscript, please pay close attention to our Digital Image Integrity Guidelines and to the following points below:

- that unprocessed scans are clearly labelled and match the gels and western blots presented in figures.
- that control panels for gels and western blots are appropriately described as loading on sample processing controls

-- all images in the paper are checked for duplication of panels and for splicing of gel lanes.

DATA AVAILABILITY

All novel DNA and RNA sequencing data, protein sequences, genetic polymorphisms, linked genotype and phenotype data, gene expression data, macromolecular structures, and proteomics data must be deposited in a publicly accessible database, and accession codes and associated hyperlinks must be provided in the "Data Availability" section.

CODE AVAILABILITY

Please include a "Code Availability" subsection in the Online Methods which details how your custom code is made available. Only in rare cases (where code is not central to the main conclusions of the paper) is the statement "available upon request" allowed (and reasons should be specified).

For more information on our code sharing policy and requirements, please see:
<https://www.nature.com/nature-research/editorial-policies/reporting-standards#availability-of-computer-code>

SUPPLEMENTARY PROTOCOL

To help facilitate reproducibility and uptake of your method, we ask you to prepare a step-by-step Supplementary Protocol for the method described in this paper. We encourage authors to share their step-by-step experimental protocols on a protocol sharing platform of their choice and report the protocol DOI in the reference list. Nature Portfolio 's Protocol Exchange is a free-to-use and open resource for protocols; protocols deposited in Protocol Exchange are citable and can be linked from the published article. More details can found at www.nature.com/protocolexchange/about.

ORCID

Nature Methods is committed to improving transparency in authorship. As part of our efforts in this direction, we are now requesting that all authors identified as 'corresponding author' on published papers create and link their Open Researcher and Contributor Identifier (ORCID) with their account on the Manuscript Tracking System (MTS), prior to acceptance. This applies to primary research papers only. ORCID helps the scientific community achieve unambiguous attribution of all scholarly contributions. You can create and link your ORCID from the home page of the MTS by clicking on 'Modify my Springer Nature account'. For more information please visit please visit www.springernature.com/orcid.

Sincerely,
Lei

Lei Tang, Ph.D.
Senior Editor
Nature Methods

Reviewers' Comments:

Reviewer #1:

Remarks to the Author:

This manuscript by Wu et al reports a new single-cell multi-omics approach to simultaneously characterize gene expression and genome architecture. Methods development in this area are moving rapidly. Overall, the approach appears quantitative, rigorous, and seems to be a significant advance based on (ample) compelling data. The authors have chosen an interesting question, olfactory receptor gene choice, to apply their new approach. I found the results exciting and novel. I think this will be of broad interest and will add important insight into the basic properties of olfactory gene regulation. The manuscript itself needs some editing for clarity, in particular the Discussion and some of the figure legends. In addition, I have a few outstanding questions and comments.

General Comments

1. It seems like the authors are trying to fit their data into the Lomvardas model of OR gene expression, despite their results appearing to argue against some aspects of it. It is safe to say that the authors are not going out of their way to point out very strongly the inconsistency of their observations with the published model. The premise of the Lomvardas model had been that H is the singular enhancer that chooses one locus. Then the model became H and the Greek Islands forming a single superenhancer that chooses one locus. More recently, the Lomvardas lab has acknowledged that there are multiple enhancer aggregates and they have shifted to a "winner-take-all" model. In all of this chaos, the current manuscript clearly shows (better than in any prior publications) that the nucleus contains several enhancer clusters, only one of which is associated with the expressed OR—not always the largest one. This argues against the notion that singularity of OR expression can be explained by formation of single super enhancer. The data also show that there is nothing extraordinary about the location of the chosen OR with respect to the enhancer aggregations. The mechanism of singularity is completely unknown. The authors may be unwilling to rock the boat too much in a methods paper, but I think all of this should be stated very clearly and strongly in the Results, and addressed in the Discussion as it would increase the significance of the conclusions.
2. On a related note, the authors put forth a model Figure 3h in which Lhx2 expression and enhancers accessibility precede OR accessibility/expression. It is difficult to see on what basis the authors order these events. The current data do not seem to have the temporal resolution to resolve the sequence of events at this level of detail.
3. Based on their new approach, and published data, the authors put forth an additional set of putative OR enhancers. The authors should be clear about the quantitative criteria used to call these new enhancers. It is difficult to see from Figure 2f that the novel sites are always marked by peaks as stated in the authors' criteria. Also, what is one supposed to conclude about ChIP-seq signals that span an entire OR gene cluster?
4. To determine the phases of OR gene choice (Figure 3a), it looks like the authors omitted cells showing singular expression when below $\log_{10}(\text{FPKM}+1)=3$ or an FPKM of 999. If I read this correctly, it is not entirely clear why cells with this level of expression were omitted.

Specific comments

- Line 42: "...captured a small fraction of the whole cell's RNA..." The authors critique a previous method (only capturing nuclear RNA) is overly harsh. The method captures actively transcribed genes, which would appear to be the most relevant to compare with a snapshot of genomic architecture. Of course this also has its limitations. The authors might say the method was "limited to capturing nuclear RNA".
- Line 68: "...rather than bulk Hi-C proposed..." This is not clear. Each OSN forms multiple hubs in contrast to a single hub proposed based on bulk Hi-C data? (This paragraph needs editing for clarity and grammar).
- Line 121: "...provides full length transcript information..." Do the authors make use of this information in the current study? The relevance and strength of the method would be underscored if full length transcript information were more obviously used in a practical way in the manuscript. From Extended Figure 7, it looks like this information was used to call genuine OR expression? Perhaps this should be pointed out more strongly in the Results as an advantage of the approach.
- Line 177: "...comprised of 412 cells..." The sum of the numbers in Figure 2 is 411. Is this an error?
- Line 245: "...but one is leaved." Should read "but one remains"
- Line 554 and elsewhere: There are references missing
- The authors should go through the figure legends carefully and make sure all graph elements are described clearly. Some examples...
- Figure 1g: Define what the green and yellow dots/bars mean.
- Figure 2b: The entries for scRNAseq are shifted to the right and do not line up with the experimental groups (ages). Is this intentional? Also, it would be helpful if the authors changed the blank spaces to ND (for not done) if that is the case.

Figure 1D: The authors should state in the legend and Results what type of figure this is (imputed contact map?). Also they should provide more information/clarity regarding how these nuclear architecture figures are made. I could not find much information in the text and the in-text reference was missing. Also, the labeling scheme makes it difficult to keep track of the expressed ORs and enhancers. I would suggest that the authors show the OR genes with different shapes than the enhancers. Maybe the OR genes would be squares and the enhancers dots (as in Figure 3h). Also, the fact that colors are assigned randomly for ORs and enhancers is confusing. Color could be used more effectively. For example, show the different ORs as shades of a given color family, and show the different enhancers as shades of a different color family. That way the reader can immediately compare where the ORs and the enhancers are by color family and shape.

Figure 3f and 3g: It is difficult to know how to read these graphs. It is not immediately clear what "16 vs 4" means.

Figure 3f x-axis of reads "Second Enhancer Aggregate". Do the authors mean "Second Largest Enhancer Aggregate"?

Figure 2f: The figure compares ATAC and ChIPseq data. Again, color seems to be used arbitrarily. Could the authors use color to differentiate ATAC vs ChIPseq data sets? Again they could use shades of color to differentiate data of the same type.

Reviewer #2:

Remarks to the Author:

In the manuscript titled "Simultaneous Single-Cell Three-dimensional Genome and Gene Expression Profiling Uncovers Dynamic Enhancer Connectivity Underlying Olfactory Receptor Choice", Wu et al. introduce LiMCA (Linking mRNA Chromatin Architecture), a new approach to map simultaneously chromatin organisation and mRNA in single cells. The authors first validated their method using the GM12878 cell line and further applied LiMCA to the developing murine Olfactory Sensory Neurons (OSNs). Wu et al. also produced single-cell ATAC and RNA atlases of the developing OSN and, in combination with the LiMCA data, corroborated previous findings and extended the current knowledge of how single Olfactory Receptor (OR) choice is made. The authors identified new candidate enhancers that potentially play a role in OR choice and examined the interplay between the whole set of candidate enhancers and OR expression dynamics throughout different stages of OSN development. Their work suggests that only the enhancer hub associated with the expressed OR remains active at the latest stage of maturation, while the rest of the enhancers aggregate and associate with non-expressed ORs to become progressively silenced during maturation.

The newly developed method and the datasets generated are highly promising and of great interest to the community. However, some of the claims are not well supported, and require further evidence before publication.

Major comments

1. In lines 102-104, the manuscript states that the contacts detected by LiMCA and Dip-C are of 'identical proportions'. However, this is not evident in Extended Figure 1b in which LiMCA appears to detect a higher percentage of contacts at short range and a lower percentage at longer range. The authors must provide statistical evidence to support their claims, on this topic and throughout and throughout.

2. The parameters used to select the cells used to produce the high-resolution chromatin organisation data need to be clearly explained. In Extended Figure 1g, many of the chromatin loops identified in ensemble LiMCA are missed when compared to in situ Hi-C. The manuscript does not provide evidence for the claims made in lines 104-108 that there is a high concordance between LiMCA and Hi-C. What

would lack of concordance look like? It is essential to explain the relationship between the number of cells that comprise the ensemble LiMCA and the proportion of contacts detected that overlap with the in situ Hi-C.

3. The criteria applied to select the cells with high-resolution contacts must be stated more clearly. The criteria and evidence used to claim that these datasets are of good quality at 20 kb resolution needs to be provided in the manuscript to enable reproduction of the study and further applications of the approach.

4. The methods applied for phasing the data are also not clear, and a full description is required. This includes a quantification of the efficiency of phasing, together with an evaluation of whether the efficiency varies between genomic regions, and a clear discussion of how limited or uneven power to phasing contacts affects the possible outcomes of the work.

5. The results in Figure 1g are potentially very interesting but unfortunately lack sufficient evidence. Additional information is required to understand whether the differences observed are simply due to the number of cells, and to variable power to detect contacts, included in each condition. The manuscript needs to state how many cells were included for each condition. In case there is great variation in the number of cells, then the effect of cell numbers needs to be assessed and included in the manuscript, together with analyses of sub-samples with the same numbers of cells per condition (selected randomly from each set) to test the validity of the results using different numbers of cells. The differences in contacts between the two groups should be represented by a differential matrix for better visualization. In particular, the manuscript must explicitly report whether the number of genes and UMIs per cell are comparable between cells above medium NFKB1 expression and cells below medium NFKB1 expression.

6. The methodology used to measure the compaction of NFKB1 to support the claim the authors make in lines 152-154 needs to be clearly described to enable its evaluation and reproduction.

7. Full details about how random controls were generated across the study (e.g. Figure 1h, i) must be included, and are essential to understand the meaning and significance of the results presented and conclusions.

8. The table presented in Figure 2b is confusing. It is not clear why two different mouse strains were used in the study, and especially why, for some time points, the data was produced from one mouse strain only. The manuscript should explicitly state the motivation for using two different mouse strains in the LiMCA experiments. A revised manuscript has to clarify or provide additional evidence to explain/test whether the differences in the downstream analysis are not due to differences in the mouse lines used. Are the samples from the same animal (for instance were the 70 cells used for P28 from one animal only)? Was the scRNA-seq data produced from 3 different developmental time points or were the samples pulled from different developmental stages?

9. The results presented in Figure 2c require additional controls to rule out that the clustering of the progenitors is not due to differences in the mouse lines used or the number of contacts per cell.

10. The contribution of scRNA-Seq data produced in the study must be made clear in the manuscript. Are the cell stages 1-3 identified with LiMCA RNA also reflected in the scRNA-Seq pseudo-time? Do these 3 groups also follow the pseudo-timeline identified in the scRNA-Seq. This information can be extracted from Extended Figure 8e.

11. Extended data Figure 10g, the differences in contact densities between expressed ORs and randomly selected ORs do not seem to be considerably different. A more quantitative approach is

required to support the claims in lines 279-282. Further, what do numbers displayed in the top left corner of the matrices represent?

Minor comments

12. A schematic overview of LiMCA is presented in Figure 1a, however, some of the labels are not clear. What does DMSO 19 refer to? The example of the contact map and gene expression tracks shown is missing the genomic coordinates, are these also produced from cell 100? Please clarify.

13. On the contact maps shown in Figure 1b, black boxes and numbers are displayed. What do they represent? Please elaborate in the figure text.

14. Was the bulk RNA-Seq in Figure 1d produced or obtained from published data? If produced then the protocol should be included in Methods, and if from published data then reference and data resource should be provided.

15. The schematic in Figure 2a is not clear. Are the schematics for the olfactory epithelium supposed to represent the different developmental stages? If so, we suggest changing it either according to the number of time points used in the study (six and not three) or making the developmental timeline more fluent and visual. Also, what does 'sc joint Hi-C-RNA' refer to?

16. There are some inconsistencies in the abbreviations used. For instance, METATAC and single-cell ATAC-Seq are often used interchangeably, and the naming of progenitor cells is inconsistent (e.g. in Figure 2c are called progenitors and in Figure 2e are called INPs). To make it easier for the reader to follow, we highly recommend to use the same abbreviations throughout the paper. Please also include a description of the abbreviation in the figure legend when used in figures.

17. The figure references in the text should be checked and updated. For instance, we want to draw the attention of the authors to the figure references in lines 142, 154, 163, 164, 165 and 199.

18. In lines 184-186, the authors claim that four clusters are obtained with RNA and 3C (visualised in Figure 2c). However, based on the gene expression UMAP (left), five clusters are defined; non-neuronal, progenitor-1, progenitor-2, iOSN, mOSN, while the number of clusters obtained with 3C (right) also is not completely clear. The authors should rephrase the corresponding lines to be in agreement with the UMAPs or provide additional explanations for their conclusion.

19. In the figure legend for Extended Figure 3i, k: the scatterplot shows the expression level vs the normalized radial position (not the other way around as written in the legend)

20. In Extended Figure 4a the discrete labels that the authors propose for each group of progenitors do not correspond to the developmental trajectory from the pseudo-time analysis (Extended Figure 4 d). Could the authors comment on why this is the case?

21. The explanation for how the pseudo-time in the ATAC-seq dataset was performed is missing. Please clarify how early and late INPs were defined in the ATAC-seq data and why these stages are not represented in the scRNA-seq pseudo-time.

22. The authors should be more specific about which cell types were used for each of the analyses. If missing, please indicate either in the text, on the figures or in the figure legend.

23. The colour coding is inconsistent and several plots/markings are missing labels. For instance, please

specify what the dots/lines in Figure 1g and Extended Figure 3c-f correspond to, and be consistent with colours (yellow in Figure 1g vs blue in Extended Figure 3c-f). Furthermore, it is not clear what the different colours of the OR enhancers represent (are the OR enhancers from different chromosomes?) (Figure 2d and Figure 3b,d-e). Extended Figure 4b and Extended Figure 10a are missing axis labels. Figure 2g is missing a control to show that the enrichment is specific for the newly identified candidate peaks.

24. Extended Figure 9 lacks a figure legend and isoform labelling in the figure. Please provide.

25. The text and figures should be checked for general typos. To highlight some: in Figure 1a (right) and Extended Figure 1h "CpG Frenquency" should be changed to "CpG Frequency"; in figure text for Extended Figure 3h "Scatterplot" should be changed to "Dot plot"; Extended Figure 4f the y-axis should be changed from 300nm to 150nm; in the figure text for Extended Figure 5d,e "scatterplot" should be changed to "boxplot".

Reviewer #3:

Remarks to the Author:

The manuscript by Xie and colleague describes a powerful new protocol called LimCA (Linking mRNA to Chromatin Architecture) that enables multiome single cell analysis of mRNA and genomic contacts. Since the emergence of single cell HiC technologies, there is an increasing need to combine genomic contacts with RNA expression information. A recently published protocol, termed HiRES, provided the first such methodology, with LimCA emerging as a significant improvement of coverage both at the HiC (with an impressive 1million contacts per cells) and the RNA detection front. The authors validate their methodology in commonly used cell lines, allowing direct comparison with other protocols and then they apply this protocol to the mouse olfactory neurons. These neurons provide the ideal biological system for LimCA, since previous work established an intimate connection between the assembly of multi-enhancer hubs and the transcriptional activation of one out of ~1000 olfactory receptor genes. Using this system, the authors make several important observations that highlight the power of their new technology. Specifically:

1. They show that at the early polygenic state, co-transcribed olfactory receptor genes associate with enhancer hubs preferentially consisting of enhancers from the same chromosome.
2. They show that among the competing olfactory receptor/hub combinations, usually the gene that has the higher number of enhancers in close proximity is the one that is more highly transcribed, providing a striking demonstration of the synergy between these enhancers.
3. They show that in mature olfactory neurons, the prevailing gene is often not associated with the hub that has the most enhancer, a puzzling finding considering point#2.
4. They also perform scATAC/RNA-seq experiments that reveal interesting dynamics between enhancer accessibility, transcription factor expression and hub assembly.

From the biological perspective, these are important new discoveries that indeed clarify that process of olfactory receptor expression and suggest a positive feedback mechanism as a potential mechanism that transforms polygenic transcription to singular transcription in terminally differentiated neurons. From the technical perspective, it appears that this approach has the potential to surpass HiRES, providing a powerful tool for the general community. Thus, I am in favor of publication. However, I want to propose a few changes, clarifications and new analyses.

1. The authors describe the discovery of numerous new olfactory receptor enhancers that have the same motif as the previously described enhancers, and they are co-bound by Lhx2 and Ebf. I did not find description of ChIP-seq experiments, thus I am not sure where is the evidence that these new putative enhancers are co-bound by these two factors. Are they using the ChIP-seq data from Monahan

et al? If yes, and these elements are co-bound by these two TFs and have a composite motif, why they were not called as such by these authors?

2. I am trying to think of explanations for the fact that the hub that is associated with the chosen OR allele is not the biggest hub, which would be expected if the positive feedback loop is correct. One explanation I can think, is that the zonal properties of the olfactory receptors in the biggest hubs are more dorsal than the identity of the prevailing olfactory receptor: According to Bashkirova et al, the biggest hubs they would contain more dorsal receptor genes, they would have formed earlier, and they would subsequently become heterochromatic. Since the authors have zonal information of every receptor, they could easily explore this interesting possibility which would add extra biological value to their work. Even if this hypothesis is wrong, it would be very informative if the authors do a zonal analysis in their data (for example are the Zonal restrictions described by Bashkirova in late stages of polygenic expression related to hub interactions?)

3. I would have loved to see LimCA performed on olfactory neurons selected based on the expression of a GFP reporter driven by an olfactory receptor gene (ORiresGFP), not only as a confirmation for the method but also to obtain an understanding of the false negative and false positive rates of the technique. However, if such lines are not available, I would not wish to delay publication for this control experiment.

Author Rebuttal to Initial comments

Reviewer #1:

Remarks to the Author:

This manuscript by Wu et al reports a new single-cell multi-omics approach to simultaneously characterize gene expression and genome architecture. Methods development in this area are moving rapidly. Overall, the approach appears quantitative, rigorous, and seems to be a significant advance based on (ample) compelling data. The authors have chosen an interesting question, olfactory receptor gene choice, to apply their new approach. I found the results exciting and novel. I think this will be of broad interest and will add important insight into the basic properties of olfactory gene regulation. The manuscript itself needs some editing for clarity, in particular the Discussion and some of the figure legends. In addition, I have a few outstanding questions and comments.

We want to express our utmost gratitude for the Reviewers' enthusiasm and positive remarks regarding our manuscript. The Reviewer's comment characterizing our results as "exciting and novel" truly resonates with us and we sincerely appreciate the Reviewer for their kind words.

General Comments

1. It seems like the authors are trying to fit their data into the Lomvardas model of OR gene expression, despite their results appearing to argue against some aspects of it. It is safe to say that the authors are not going out of their way to point out very strongly the inconsistency of their observations with the published model. The premise of the Lomvardas model had been that H is the singular enhancer that chooses one locus. Then the model became H and the Greek Islands forming a single superenhancer that chooses one locus. More recently, the Lomvardas lab has acknowledged that there are multiple enhancer aggregates and they have shifted to a “winner-take-all” model. In all of this chaos, the current manuscript clearly shows (better than in any prior publications) that the nucleus contains several enhancer clusters, only one of which is associated with the expressed OR—not always the largest one. This argues against the notion that singularity of OR expression can be explained by formation of single super enhancer. The data also show that there is nothing extraordinary about the location of the chosen OR with respect to the enhancer aggregations. The mechanism of singularity is completely unknown. The authors may be unwilling to rock the boat too much in a methods paper, but I think all of this should be stated very clearly and strongly in the Results, and addressed in the Discussion as it would increase the significance of the conclusions.

We appreciate the Reviewers for thorough review of the related publications from the Lomvardas lab. The model proposed by Lomvardas lab is primarily based on bulk 4C/Hi-C experiments on FACS-purified mature OSNs expressing a specific OR gene. They observed that active OR interacts most frequently with all *trans* and long-range *cis*-enhancers, leading them to propose that the finally chosen OR interacts with the largest number of OR enhancers, as bulk 4C/Hi-C experiments could not capture variability at single-cell level.

In our study, we have presented three key findings. Firstly, we highlighted the important role of *cis*-enhancers in activating multigenic OR expression at progenitor stage, which is a novel contribution. Secondly, we revealed that the dominant ORs are associated with more enhancers compared those fail to compete, providing an explanation for their dominance. Thirdly, we found that the active OR in mOSNs is typically not located within the largest enhancer aggregate, which refutes Lomvardas model.

Although our findings contradict the Lomvardas model, it's important to note that both of our single-cell data and their bulk 4C/Hi-C data are correct. Our single-cell data provide alternative

explanation for the bulk data, as we have discussed in the revised manuscript (please refers to the discussion section of revised manuscript).

Again, we thank the Reviewers for their thoughtful suggestions and comments. In the revised manuscript, we have made a clear and strong statement of our results and conclusion in the “Stepwise olfactory receptor determination and their spatial relationship with Greek islands revealed by joint profiling of chromatin architecture and OR expression” section (Page 10, line 283-286; Page 10, line 290-294; Page 10, line 296-300, Page 11, line 318-323). Furthermore, we also added a detailed discussion for the inconsistency between our finding and Lomvardas model in the **discussion** section (Page 12-13, line 355-364). We hope that these revisions address the Reviewers’ concerns and improve the manuscript's clarity.

2. On a related note, the authors put forth a model Figure 3h in which Lhx2 expression and enhancers accessibility precede OR accessibility/expression. It is difficult to see on what basis the authors order these events. The current data do not seem to have the temporal resolution to resolve the sequence of events at this level of detail.

We thank the Reviewers for their valuable comment. We would like to apologize for any confusion caused by the schematic representation in Figure 3h. Based on our METATAC data, we have observed that the accessibility of OR enhancers and OR genes are synchronized, as depicted in Figure 2i.

In order to figure out whether Lhx2 expression precedes OR enhancer activation, we additionally performed an integration analysis of METATAC and scRNA-seq datasets by extracting the continuous developmental lineage from GBC, early/late INP, iOSN to mOSN. According to the pseudotime analysis on the integrated data, we found that Lhx2 expression happens before OR enhancer accessibility. Thus, we could unambiguously order Lhx2 expression before OR enhancer activation, and OR gene accessibility changes concomitantly with OR enhancers (see the modified schematics Fig. 3h).

Furthermore, we have updated this new analysis to the revised manuscript (Page 9, line 246-254) and figures (Fig. 2j-l and Extended Data Fig. 8i). We hope that these revisions will address the reviewers' concerns and provide greater clarity in our manuscript.

3. Based on their new approach, and published data, the authors put forth an additional set of putative OR enhancers. The authors should be clear about the quantitative criteria used to call these new enhancers. It is difficult to see from Figure 2f that the novel sites are always marked by peaks as stated in the authors' criteria. Also, what is one supposed to conclude about ChIP-seq signals that span an entire OR gene cluster?

We thank the Reviewers for this comment. The newly identified enhancers in our study were called using the same criteria that were previously used to identify OR enhancers (Monahan et al., *eLife* 2017). These criteria include their location within OR gene clusters, presence of open chromatin peaks in mature OSNs, as well as co-binding of Lhx2 and Ebf. In Figure 2f, the relative scale of each track has been normalized to the highest value within individual OR gene clusters. However, it is important to note that some putative or

previously identified enhancers may not appear to meet the criteria due to relatively lower values compared to stronger enhancers within the same OR gene cluster.

To provide more detailed information, we have included additional data in Extended Data Figure 7c and d. These figures display the ATAC peaks, Lhx2 binding, and Ebf binding for each individual enhancer. This provides a comprehensive view of the specific characteristics of each enhancer and allows for a more thorough evaluation.

4. To determine the phases of OR gene choice (Figure 3a), it looks like the authors omitted cells showing singular expression when below $\log_{10}(\text{FPKM}+1) = 3$ or an FPKM of 999. If I read this correctly, it is not entirely clear why cells with this level of expression were omitted.

We sincerely appreciate the Reviewers for bringing up this concern. In response, we have included two supplementary figures (Extended Data Figure 10a and b) to provide a clearer understanding of how the phases of OR gene choice were determined.

To address this, we would like to clarify that the majority of these omitted cells were progenitor cells (9 progenitor cells, 3 iOSN, and only 1 mOSN). On the other hand, most iOSN and mOSN cells have total OR expression > 1000 FPKM, and cells showing singular expression and with a total OR expression greater than 1000 were largely iOSN and mOSN, with only 1 progenitor cell among them. Therefore, both the OR expression level and developmental stage suggests that the omitted cells were distinct from Stage 3 cells.

We identified these cells as “outliers”, and excluded them for downstream analysis, as we couldn't ascertain whether these "singular" ORs were ultimately selected and there were no other expressed ORs available for meaningful comparisons regarding OR-GI interactions. Furthermore, we have corrected the y-axis label in Figure 3a, which should read $\log_{10}(\text{FPKM})$ instead of $\log_{10}(\text{FPKM}+1)$.

We have added this updated figure in our revised figures (Extended Data Fig. 10a-b).

Specific comments

1. Line 42: “...captured a small fraction of the whole cell’s RNA...” The authors critique a

previous method (only capturing nuclear RNA) is overly harsh. The method captures actively transcribed genes, which would appear to be the most relevant to compare with a snapshot of genomic architecture. Of course this also has its limitations. The authors might say the method was “limited to capturing nuclear RNA”.

We have modified the corresponding text (Page 2, Line 40-43), which is “The only published sequencing-based methods, HiRES, had limited sensitivity (~0.3 million contacts per cell) because genomic DNA was damaged during reverse transcription, captured only nuclear RNAs because the cytoplasm was destroyed during the procedure, and only detected the 3’ end of the transcript.”

2. Line 68: “...rather than bulk Hi-C proposed...” This is not clear. Each OSN forms multiple hubs in contrast to a single hub proposed based on bulk Hi-C data? (This paragraph needs editing for clarity and grammar).

We sincerely appreciate the Reviewers for this suggestion. We have carefully revised the paragraph (Page 3, line 64-73) to enhance clarity. The updated version is as follows: “However, this model fails to address several unresolved issues. Firstly, during OSN development, progenitors transiently express random sets of OR genes^{19,20}. Additionally, the onset of multigenic OR expression precedes the formation of repressive OR-OR compartments. Furthermore, each OSNs forms multiple enhancer aggregates, rather than a singular one proposed based on bulk Hi-C data. Unfortunately, existing bulk and single-cell techniques are unable to resolve these mysteries due to the lack of OR expression information and an inability to isolate a population expressing a random set of ORs. Ideally, a technique that can simultaneously measure OR expression and 3D genome organization in the same cells would be necessary to elucidate how OR selection process is initiated and proceeded.”

3. Line 121: “...provides full length transcript information...” Do the authors make use of this information in the current study? The relevance and strength of the method would be underscored if full length transcript information were more obviously used in a practical way in the manuscript. From Extended Figure 7, it looks like this information was used to call genuine OR expression? Perhaps this should be pointed out more strongly in the Results as an advantage of the approach.

We sincerely apologize for the lack of clarity in our previous description. In our study, we indeed relied on the full-length transcript information to accurately identify genuine OR

expression. Specifically, we only considered OR genes with a fully covered coding sequence as indicative of true expression. This approach was necessary due to the presence of truncated and non-coding transcripts often found in OR genes that lack functionality. To address this point more clearly, we have revised our manuscript to emphasize the importance of utilizing full-length transcript information for accurate identification of genuine OR expression. As stated in the revised version (Page 9-10, line 269-273), “The presence of truncated and non-functional olfactory receptor (OR) transcripts necessitates the utilization of full-length transcript information, a feature uniquely provided by LiMCA as opposed to HiRES. This capability plays a crucial role in accurately discerning genuine OR expression, as demonstrated in Extended Data Figure 9.”

4. Line 177: “...comprised of 412 cells...” The sum of the numbers in Figure 2 is 411. Is this an error?

We sincerely appreciate the meticulousness of the reviewers in noticing this discrepancy. We apologize for the typographical error in the manuscript. Indeed, the correct number of cells in Figure 2 is 411, not 412.

5. Line 245: “...but one is leaved.” Should read “but one remains”

We have revised the corresponding text (Page 9, line 260).

6. Line 554 and elsewhere: There are references missing.

We apologize for this error. We have promptly addressed this issue by including the relevant references in the revised version of the text.

7. The authors should go through the figure legends carefully and make sure all graph elements are described clearly. Some examples...

Figure 1g: Define what the green and yellow dots/bars mean.

We sincerely appreciate the Reviewers' feedback regarding the clarity of the figure legends. In response to this concern, we have revisited the figure legends (Page 30, line 787-line789) and made the necessary revisions to ensure all graph elements are clearly defined.

Specifically, for Figure 1g, we have now included a detailed figure legend that explicitly states the meaning of the green and yellow dots/bars. The green dot represents a potential NFkB1 enhancer, while the yellow dots represent the transcription start site (TSS) and transcription termination site (TTS) of the NFkB1 gene, respectively.

8. Figure 2b: The entries for scRNAseq are shifted to the right and do not line up with the experimental groups (ages). Is this intentional? Also, it would be helpful if the authors changed the blank spaces to ND (for not done) if that is the case.

We sincerely appreciate the valuable suggestion from the reviewers regarding Figure 2b. In response to this feedback, we have made the necessary modifications to improve the clarity of the figure.

Firstly, we have replaced the blank spaces in Figure 2b with “ND” (not done) to indicate when data was not available for certain experimental groups.

Furthermore, we would like to clarify that the shifting of entries for single-cell RNA-seq in Figure 2b is intentional. We conducted three 10x runs, one of which involved pooling samples from postnatal day 4 (P4) and postnatal day 7 (P7). The sampling details have been thoroughly documented in the revised methods section (**Droplet scRNA-seq**) (Page 16, line 435-442).

9. Figure 1D: The authors should state in the legend and Results what type of figure this is (imputed contact map?). Also they should provide more information/clarity regarding how these nuclear architecture figures are made. I could not find much information in the text and the in-text reference was missing. Also, the labeling scheme makes it difficult to keep track of the expressed ORs and enhancers. I would suggest that the authors show the OR genes with different shapes than the enhancers. Maybe the OR genes would be squares and the enhancers dots (as in Figure 3h). Also, the fact that colors are assigned randomly for ORs and enhancers is confusing. Color could be used more effectively. For example, show the different ORs as shades of a given color family, and show the different enhancers as shades of a different color family. That way the reader can immediately compare where the ORs and the enhancers are by color family and shape.

We are grateful to the Reviewers for their valuable suggestion. We apologize for any confusion regarding the generation of nuclear architecture and the labeling scheme in our figures. To address this, we have added a new section in the Methods section to provide detailed information on the generation of the nuclear architecture.

In brief, three-dimensional nuclear architectures were generated using dip-c (<https://github.com/tanlongzhi/dip-c>) and hickit (<https://github.com/lh3/hickit>) package, and visualized using PyMOL (<https://pymol.org/2/>). After creating the whole cell structures, the enhancer and OR genes were located with “dip-c pos” from the whole structure. We have thoroughly reorganized and improved the “3D genome structure analysis” methods section, please refers to the revised methods (3D genome structure analysis).

Unfortunately, the shapes of enhancers and ORs are not allowed to modify in PyMOL. In the original figures, we used the rainbow color set to distinguish OR enhancers from 17 chromosomes (one color for one chromosome). We apologize for any distraction caused by the rainbow color scheme we used previously. In response to the suggestion, we have

revised our figures and used a sequential color set from blue to red to label enhancers. We hope that this improves the clarity of our figures.

10. Figure 3f and 3g: It is difficult to know how to read these graphs. It is not immediately clear what “16 vs 4” means.

We apologize for the confusion. We have modified the figures and figure legend to clarify the figures. Specifically, we change the label “16 vs. 4” into “16 cells vs. 4 cells” and color them according to the line trend, moreover, we supplemented a Two-sided Mann-Whitney U test for significance test, please see revised Fig.3 f.

11. Figure 3f x-axis of reads “Second Enhancer Aggregate”. Do the authors mean “Second Largest Enhancer Aggregate”?

We are sorry for the confusion. We have changed the “Second Enhancer Aggregate” into “Second Largest Enhancer Aggregate” in the revised figure, please see Fig. 3g.

12. Figure 2f: The figure compares ATAC and ChIPseq data. Again, color seems to be used arbitrarily. Could the authors use color to differentiate ATAC vs ChIPseq data sets? Again they could use shades of color to differentiate data of the same type.

We thank the Reviewers for this suggestion. In the revised figure, we have modified the colors to make ATAC and ChIP-seq more distinguishable, please see Fig. 2f.

Reviewer #2:

Remarks to the Author:

In the manuscript titled “Simultaneous Single-Cell Three-dimensional Genome and Gene Expression Profiling Uncovers Dynamic Enhancer Connectivity Underlying Olfactory Receptor Choice”, Wu et al. introduce LiMCA (Linking mRNA Chromatin Architecture), a new approach to map simultaneously chromatin organisation and mRNA in single cells. The authors first validated their method using the GM12878 cell line and further applied LiMCA to the developing murine Olfactory Sensory Neurons (OSNs). Wu et al. also produced single-cell ATAC and RNA atlases of the developing OSN and, in combination

with the LiMCA data, corroborated previous findings and extended the current knowledge of how single Olfactory Receptor (OR) choice is made. The authors identified new candidate enhancers that potentially play a role in OR choice and examined the interplay between the whole set of candidate enhancers and OR expression dynamics throughout different stages of OSN development. Their work suggests that only the enhancer hub associated with the expressed OR remains active at the latest stage of maturation, while the rest of the enhancers aggregate and associate with non-expressed ORs to become progressively silenced during maturation.

The newly developed method and the datasets generated are highly promising and of great interest to the community. However, some of the claims are not well supported, and require further evidence before publication.

We deeply appreciate the Reviewers' enthusiastic assessment of our manuscript. We would like to express our sincere gratitude to the Reviewers for their thorough review and thoughtful comments on our manuscript. We are truly grateful for their many insightful questions and suggestions, which we believe have significantly contributed to the quality and clarity of our work.

Major comments

1. In lines 102-104, the manuscript states that the contacts detected by LiMCA and Dip-C are of 'identical proportions'. However, this is not evident in Extended Figure 1b in which LiMCA appears to detect a higher percentage of contacts at short range and a lower percentage at longer range. The authors must provide statistical evidence to support their claims, on this topic and throughout and throughout.

We thank the Reviewers for this comment. We apologize for any confusion caused by the imprecise language used in lines 102-104 of our manuscript. We fully acknowledge that the contacts detected by LiMCA and Dip-C are not of "identical proportions," but instead, LiMCA detects a higher percentage of short-range contacts and a lower percentage of long-range contacts than Dip-C.

We agree that statistical evidence is required to support our claims throughout the manuscript, including this point. We have revised the manuscript text accordingly (Page 4, line 102-104). We further suggest that this difference may be due to the use of different restriction enzymes (NlaIII versus MboI) and potential variance in digestion efficiency in single cells compared to bulk samples.

2. The parameters used to select the cells used to produce the high-resolution chromatin organization data need to be clearly explained. In Extended Figure 1g, many of the chromatin loops identified in ensemble LiMCA are missed when compared to in situ Hi-C. The manuscript does not

provide evidence for the claims made in lines 104-108 that there is a high concordance between LiMCA and Hi-C. What would lack of concordance look like? It is essential to explain the relationship between the number of cells that comprise the ensemble LiMCA and the proportion of contacts detected that overlap with the in situ Hi-C.

We thank the Reviewers for valuable feedback. We have carefully considered their comments and made the following revisions to address their concerns.

In response to the first point, we agree that we could have provided more details about the parameters used to select the cells for our high-resolution chromatin organization data. Specifically, we filtered out cells with fewer than 100,000 unique contacts. We have updated the Methods section to include this information (Page 22, line 600-602). Regarding the comparison of chromatin loops detected by ensemble LiMCA and *in situ* Hi-C, we acknowledge that there were some discrepancies between the two methods. However, we would like to emphasize that the precision rate of our data (6011/8123 (0.74)), which is detected by Chromosight on ensemble LiMCA data, is consistent with the performance reported in the original Chromosight paper (0.75) (Matthey-Doret et al., *Nature Communications* 2020) (as shown in their Fig. 1c, below). To further examine the relationship between the number of cells used and the proportion of chromatin loops detected that overlap with *in situ* Hi-C, we performed a downsample analysis. We found that the proportion of chromatin loops detected that overlap with *in situ* Hi-C reached a plateau after 150 cells, accompanied by the increasing number of chromatin loops detected. We have added these results to the revised manuscript.

The conclusion of high concordance between ensemble LiMCA and bulk Hi-C is evidenced by the high correlation of A/B compartment (Fig. 1c) and insulation score (Extended Data Fig. 1d) between ensemble LiMCA and bulk Hi-C for GM12878. To further demonstrate this conclusion, we have calculated the correlation for all four cell lines and added this information to the revised Extended Data Fig. 1, which confirms the high concordance between ensemble LiMCA and bulk Hi-C within the same cell types and low correlation across cell types (lack of concordance).

Overall, we hope that these additional analyses and explanations address the Reviewers' concerns.

3. The criteria applied to select the cells with high-resolution contacts must be stated more clearly. The criteria and evidence used to claim that these datasets are of good quality at 20 kb resolution needs to be provided in the manuscript to enable reproduction of the study and further applications of the approach.

We sincerely appreciate this suggestion from the Reviewers. Prior to 3D genome reconstruction, we filtered out cells with less than 100,000 unique chromatin contacts. The methods and criteria employed for generating 3D genome structures were described in detail in our original Dip-C paper (Tan et al., *Science* 2018) and subsequent publication (Tan et al., *Nat. Struct. Mol. Biol.* 2019).

To summarize, Hi-C reads were processed using hickit (<https://github.com/lh3/hickit>) and dip-c package (<https://github.com/tanlongzhi/dip-c>) following the recommended workflow outlined in the dip-c GitHub repository. This allowed us to obtain allele-imputed chromatin contacts. Each cell underwent 3D genome structure reconstruction for three independent replicates. The resulting structures were aligned, and the root-mean-square deviation (r.m.s.d) was calculated for each 20 kb particle across all three pairs of replicates using "dip-c align". Only cells with a root-mean-square

(r.m.s) (over all 20 kb particles) r.m.s.d. value below 1.5 (indicating low structural uncertainty) were considered of good quality at this resolution and utilized for structure-related analysis.

We would like to emphasize that the r.m.s.d. provides an estimate of the uncertainty associated with the 3D structure, as explained in our original Dip-C paper (Tan et al., *Science* 2018).

For the reproduction of the study and clear understanding of how the 3D genome structure analysis was performed, we thoroughly reorganized our methods and supplemented all the necessary information, please see the revised methods section “3D genome structure analysis”

4. The methods applied for phasing the data are also not clear, and a full description is required. This includes a quantification of the efficiency of phasing, together with an evaluation of whether the efficiency varies between genomic regions, and a clear discussion of how limited or uneven power to phasing contacts affects the possible outcomes of the work.

We are grateful to the Reviewers for providing valuable feedback and apologize for any confusion that may have arisen.

To address the concern regarding the phasing methods, we would like to clarify that we used our previously developed dip-c algorithm (Tan et al., *Science* 2018) to resolve the haplotypes of contacts based on SNPs. More detailed information on the haplotype imputation process can be found in the methods section "Haplotype imputation (2D)" of original Dip-C paper. The imputation accuracy was assessed to be approximately 96% through cross-validation in the original dip-c study. This imputation algorithm has also been used in others' studies (Liu et al., *Science* 2023; Li et al., *Nature Methods* 2023; Bashkirova et al., *eLife* 2023;). We have incorporated the total number of haplotype-resolved contacts for each cell into the revised supplementary tables, including a new column indicating the phase information (phased legs %), which indicates the percentage of contact reads that can be phased. We have added the detailed haplotype imputation information in the revised methods section “*Haplotype imputation of contacts*” (Page 22, line 599-603).

Furthermore, we concur with the Reviewer's observation regarding the uneven distribution of SNPs affecting the phasing and imputation of contact haplotypes. In response to this concern, we have generated a figure comparing the raw contact coverage and haplotype-resolved contact coverage at a 20-kb resolution. Our analysis revealed a strong linear relationship and high Pearson correlation ($r = 0.87$), suggesting that the efficiency varies only marginally between genomic regions at this resolution.

If the “phasing” means how to get the SNPs information, we would like to clarify that we obtained the phased SNPs data from public datasets instead of performing de novo phasing. Specifically, the SNP data of GM12878 was downloaded from the Illumina Platinum Genomes FTP site. The phased SNPs of the mouse were downloaded from the Sanger Institute Mouse Genome Project ("mcp.v5.merged.snps_all.dbSNP142.vcf.gz") and processed in accordance with the methods outlined in our previous publication (Tan L., et al, *Nat. Struct. Mol. Biol.* 2019).

Based on the Reviewers’ suggestion, we have now included a detailed description of the SNPs data and its source in the Methods section (Page 18, line 485-486).

5. The results in Figure 1g are potentially very interesting but unfortunately lack sufficient evidence. Additional information is required to understand whether the differences observed are simply due to the number of cells, and to variable power to detect contacts, included in each condition. The manuscript needs to state how many cells were included for each condition. In case there is great variation in the number of cells, then the effect of cell numbers needs to be assessed and included in the manuscript, together with analyses of sub-samples with the same numbers of cells per condition (selected randomly from each set) to test the validity of the results using different numbers of cells. The differences in contacts between the two groups should be represented by a differential matrix for better visualization. In particular, the manuscript must explicitly report whether the number of genes and UMIs per cell are comparable between cells above medium NFkB1 expression and cells below medium NFkB1 expression.

We thank the Reviewers for this valuable suggestion. We totally agree that additional information and analyses are necessary to address the concerns raised regarding the differences observed in Figure 1g.

First of most, we would like to clarify that we have taken great care in grouping the cells based on their expression levels relative to the median expression of NFkB1. As a result,

the number of cells in both the high and low expression groups is almost identical, minimizing any potential bias due to variable cell numbers.

Additionally, we have thoroughly examined the number of contacts, detected genes, and RNA counts in these two groups. We found that only the detected gene numbers was slightly higher in NFKB1 high group. To ensure that the observed differences are not solely caused by variations in the number of detected genes, we performed a downsample analysis where we controlled for gene number, RNA counts, and contact number to be identical between the two groups. However, even after controlling for these factors, we still observed differential interactions with upstream enhancers between the NFKB1 high and NFKB1 low groups (revised Extended Data Figure 3e and g). In order to further validate this observation, we randomly grouped the cells while ensuring that NFKB1 expression level, detected gene number, RNA counts, and contact number were all identical. Interestingly, the differential interactions with upstream enhancers disappeared when the cells were randomly grouped (revised Extended Data Figure 3f and h). We have included this result in our revised manuscript (Page 6, line 153-155).

As suggested, we have now included the cell number information for each condition in the revised Figure 1g and Extended Data Figure 3. To better visualize the differences in contacts between the two groups, we have also added a differential matrix heatmap (revised Extended Data Figure 3d-f).

We believe that these additional analyses and controls effectively address the concerns raised by the reviewer. Furthermore, they provide a clearer and more robust presentation of our results.

6. The methodology used to measure the compaction of NFKB1 to support the claim the authors make in lines 152-154 needs to be clearly described to enable its evaluation and reproduction.

We are grateful to the Reviewers for this constructive suggestion. We agree that it is important to provide a clear description of the methodology used to measure the compaction of the NFKB1 gene, and we apologize for any confusion or lack of detail in our original manuscript.

To address this concern, we measured the compaction of the gene by counting the normalized contact number within the gene. Moreover, we extended our analysis to include all genes that were 100 kb or larger in size. We found that more than half of the analyzed genes (57%) exhibited a similar trend of compaction as the NFKB1 gene.

However, in the interest of accuracy and clarity, we have removed the conclusion regarding the compaction of the NFKB1 gene from the manuscript (Page 6, line 153). We believe that this decision will improve the overall quality and interpretation of our findings.

7. Full details about how random controls were generated across the study (e.g. Figure 1h, i) must be included, and are essential to understand the meaning and significance of the results presented and conclusions.

We thank the Reviewers for this suggestion. To better understanding, we have included the information about how the random control is generated in each cell in the revised methods section “spatial analysis of active genes”.

8. The table presented in Figure 2b is confusing. It is not clear why two different mouse strains were used in the study, and especially why, for some time points, the data was produced from one mouse strain only. The manuscript should explicitly state the motivation for using two different mouse strains in the LiMCA experiments. A revised manuscript has to clarify or provide additional evidence to explain/test whether the differences in the downstream analysis are not due to differences in the mouse lines used. Are the samples from the same animal (for instance were the 70 cells used for P28 from one animal only)? Was the scRNA-seq data produced from 3 different developmental time points or were the samples pulled from different developmental stages?

We appreciate the valuable feedback from the Reviewers. Allow us to address your concerns and provide further clarification. Initially, our intention was to solely use the CAST/EiJ x C57BL/6J mouse strain due to its higher SNP density (20.7 million SNPs) compared to DBA/2J x C57BL/6J (5.18 million SNPs). This higher SNP density facilitates a higher successful rate in reconstruction of 3D genome structures and a more efficient differentiation of expressed OR gene alleles. Unfortunately, the CAST/EiJ x C57BL/6J cross did not yield enough offspring to cover all the required time points. Acquiring an additional batch from the Jackson Laboratory (JAX 000928) would have taken over six months, which was impractical for our study timeline. Consequently, we employed the DBA/2J x C57BL/6J strain for the remaining time points. We apologize for any confusion caused by this shift.

It is important to note that most time points did not rely on cells collected solely from a single animal or batch. In order to provide clarity, we have included a new column in the revised Supplementary Table 2 to indicate the batch information.

The generation of scRNA-seq data involved three separate 10x Chromium runs. One run, named P4-P7, consisted of pooled cells from postnatal day 4 (one male and one female) and day 7 (one male and one female). The other two samples, named P14 and P28, were derived from mice at postnatal day 14 (one male and one female) and postnatal day 28 (one male and one female), respectively. We have incorporated these details into the **Droplet scRNA-seq** methods section (Page 16, line 435-442) for better understanding.

We sincerely hope that this detailed explanation and clarification addresses the concerns of the Reviewers. Additionally, we would like to highlight that our previous 3D genome work about olfactory sensory neurons also involved these two strains of mice (Tan et al., *Nat. Struct. Mol. Biol.* 2019).

9. The results presented in Figure 2c require additional controls to rule out that the clustering of the progenitors is not due to differences in the mouse lines used or the number of contacts per cell.

We are grateful to the Reviewers for this valuable suggestion. In response to this concern, we have conducted a thorough examination of the contact numbers in the progenitor1 and progenitor2 clusters, ensuring that there is no significant difference in the number of contacts between these two clusters (see below figure **a**).

Regarding the potential influence of mouse lines, we acknowledge that our dataset contains a limited number of CAST x B6 cells, and the sampling distribution is biased (20 P7 cells, 43 P14 cells, 35 P120 cells), which may not be sufficient to fully exclude this possibility. To address this, we integrated our previously published mouse MOE data (Tan et al., *Nat. Struct. Mol. Biol.* 2019), which includes 204 B6 x CAST cells and 18 DBA2 x B6 cells. The integration of this additional dataset validated the robustness of the separation of progenitor1 and progenitor2 (see below figure **b**) and effectively ruled out the potential contribution of mouse lines to the observed clustering (see below figure **c**). Specifically, within the integrated dataset, 43 CAST x B6 cells labeled as non-neuronal in Tan et al., 2019, resulted in 9 of them being clustered to progenitor1. We have included this result in our revised figures (Extended Data Fig. 5c-d) and text (Page 7, line 189-192).

We believe that this additional analysis effectively addresses the concerns raised by the Reviewers.

10. The contribution of scRNA-Seq data produced in the study must be made clear in the manuscript. Are the cell stages 1-3 identified with LiMCA RNA also reflected in the scRNA-Seq pseudo-time? Do these 3 groups also follow the pseudo-timeline identified in the scRNA-Seq. This information can be extracted from Extended Figure 8e.

We sincerely appreciate the insightful comment from the Reviewers. In our study, we used the scRNA-seq data for two main purposes.

Firstly, we employed the scRNA-seq atlas as a reference to annotate the cell types in our METATAC atlas, as depicted in Extended Data Figure 6f. This allowed us to enhance the accuracy of cell type annotation within our dataset. We added a sentence in our revised manuscript that is “We utilized the scRNA-seq atlas as a reference to annotate the cell types in our METATAC atlas (Extended Data Fig. 6f).” (Page 7, line 205-207)

Additionally, we integrated the METATAC and scRNA-seq data to investigate the sequential relationship between OR enhancer activation and the expression of their regulating TFs, Lhx2 and Ebf. As detailed in the revised manuscript and figures (Fig. 2j-l and Extended Data Fig.8i), our analysis, including pseudotime analysis on the integrated lineage, led us to the conclusion that Lhx2/Ebf expression precedes OR enhancer accessibility. This new analysis results was updated in the revised text (Page 9, line 246 – line 254.

To further explain the stages of OSNs, we classified OSNs into three stages according to the OR gene expression pattern (Fig. 3a and Extended Data Fig. 9a-c). These stages largely correspond to the three developmental stages (progenitors, iOSNs and mOSNs) with some interchange due to the continuous developmental lineage.

11. Extended data Figure 10g, the differences in contact densities between expressed ORs and randomly selected ORs do not seem to be considerably different. A more quantitative approach is required to support the claims in lines 279-282. Further, what do numbers displayed in the top left

corner of the matrices represent?

We are grateful to the Reviewers for their valuable feedback. We apologize for any misleading caused by the inappropriate display of data in Extended Data Figure 10g. In order to address this concern, we have made the necessary changes by presenting the data in a more intuitive manner using a 3D surface plot to highlight the differences in interaction strength between the expressed OR genes and randomly selected inactive control ORs with inter-chromosomal OR enhancers (revised Extended data Figure 10k). In response to the quantification suggestion, we have also included a new plot in the revised Extended Data Figure 10j to quantitatively demonstrate the variation in interaction strength (random OR controls were randomly sampled 10 times, one sample t-test was used).

To further validate our observation, we have referred to the published bulk Hi-C data on OSNs expressing a specific OR gene (Monahan et al., *Nature* 2019). This reference was consistent with our findings as it shows that inactive ORs also interacts with inter-chromosomal OR enhancers, albeit with a weaker magnitude compared to active ORs. We believe this is a reasonable result, given that both active enhancer hubs and inactive enhancer hubs coexist in OSNs. Inactive enhancer hubs tend to interact with inactive ORs in a more random manner due to the extensive repressive OR-OR gene cluster interactions.

Minor comments

12. A schematic overview of LiMCA is presented in Figure 1a, however, some of the labels are not clear. What does DMSO 19 refer to? The example of the contact map and gene expression tracks shown is missing the genomic coordinates, are these also produced from cell 100? Please clarify.

We apologize for the confusion caused by unclear labeling in Figure 1a. We wish to clarify that the DMSO vehicle control group, consisting of 33 cells from a LiMCA experiment on

GM12878 cells treated with small molecules to perturb transcription, this information has been included in the revised Supplementary Table 1 for better referencing. Please note that the cells treated with small molecules are not included in this manuscript.

We would like to reaffirm that the exemplified contact maps and gene expression tracks (chromosome 2) were produced using the same cells (cell 100). We have made necessary modifications to Figure 1a to ensure that this label is clearer and more easily understandable.

13. On the contact maps shown in Figure 1b, black boxes and numbers are displayed. What do they represent? Please elaborate in the figure text.

We thank the Reviewers for this comment. The numbers indicate the maximum intensity of the corresponding contact map. We have supplemented this information in the revised figure legend as “The maximum intensity of the corresponding contact maps is displayed in the upper left or lower right corners.”

14. Was the bulk RNA-Seq in Figure 1d produced or obtained from published data? If produced then the protocol should be included in Methods, and if from published data then reference and data resource should be provided.

We apologize for the missing information in our manuscript. The bulk RNA-seq data presented in Figure 1d was obtained from ENCODE and can be referenced via accession number ENCFF897XES. We have included the data source information in the **Published data** methods section of our manuscript (Page 18, line 485-491).

15. The schematic in Figure 2a is not clear. Are the schematics for the olfactory epithelium supposed to represent the different developmental stages? If so, we suggest changing it either according to the number of time points used in the study (six and not three) or making the developmental timeline more fluent and visual. Also, what does ‘sc joint Hi-C-RNA’ refer to?

We sincerely appreciate the Reviewers’ valuable suggestion. To address the concern, we have made modifications to the schematic in Figure 2a to accurately represents the sampling time points, please see the revised figure.

Additionally, we would like to clarify that ‘sc joint Hi-C-RNA’ refers to single-cell joint Hi-C and RNA and we have revised it to ‘LiMCA’ for consistency throughout the manuscript.

16. There are some inconsistencies in the abbreviations used. For instance, METATAC and

single-cell ATAC-Seq are often used interchangeably, and the naming of progenitor cells is inconsistent (e.g. in Figure 2c are called progenitors and in Figure 2e are called INPs). To make it easier for the reader to follow, we highly recommend to use the same abbreviations throughout the paper. Please also include a description of the abbreviation in the figure legend when used in figures.

We appreciate the feedback provided by the Reviewers, and we fully agree with their observations. As suggested, we have now unified the name of single-cell ATAC-seq as "METATAC" in the entire manuscript.

The progenitors of olfactory sensory neurons include HBCs, GBCs and INPs, which represents different stages of differentiation. However, in the LiMCA dataset (Fig. 2c), we were unable to distinguish between these different stages of progenitors due to limitations in cell numbers. On the other hand, in the METATAC and scRNA-seq datasets, we have an adequate number of cells to discriminate between HBCs, GBCs, early INPs, and late INPs. We have taken the Reviewers' suggestion into consideration and included a comprehensive description of the abbreviations in the figure legends (Page 29, line 769-771).

17. The figure references in the text should be checked and updated. For instance, we want to draw the attention of the authors to the figure references in lines 142, 154, 163, 164, 165 and 199.

We sincerely appreciate the Reviewers' attention to detail and would like to apologize for the errors in the figure references mentioned. We have carefully reviewed and updated the figure citations in the manuscript to address these issues. We kindly ask the Reviewers to refer to the revised manuscript for the corrected figure references.

18. In lines 184-186, the authors claim that four clusters are obtained with RNA and 3C (visualised in Figure 2c). However, based on the gene expression UMAP (left), five clusters are defined; non-neuronal, progenitor-1, progenitor-2, iOSN, mOSN, while the number of clusters obtained with 3C (right) also is not completely clear. The authors should rephrase the corresponding lines to be in agreement with the UMAPs or provide additional explanations for their conclusion.

We sincerely apologize for this confusion. In RNA embedding, we identified only 4 clusters via unsupervised clustering: non-neuronal, progenitors, iOSN, mOSN. The progenitor1 and progenitor2 was indistinguishable in RNA embedding. While in 3D genome structure embedding, we observed that the progenitors cluster from the RNA embedding separated into two distinct clusters, which we named progenitor1 and progenitor2 (see below figure). Furthermore, we integrated our published mouse MOE data (Tan et al., *Nat. Struct. Mol. Biol.* 2019) to validate the separation of progenitor1 and progenitor2 in 3D genome embedding (Extended Data Fig. 5c-d). For clarity, we rewrite this paragraph, please check the revised manuscript (Page 7, line 185-192). Furthermore,

we have provided detailed information about cell type annotation in the methods section titled "Mouse olfactory cell type annotation." (Page 23, line 634-642)

19. In the figure legend for Extended Figure 3i, k: the scatterplot shows the expression level vs the normalized radial position (not the other way around as written in the legend).

We appreciate the Reviewers for pointing this out to us. We want to apologize for any confusion this may have caused. We have made the necessary correction in the corresponding figure legend.

20. In Extended Figure 4a the discrete labels that the authors propose for each group of progenitors do not correspond to the developmental trajectory from the pseudo-time analysis (Extended Figure 4 d). Could the authors comment on why this is the case?

We thank the Reviewers for this comment. This comment is related to comment 18. In RNA embedding, the progenitor1 and progenitor2 cannot be distinguished. We have modified the corresponding figures, including Fig. 2c and Extended Data Fig. 4a, d and e to avoid confusion.

21. The explanation for how the pseudo-time in the ATAC-seq dataset was performed is missing. Please clarify how early and late INPs were defined in the ATAC-seq data and why these stages are not represented in the scRNA-seq pseudo-time.

We thank the Reviewers for this comment. In response to this concern, we have added a section in revised methods to add the details of METATAC pseudo-time analysis. The early and late INP in METATAC data were identified via unsupervised clustering. In scRNA-seq data, when we extract the OSN lineage (from GBC to mOSN) and perform subclustering, we were also able to identify the early and late INP cell clusters (Extended Data Figure 8e left and middle panel).

22. The authors should be more specific about which cell types were used for each of the analyses. If missing, please indicate either in the text, on the figures or in the figure legend.

We totally agree with the Reviewers for this suggestion. We have added the cell type information in the corresponding text and figure legend. Please refer to the revised manuscript (Page 6, line 237-239; Page 10, line 273-274).

23. The colour coding is inconsistent and several plots/markings are missing labels. For instance, please specify what the dots/lines in Figure 1g and Extended Figure 3c-f correspond to, and be consistent with colours (yellow in Figure 1g vs blue in Extended Figure 3c-f). Furthermore, it is not clear what the different colours of the OR enhancers represent (are the OR enhancers from different chromosomes?) (Figure 2d and Figure 3b,d-e). Extended Figure 4b and Extended Figure 10a are missing axis labels. Figure 2g is missing a control to show that the enrichment is specific for the newly identified candidate peaks.

We greatly appreciate the Reviewers' feedback and apologize for any confusion caused by inconsistent color coding and missing labels in our figures.

Based on the Reviewers' suggestions, we have made significant revisions to address these issues. In Figure 1g and Extended Figure 3i-j, we have now included a comprehensive figure legend that explicitly explains the meaning of the green and yellow dots/bars. Specifically, the green dot represents potential enhancers, while the yellow dots represent the transcription start site (TSS) or transcription termination site (TTS) of the gene. Additionally, we have ensured consistency in color coding between Figure 1g and Extended Figure 3c-f.

Yes, the colors of OR enhancers represent different chromosomes. We have taken into consideration the suggestion from another reviewer and modified the colors to form a sequential color set from blue to red, which corresponds to different chromosomes. Please refer to the revised Figure 2d and Figure 3b,d-e for the updated color scheme.

Furthermore, we have added axis labels to Extended Figure 4b and Extended Figure 10a to improve clarity and ensure a better understanding of the data.

Finally, as advised, we have included a control to show the significant enrichment of composite motif in candidate OR enhancers. Please see Fig. 2g. Specifically, we selected other ATAC peaks residing in OR gene clusters and ATAC peaks outside of OR gene clusters to showcase the depletion of the composite motif. Previously identified Greek islands have slightly higher Lhx2/Ebf composite motif scores than our newly identified candidate OR enhancers.

24. Extended Figure 9 lacks a figure legend and isoform labelling in the figure. Please provide. **We thank the Reviewers for this suggestion. In response to this feedback, we have updated Extended Data Figure 9 with a clear figure legend and added labeling for isoforms and alleles. We hope that this revised figure provides the necessary information and clarity for the readers.**
25. The text and figures should be checked for general typos. To highlight some: in Figure 1a (right) and Extended Figure 1h “CpG Frenquency” should be changed to “CpG Frequency”; in figure text for Extended Figure 3h “Scatterplot” should be changed to “Dot plot”; Extended Figure 4f the y-axis should be changed from 300nm to 150nm; in the figure text for Extended Figure 5d,e “scatterplot” should be changed to “boxplot”.

Thank you for figuring out these typos. We have corrected the typos in the corresponding figures and text.

Reviewer #3:

Remarks to the Author:

The manuscript by Xie and colleague describes a powerful new protocol called LimCA (Linking mRNA to Chromatin Architecture) that enables multiome single cell analysis of mRNA and genomic contacts. Since the emergence of single cell HiC technologies, there is an increasing need to combine genomic contacts with RNA expression information. A recently published protocol, termed HiRES, provided the first such methodology, with LimCA emerging as a significant improvement of coverage both at the HiC (with an impressive 1million contacts per cells) and the RNA detection front. The authors validate their methodology in commonly used cell lines, allowing direct comparison with other protocols and then they apply this protocol to the mouse olfactory neurons. These neurons

provide the ideal biological system for LimCA, since previous work established an intimate connection between the assembly of multi-enhancer hubs and the transcriptional activation of one out of ~1000 olfactory receptor genes. Using this system, the authors make several important observations that highlight the power of their new technology. Specifically:

1. They show that at the early polygenic state, co-transcribed olfactory receptor genes associate with enhancer hubs preferentially consisting of enhancers from the same chromosome.
2. They show that among the competing olfactory receptor/hub combinations, usually the gene that has the higher number of enhancers in close proximity is the one that is more highly transcribed, providing a striking demonstration of the synergy between these enhancers.
3. They show that in mature olfactory neurons, the prevailing gene is often not associated with the hub that has the most enhancer, a puzzling finding considering point#2.
4. They also perform scATAC/RNA-seq experiments that reveal interesting dynamics between enhancer accessibility, transcription factor expression and hub assembly. From the biological perspective, these are important new discoveries that indeed clarify that process of olfactory receptor expression and suggest a positive feedback mechanism as a potential mechanism that transforms polygenic transcription to singular transcription in terminally differentiated neurons. From the technical perspective, it appears that this approach has the potential to surpass HiRES, providing a powerful tool for the general community. Thus, I am in favor of publication. However, I want to propose a few changes, clarifications and new analyses.

We sincerely appreciate the Reviewers' positive feedback on our manuscript and their recognition of the significance and value of our work. We are grateful for the Reviewers' insightful suggestions, which undoubtedly strengthen our manuscript and enhance its clarity. We sincerely thank the Reviewer for their enthusiastic support, kind words, and valuable feedback.

1. The authors describe the discovery of numerous new olfactory receptor enhancers that have the same motif as the previously described enhancers, and they are co-bound by Lhx2 and Ebf. I did not find description of ChIP-seq experiments, thus I am not sure where is the evidence that these new putative enhancers are co-bound by these two factors. Are they using the ChIP-seq data from Monahan et al? If yes, and these elements are co-bound by these two TFs and have a composite motif, why they were not called as such by these authors?

We sincerely appreciate the Reviewers for raising this question and providing valuable feedback. We apologize for any confusion caused regarding the evidence of co-binding of Lhx2 and Ebf, as well as the source of the Lhx2/Ebf ChIP-seq data.

To clarify, we obtained the Lhx2/Ebf ChIP-seq data from the study by Monahan et al. (*eLife* 2017). We have now included this information in both the figure legend and the Methods section to ensure transparency and accuracy in our manuscript.

In terms of evidence for the co-binding of these two transcription factors (TFs) at the newly identified enhancers, as shown in Figure 2f, which demonstrates their co-binding pattern. It is important to note that the relative scale of each track has been normalized to the highest value within individual olfactory receptor (OR) gene clusters in Figure 2f. Consequently, some putative or previously identified enhancers may not appear to meet the co-binding criteria due to relatively lower values compared to stronger enhancers within the same OR gene cluster. For clarification, we have included a heatmap to show the details of individual enhancer in Extended Data Figure 7c.

Addressing the question of why these peaks were not identified in the previous work (Monahan et al. *eLife* 2017), it is important to highlight that our study benefits from single-cell chromatin accessibility data, which provides a higher signal-to-noise ratio compared to bulk ATAC-seq data. To support this claim, we have exemplified the presence of these enhancers for two specific enhancers and provided heatmaps displaying all enhancers (see below).

We must acknowledge that the candidate OR enhancers exhibit relatively weaker Lhx2/Ebf binding compared to previously identified Greek islands. This discrepancy may be attributed to the relatively rare populations of olfactory sensory neurons (OSNs) possessing these accessible enhancers. According to our METATAC results, OR enhancers exhibit the highest accessibility at the late INP stage, which gives rise to our speculation that Lhx2/Ebf might exhibit stronger binding at these enhancers. However, due to the unavailability of the corresponding mouse strain, we were unable to validate this assumption experimentally. We sincerely hope that this comprehensive explanation adequately addresses the question raised by the reviewers and resolves any concerns they may have had.

2. I am trying to think of explanations for the fact that the hub that is associated with the chosen OR allele is not the biggest hub, which would be expected if the positive feedback loop is correct. One explanation I can think, is that the zonal properties of the olfactory receptors in the biggest hubs are more dorsal than the identity of the prevailing olfactory receptor: According to Bashkirova et al, the biggest hubs they would contain more dorsal receptor genes, they would have formed earlier, and they would subsequently become heterochromatic. Since the authors have zonal information of every receptor, they could easily explore this interesting possibility which would add extra biological value to their work. Even if this hypothesis is wrong, it would be very informative if the authors do a zonal analysis in their data (for example are the Zonal restrictions described by Bashkirova in late stages of polygenic expression related to hub interactions?)

We are grateful to the Reviewers for their constructive suggestion. The consideration of OR zone identity is indeed intriguing and could potentially explain why the finally chosen OR does not reside within the largest enhancer hub, as expected in the context of a positive feedback loop.

To investigate this hypothesis, we performed an analysis of OR zone identity in stage 2 and stage 3 OSNs with a dominant OR. We specifically compared the zone identity of the dominant OR with ORs residing in the largest OR enhancer hub and second largest OR enhancer hub within the same cell. Our findings revealed a significant difference in zone

identity between the dominant OR and the ORs residing in the largest or second largest enhancer hub (see below). Indeed, the biggest enhancer hub typically contains more dorsal ORs, which indicates that the largest enhancer hub tends to be inactive.

We have meticulously incorporated this result into our revised manuscript by including the updated figure and discussing these findings in the revised discussion section (Page 13, line 365-377).

Once again, we sincerely thank the reviewer for this insightful suggestion, which has enhanced the biological value of our work.

- I would have loved to see LimCA performed on olfactory neurons selected based on the expression of a GFP reporter driven by an olfactory receptor gene (ORiresGFP), not only as a confirmation for the method but also to obtain an understanding of the false negative and false positive rates of the technique. However, if such lines are not available, I would not wish to delay publication for this control experiment.

We thank the Reviewers for their valuable feedback and suggestion. We fully agree with the Reviewers that performing LiMCA on OSNs expressing a specific OR gene would serve as a strong validation of our technique. However, unfortunately, such mice lines are not currently available for use, and cryorecovery of these mouse strains from Jaxson Laboratory (JAX:021206 (Olf160), JAX:006643 (Olf16), JAX:007767 (Olf17), JAX:006638 (Olf155), JAX:007762 (Olf15)) would take more than one year. Thus, we are not able to perform LiMCA to OSNs expressing a known OR gene within the expected date.

Despite this limitation, we took several steps to ensure the accurate pairing of gene expression and 3D genome information of individual cells. Specifically, we meticulously labeled the DNA and RNA for each single cell during sample preparation and library preparation to minimize any potential mismatches. Additionally, we observed excellent concordance between the cell typing by Hi-C and RNA, both in human cell lines and olfactory sensory neurons, which provided further confidence in our methodology. Finally, we performed a permutation analysis to test whether the observed interactions between

expressed ORs and inter-chromosomal enhancers could have occurred by random chance pairing between OR expression and 3D genome. The results demonstrated that the strong and specific interaction between expressed ORs and inter-chromosomal enhancers disappeared when we randomly mismatched the expressed OR and 3D genome (below **a**). Moreover, the differential association with OR enhancers between dominant OR and second highest expressed OR gene also vanished (below **b**). These findings further reinforce the validity of our results.

While we acknowledge the importance of validating our technique with LiMCA on OSNs expressing a specific OR gene, the unavailability of appropriate mouse strains prevented us from performing this experiment at this time. We genuinely hope that this revised analysis persuades the Reviewers regarding the robustness and validity of our methodology.

Decision Letter, first revision:

Our ref: NMETH-A53325A

20th Dec 2023

Dear Dr. Xie,

Thank you for submitting your revised manuscript "Simultaneous single-cell three-dimensional genome and gene expression profiling Uncovers Dynamic Enhancer Connectivity Underlying Olfactory Receptor Choice" (NMETH-A53325A). It has now been seen by the original referees and their comments are below. The reviewers find that the paper has improved in revision, and therefore we'll be happy in principle to publish it in Nature Methods, pending minor revisions to satisfy the referees' final requests and to comply with our editorial and formatting guidelines.

We are now performing detailed checks on your paper and will send you a checklist detailing our editorial and formatting requirements within two weeks or so. Please be aware that the holiday season may delay this process. Please do not upload the final materials and make any revisions until you receive this additional information from us.

TRANSPARENT PEER REVIEW

Please note: we allow redactions to authors' rebuttal and reviewer comments in the interest of confidentiality. If you are concerned about the release of confidential data, please let us know specifically what information you would like to have removed. Please note that we cannot incorporate redactions for any other reasons. Reviewer names will be published in the peer review files if the reviewer signed the comments to authors, or if reviewers explicitly agree to release their name. For more information, please refer to our FAQ page.

ORCID

Sincerely,
Lei

Lei Tang, Ph.D.
Senior Editor
Nature Methods

Reviewer #1 (Remarks to the Author):

The authors have addressed all of my previous concerns. As stated, I think the paper is important and timely.

Minor comment: I still find the display of genetic loci in Figures 2d and 3b,d to be confusing. It is not immediately clear how the rainbow color palette legend relates to the spheres. The reader has to

understand that the grey spheres are replaced with spheres shaded with the indicated colors. That is not the way figure legends are typically interpreted. The legend should show the actual data markers in the figure. Or at least a subset. Maybe just change the strip of colored boxes into a set of colored spheres? Also, the data markers in the legend should match more closely the ones actually plotted.

Reviewer #1 (Remarks on figshare data availability):

The data appear suitable and accurate

Reviewer #2

The authors have done a great effort by providing more comprehensive details, specifically in the method section, effectively addressing most of the primary concerns.

Major comments:

Comment 4:

The methods applied for phasing the data are also not clear, and a full description is required. This includes a quantification of the efficiency of phasing, together with an evaluation of whether the efficiency varies between genomic regions, and a clear discussion of how limited or uneven power to phasing contacts affects the possible outcomes of the work.

[...] In response to this concern, we have generated a figure comparing the raw contact coverage and haplotype-resolved contact coverage at a 20-kb resolution. Our analysis revealed a strong linear relationship and high Pearson correlation ($r = 0.87$), suggesting that the efficiency varies only marginally between genomic regions at this resolution. [...]

I acknowledge the authors' work to clarify the methodology followed to phasing the data. However, the correlation between haplotype-resolved contact coverage and overall contact coverage remains unclear. Specifically, the units for both metrics are missing. Does the X-axis indicate that the maximum coverage corresponds to four contacts? For the manuscript to be fit for publication, the missing details must be added in the figure showing the correlation.

Comment 5:

The results in Figure 1g are potentially very interesting but unfortunately lack sufficient evidence. Additional information is required to understand whether the differences observed are simply due to the number of cells, and to variable power to detect contacts, included in each condition. The manuscript needs to state how many cells were included for each condition. In case there is great variation in the number of cells, then the effect of cell numbers needs to be assessed and included in the manuscript, together with analyses of sub-samples with the same numbers of cells per condition (selected randomly from each set) to test the validity of the results using different numbers of cells. The differences in contacts between the two groups should be represented by a differential matrix for better visualization. In particular, the

manuscript must explicitly report whether the number of genes and UMIs per cell are comparable between cells above medium NFkB1 expression and cells below medium NFkB1 expression.

[...] In order to further validate this observation, we randomly grouped the cells while ensuring that NFkB1 expression level, detected gene number, RNA counts, and contact number were all identical. Interestingly, the differential interactions with upstream enhancers disappeared when the cells were randomly grouped [...]

The total number of cells per NFkB1 high and low expression groups (108 and 111, respectively) appear to be relatively low to support the claim that cells expressing higher levels of NFkB1 engage in the observed enhancer-promoter interactions which are absent in cells with lower NFkB1 expression. While the differences between the NFkB1-high and NFkB1-low expression groups seem convincing, compared to the randomized data, more iterations of randomized data have to be included to support that the enhancer-promoter interactions are specific to cells expressing higher levels of NFkB1.

Additionally, plotting the differential matrix as a mirror is misleading and redundant, as both triangles are identical. Plotting one unique triangle for the differential matrix will enhance the clarity and interpretation of the claims.

Comment 9:

The results presented in Figure 2c require additional controls to rule out that the clustering of the progenitors is not due to differences in the mouse lines used or the number of contacts per cell.

[...] To address this, we integrated our previously published mouse MOE data (Tan et al., Nat. Struct. Mol. Biol. 2019), which includes 204 B6 x CAST cells and 18 DBA2 x B6 cells. The integration of this additional dataset validated the robustness of the separation of progenitor1 and progenitor2 (see below figure b) and effectively ruled out the potential contribution of mouse lines to the observed clustering (see below Figure c). Specifically, within the integrated dataset, 43 CAST x B6 cells labeled as non-neuronal in Tan et al., 2019, resulted in 9 of them being clustered to progenitor1. We have included this result in our revised figures (Extended Data Fig. 5c-d) and text (Page 7, line 189-192) [...]

The additional plots provided in the revision letter support that the cell clustering is not due to biological bias (the use of two different mouse strains). However, to confirm no biological bias or batch effects it is important to plot the B6xDBA2 data sets (from current study and Tan et al 2019) in distinct colors as it will more clearly support the claim.

Furthermore, the extended data figures 5c-d are of bad quality as they are difficult to deduce anything from. They must be redone or substituted with the versions used in the revision letter (see comparison below).

Current revised Extended data figure 5c-d:

labs(title = glue("{group}: TF-Motifs enriched in {group1}")) +
 Figures from revision letter:

Minor comments:

In general, all minor comments have been addressed satisfactorily, although a few remarks remain to be addressed.

Comment 15:

The schematic in Figure 2a is not clear. Are the schematics for the olfactory epithelium supposed to represent the different developmental stages? If so, we suggest changing it either according to the number of time points used in the study (six and not three) or making the developmental timeline more fluent and visual. Also, what does 'sc joint Hi-C-RNA' refer to?

We sincerely appreciate the Reviewers' valuable suggestions. To address the concern, we have made modifications to the schematic in Figure 2a to accurately represent the sampling time points, please see the revised figure [...]

It is now unclear (and a bit concerning) whether the samples were taken on postnatal day 3 or 4, as this is inconsistent in figures and text. The authors must coherently report from which postnatal day were the samples taken.

Comment 23:

The colour coding is inconsistent and several plots/markings are missing labels. For instance, please specify what the dots/lines in Figure 1g and Extended Figure 3c-f correspond to, and be consistent with colours (yellow in Figure 1g vs blue in Extended Figure 3c-f). Furthermore, it is not clear what the different colours of the OR enhancers represent (are the OR enhancers from different chromosomes?) (Figure 2d and Figure 3b,d-e). Extended Figure 4b and Extended Figure 10a are missing axis labels. Figure 2g is missing a control to show that the enrichment is specific for the newly identified candidate peaks.

[...] Based on the Reviewers' suggestions, we have made significant revisions to address these issues. In Figure 1g and Extended Figure 3i-j, we have now included a comprehensive figure legend that explicitly explains the meaning of the green and yellow dots/bars. Specifically, the green dot represents potential enhancers, while the yellow dots represent the transcription start site (TSS) or transcription termination site (TTS) of the gene. Additionally, we have ensured consistency in color coding between Figure 1g and Extended Figure 3c-f [...]

The colors used in extended figure 3f have not been changed according to our previous recommendation. This is important as it is detrimental to correctly interpret the figure and conclusions drawn from it.

[...] Finally, as advised, we have included a control to show the significant enrichment of composite motifs in candidate OR enhancers. Please see Fig. 2g. Specifically, we selected other ATAC peaks residing in OR gene clusters and ATAC peaks outside of OR gene clusters to showcase the depletion of the composite motif. Previously identified Greek islands have slightly higher Lhx2/Ebf composite motif scores than our newly identified candidate OR enhancers.

Figure 2f in original version:

Figure 2g in revised manuscript:

The p -value (*Lhx2/Ebf* composite motif scores of previously identified Greek Islands vs newly identified candidate OR enhancers) was in the original manuscript found to be $p=0.22$ and has now been changed to $p=0.023$ in the revised manuscript, however the change from non-significant to significant is not commented in the revised manuscript. It is clear that the analysis of composite motif score is visualized differently in the revised manuscript, however, was the p -value calculated differently in the revised version (it is not stated in the original version how the p -value was obtained)? Which p -value is the correct one, was it a mistake in the previous version? Please verify and correct in revised manuscript if required.

Reviewer #3 (Remarks to the Author):

The authors did a great job responding to the critiques from the original submission. I find the new

analysis, showing that the largest hub in each nucleus representing a locus of transcription of more dorsal OR genes, very interesting and consistent with recent reports describing mechanisms of OR regulation across the dorsoventral axis of the olfactory epithelium. A minor comment is that I could not find a reference from the Lomvardas group proposing that the active OR is associated with the largest enhancer hub in each olfactory neuron. If the authors do not cite such claim, they should remove it from the document. Overall, LimCA is a powerful technology, and this article provides critical insight to the mechanisms of olfactory receptor gene regulation. I support publication

Reviewer #3 (Remarks on figshare data availability):

Data are in agreement with the claims of the article.

Author Rebuttal, first revision:

Reviewer #1 (Remarks to the Author):

The authors have addressed all of my previous concerns. As stated, I think the paper is important and timely.

Minor comment: I still find the display of genetic loci in Figures 2d and 3b,d to be confusing. It is not immediately clear how the rainbow color palette legend relates to the spheres. The reader has to understand that the grey spheres are replaced with spheres shaded with the indicated colors. That is not the way figure legends are typically interpreted. The legend should show the actual data markers in the figure. Or at least a subset. Maybe just change the strip of colored boxes into a set of colored spheres? Also, the data markers in the legend should match more closely the ones actually plotted.

Response: We thank the Reviewers for the kind suggestion. We have modified the label of enhancers in Figure 2d, 3b, d and e as suggested. Please check the revised Figures.

Reviewer #2:

The authors have done a great effort by providing more comprehensive details, specifically in the method section, effectively addressing most of the primary concerns.

Major comments:

Comment 4:

The methods applied for phasing the data are also not clear, and a full description is required. This includes a quantification of the efficiency of phasing, together with an evaluation of whether the efficiency varies between genomic regions, and a clear discussion of how limited or uneven power to phasing contacts affects the possible outcomes of the work.

[...] In response to this concern, we have generated a figure comparing the raw contact coverage and haplotype-resolved contact coverage at a 20-kb resolution. Our analysis revealed a strong linear relationship and high Pearson correlation ($r = 0.87$), suggesting that the efficiency varies only marginally between genomic regions at this resolution. [...]

I acknowledge the authors' work to clarify the methodology followed to phasing the data. However, the correlation between haplotype-resolved contact coverage and overall contact coverage remains unclear. Specifically, the units for both metrics are missing. Does the X-axis indicate that the maximum coverage corresponds to four contacts? For the manuscript to be fit for publication, the missing details must be added in the figure showing the correlation.

Response: We thank the Reviewers for this comment. We have modified the axis label of this figure. Please reviewed the revised figure shown below.

Comment 5:

The results in Figure 1g are potentially very interesting but unfortunately lack sufficient evidence. Additional information is required to understand whether the differences observed are simply due to the number of cells, and to variable power to detect contacts, included in each condition. The manuscript needs to state how many cells were included for each condition. In case there is great variation in the number of cells, then the effect of cell numbers needs to be assessed and included in the manuscript, together with analyses of sub-samples with the same numbers of cells per condition (selected randomly from each set) to test the validity of the results using different numbers of cells. The differences in contacts between the two groups should be represented by a differential matrix for better visualization. In particular, the manuscript must explicitly report whether the number of genes and UMIs per cell are comparable between cells above medium NFKB1 expression and cells below medium NFKB1 expression.

[...] In order to further validate this observation, we randomly grouped the cells while ensuring that NFKB1 expression level, detected gene number, RNA counts, and contact number were all identical. Interestingly, the differential interactions with upstream enhancers disappeared when the cells were randomly grouped [...]

The total number of cells per NFKB1 high and low expression groups (108 and 111, respectively) appear to be relatively low to support the claim that cells expressing higher levels of NFKB1 engage in the observed enhancer-promoter interactions which are absent in cells with lower NFKB1 expression. While the differences between the NFKB1-high and NFKB1-low expression groups seem convincing, compared to the randomized data, more iterations of randomized data have to be included to support that the enhancer-promoter interactions are specific to cells expressing higher levels of NFKB1.

Additionally, plotting the differential matrix as a mirror is misleading and redundant, as both triangles are identical. Plotting one unique triangle for the differential matrix will enhance the clarity and interpretation of the claims.

Response: We sincerely appreciate the Reviewers for the valuable comment. We have totally performed the randomization three times, which further confirmed the difference observed between *NFKB1-high* and *NFKB1-low* expression groups.

And we changed all the differential matrix into triangle plot to ensure clarity.

Comment 9:

The results presented in Figure 2c require additional controls to rule out that the clustering of the progenitors is not due to differences in the mouse lines used or the number of contacts per cell.

[...] To address this, we integrated our previously published mouse MOE data (Tan et al., Nat. Struct. Mol. Biol. 2019), which includes 204 B6 x CAST cells and 18 DBA2 x B6 cells. The integration of this additional dataset validated the robustness of the separation of progenitor1 and progenitor2 (see below figure b) and effectively ruled out the potential contribution of mouse lines to the observed clustering (see below Figure c). Specifically, within the integrated dataset, 43 CAST x B6 cells labeled as non-neuronal in Tan et al., 2019, resulted in 9 of them being clustered to progenitor1. We have included this result in our revised figures (Extended Data Fig. 5c-d) and text (Page 7, line 189-192) [...]

The additional plots provided in the revision letter support that the cell clustering is not due to biological bias (the use of two different mouse strains). However, to confirm no biological bias or batch effects it is important to plot the B6xDBA2 data sets (from current study and Tan et al 2019) in distinct colors as it will more clearly support the claim.

Furthermore, the extended data figures 5c-d are of bad quality as they are difficult to deduce anything from. They must be redone or substituted with the versions used in the revision letter (see comparison below).

Current revised Extended data figure 5c-d:

Figures from revision letter:

Response: We thank the Reviewers for this comment. Based on the suggestion, we have made improvements to the visualization of the embedding by incorporating different colors to distinguish between mouse strains in both our dataset and the dataset from *Tan et al 2019*.

Additionally, we have replaced the previous Extended Data Figure 5c-d as suggested. We kindly request you to take a look at the revised Figures.

Minor comments:

In general, all minor comments have been addressed satisfactorily, although a few remarks remain to be addressed.

Comment 15:

The schematic in Figure 2a is not clear. Are the schematics for the olfactory epithelium supposed

to represent the different developmental stages? If so, we suggest changing it either according to the number of time points used in the study (six and not three) or making the developmental timeline more fluent and visual. Also, what does 'sc joint Hi-C-RNA' refer to?

We sincerely appreciate the Reviewers' valuable suggestions. To address the concern, we have made modifications to the schematic in Figure 2a to accurately represent the sampling time points, please see the revised figure [...]

It is now unclear (and a bit concerning) whether the samples were taken on postnatal day 3 or 4, as this is inconsistent in figures and text. The authors must coherently report from which postnatal day were the samples taken.

Response: We would like to express our gratitude to the Reviewers for their valuable comment. We deeply apologize for any confusion that may have arisen regarding the timing of sample collection. Allow us to provide clarification on this matter.

For the LiMCA and METATAC datasets, all samples were taken from mice on postnatal day 3. It is important to note that there was only one sample for the scRNA-seq analysis was obtained from postnatal day 4 mice. These details were included in the methods section.

Comment 23:

The colour coding is inconsistent and several plots/markings are missing labels. For instance, please specify what the dots/lines in Figure 1g and Extended Figure 3c-f correspond to, and be consistent with colours (yellow in Figure 1g vs blue in Extended Figure 3c-f). Furthermore, it is not clear what the different colours of the OR enhancers represent (are the OR enhancers from different chromosomes?) (Figure 2d and Figure 3b,d-e). Extended Figure 4b and Extended Figure 10a are missing axis labels. Figure 2g is missing a control to show that the enrichment is specific for the newly identified candidate peaks.

[...] Based on the Reviewers' suggestions, we have made significant revisions to address these issues. In Figure 1g and Extended Figure 3i-j, we have now included a comprehensive

figure legend that explicitly explains the meaning of the green and yellow dots/bars. Specifically, the green dot represents potential enhancers, while the yellow dots represent the transcription start site (TSS) or transcription termination site (TTS) of the gene. Additionally, we have ensured consistency in color coding between Figure 1g and Extended Figure 3c-f [...]

The colors used in extended figure 3f have not been changed according to our previous recommendation. This is important as it is detrimental to correctly interpret the figure and conclusions drawn from it.

Response: We would like to express our sincere appreciation to the Reviewers for their keen observation. We apologize for the oversight in not implementing the previous recommendation regarding the colors used in extended figure 3f. We fully acknowledge the significance of ensuring consistency in color representation to accurately interpret the figure and draw appropriate conclusions from it.

We want to assure the Reviewers that we have now rectified this error by making the necessary adjustments to the color scheme in extended figure 3f.

[...] Finally, as advised, we have included a control to show the significant enrichment of composite motifs in candidate OR enhancers. Please see Fig. 2g. Specifically, we selected other ATAC peaks residing in OR gene clusters and ATAC peaks outside of OR gene clusters to showcase the depletion of the composite motif. Previously identified Greek islands have slightly higher Lhx2/Ebf composite motif scores than our newly identified candidate OR enhancers.

Figure 2f in original version:

Figure 2g in revised manuscript:

The *p*-value (*Lhx2/Ebf* composite motif scores of previously identified Greek Islands vs newly identified candidate OR enhancers) was in the original manuscript found to be $p=0.22$ and has now been changed to $p=0.023$ in the revised manuscript, however the change from non-significant to significant is not commented in the revised manuscript. It is clear that the analysis of composite motif score is visualized differently in the revised manuscript, however, was the *p*-value calculated differently in the revised version (it is not stated in the original version how the *p*-value was obtained)? Which *p*-value is the correct one, was it a mistake in the previous version? Please verify and correct in revised manuscript if required.

Response: We sincerely appreciate the valuable feedback provided by the Reviewers. In response to their question regarding the calculation of *p*-values, we would like to clarify the methodology employed in the revised manuscript.

Basically, we utilized a different statistical method to calculate the *p*-values. In the previous manuscript, we identified the segments that matched the composite motif in the Greek islands (GIs) and the newly identified candidate peaks (with a *q*-value < 0.1 from the FIMO results). Subsequently, we compared the *q*-values between these two groups using Wilcoxon rank-sum tests.

In the revised manuscript, however, we made a modification to the analysis approach. Instead of applying the *q*-value filter, we retained all the results from FIMO. This allowed us to compare the distribution of FIMO motif scores between different groups using Kolmogorov-Smirnov tests. We believe that this adjustment provides a more elegant and precise means of comparing the motif signal between the Greek Islands and the newly identified candidate peaks.

We want to emphasize that both versions of the *p*-values are correct, and the change in methodology led to the observed difference in *p*-values. We apologize for not explicitly mentioning this change from non-significant to significant in the revised manuscript. To rectify this oversight, we will ensure that the revised

manuscript includes a clear explanation of the updated methodology and the resulting change in significance.

Reviewer #3 (Remarks to the Author):

The authors did a great job responding to the critiques from the original submission. I find the new analysis, showing that the largest hub in each nucleus representing a locus of transcription of more dorsal OR genes, very interesting and consistent with recent reports describing mechanisms of OR regulation across the dorsoventral axis of the olfactory epithelium. A minor comment is that I could not find a reference from the Lomvardas group proposing that the active OR is associated with the largest enhancer hub in each olfactory neuron. If the authors do not cite such claim, they should remove it from the document. Overall, LimCA is a powerful technology, and this article provides critical insight to the mechanisms of olfactory receptor gene regulation. I support publication

Response: We sincerely appreciate the Reviewers for this valuable comment. We apologize for the unprecise language describing that “the active OR is associated with the largest enhancer hub in each olfactory neuron.” We have now removed the corresponding text in the revised manuscript.

Page 3 line68 “rather than a singular one proposed based on bulk Hi-C data^{32, 37}” to “which means that solely being associated with enhancer hubs is insufficient to fully account for the singular OR gene.”

We remove Page 11 line 314-315 “It was proposed that the chosen OR interacts with a multi-chromosomal super-enhancer hub consisting of all Greek islands^{32, 37}.” And modified it to “Single-cell Hi-C on OSNs showed that each OSN harbors multiple enhancer aggregates and proposed that the active OR presumably resides in the largest enhancer aggregates according to the bulk observations¹⁸.”

We removed Page12 line 356-357 “contradicts the previously proposed model based on bulk 4C/Hi-C experiments^{17, 32}”

Final Decision Letter:

7th Mar 2024

Dear Professor Xie,

I am pleased to inform you that your Article, "Simultaneous single-cell three-dimensional genome and gene expression profiling Uncovers Dynamic Enhancer Connectivity Underlying Olfactory Receptor

Choice", has now been accepted for publication in Nature Methods. The received and accepted dates will be 27th Jul 2023 and 7th Mar 2024. This note is intended to let you know what to expect from us over the next month or so, and to let you know where to address any further questions.

Over the next few weeks, your paper will be copyedited to ensure that it conforms to Nature Methods style. Once your paper is typeset, you will receive an email with a link to choose the appropriate publishing options for your paper and our Author Services team will be in touch regarding any additional information that may be required. It is extremely important that you let us know now whether you will be difficult to contact over the next month. If this is the case, we ask that you send us the contact information (email, phone and fax) of someone who will be able to check the proofs and deal with any last-minute problems.

Please note that *Nature Methods* is a Transformative Journal (TJ). Authors may publish their research with us through the traditional subscription access route or make their paper immediately open access through payment of an article-processing charge (APC). Authors will not be required to make a final decision about access to their article until it has been accepted. Find out more about Transformative Journals

To assist our authors in disseminating their research to the broader community, our SharedIt initiative provides you with a unique shareable link that will allow anyone (with or without a subscription) to read the published article. Recipients of the link with a subscription will also be able to download and

print the PDF.

If you are active on Twitter/X, please e-mail me your and your coauthors' handles so that we may tag you when the paper is published.

Best regards,
Lei

Lei Tang, Ph.D.
Senior Editor
Nature Methods